# Characterisation of aerosol size properties from measurements of spectral optical depth: a global validation of the GRASP-AOD code using long-term AERONET data.

Benjamin Torres[1] and David Fuertes[2]

[1]Univ. Lille, CNRS, UMR 8518 - LOA - Laboratoire d'Optique Atmosphérique, F-59000 Lille, France
[2]GRASP-SAS, Univ. Lille, Villeneuve d'Ascq, France

**Correspondence:** Dr. Benjamin Torres
(benjamin.torres@univ-lille.fr)

**Abstract.**

A validation study is conducted regarding aerosol optical size property retrievals from only measurements of the direct Sun beam (without the aid of diffuse radiation). The study focuses on testing with real data the new GRASP-AOD application which uses only spectral optical depth measurements to retrieve the total column aerosol size distributions, assumed as bimodal log-normal. In addition, a set of secondary integral parameters of aerosol size distribution and optical properties are provided: effective radius, total volume concentration and fine mode fraction of aerosol optical depth. The GRASP-AOD code is applied to almost three million observations acquired during twenty years (1997-2016) at thirty AERONET (Aerosol Robotic Network) sites. These validation sites have been selected based on known availability of an extensive data record, significant aerosol load variability along the year, wide worldwide coverage and diverse aerosol types and source regions. The output parameters are compared to those coming from the operational AERONET retrievals. The retrieved fine mode fractions at 500 nm ($\tau_f(500)$) obtained by GRASP-AOD application are compared to those retrieved by the Spectral Deconvolution Algorithm and by AERONET aerosol retrieval algorithm. The size distribution properties obtained by GRASP-AOD are compared to their equivalent values from the AERONET aerosol retrieval algorithm. The analysis showed the convincing capacity of GRASP-AOD approach to successfully discriminate between fine and coarse mode extinction to robustly retrieve $\tau_f(500)$. The comparisons of 2 million results of $\tau_f(500)$ retrieval by GRASP-AOD and SDA showed high correlation with a root-mean-square-error (RMSE) of 0.015. Also, the analysis showed that the $\tau_f(500)$ values computed by AERONET aerosol retrieval algorithm agree slightly better with GRASP-AOD (RMSE=0.018, from 148526 comparisons) than with SDA (RMSE=0.022, from 127203 comparisons). The comparisons of the size distribution retrieval showed the agreement for fine mode median radius between GRASP-AOD and AERONET aerosol retrieval algorithm results with RMSE of 0.032 $\mu$m (or 18.7% in relative terms) for the situations when $\tau(440) > 0.2$ that occurs for more than eighty thousand pairs of the study. For the cases where fine mode is dominant (i.e. $\alpha$>1.2), the RMSE is only of 0.023 $\mu$m (or 13.9% in relative terms). Major limitations in the retrieval were found for the characterization of the coarse mode details. For example, the analysis revealed that GRASP-AOD retrieval is not sensitive to the small variations of the coarse mode volume median radius for different aerosol types observed at different locations. Nonetheless GRASP-AOD retrieval provides reasonable agreement with AERONET aerosol retrieval

algorithm for overall properties of coarse mode with with RMSE=0.500 $\mu$m (RMSRE=20%) when $\tau(440) > 0.2$. The values of effective radius and total volume concentration computed from GRASP-AOD retrieval have been compared to those estimated by AERONET aerosol retrieval algorithm. The RMSE values of the correlations were of 30% for the effective radius and 25% for the total volume concentration when $\tau(440) > 0.2$. Finally, the study discusses the importance of employing the assumption of bimodal log-normal size distribution. It also evaluates the potential of using ancillary data, in particular aureole

measurements, for improving the characterization of the aerosol coarse mode properties.

## 1 Introduction

Information regarding aerosol properties has an important role in several atmospheric activities such as weather prediction, air quality analyses, solar energy, aviation safety and climate studies (see last IPCC reports, Solomon et al., 2007; Stocker et al., 2014). Given its impact, both real-time and near real-time global aerosol forecasting products are distributed by several opera-

tional centers (to cite some: ECMWF Copernicus Atmosphere Monitoring Service (CAMS), Finnish Meteorological Institute (FMI), NOAA National Centers for Environmental Prediction (NCEP), Météo France or the Barcelona Supercomputing Center (BSC)). These products are generated by sophisticated numerical models that use aerosol-related observations (from satellite or ground-based) for data assimilation and model evaluation. However, the size distribution of the aerosol particles, which is one of the key parameter of aerosol properties, is not provided by most of these operational models or it presents difficulties in

its prediction in their current version (Benedetti et al., 2018). Nevertheless, the purpose of the aerosol prediction community is to offer a more complete description of the aerosol population, outputting both mass and number density concentration, in the next generation of aerosol model forecasts.

The main difficulty for a global characterization of the aerosol size distribution can be found in the lack of quality information, with enough temporal and spacial resolution, coming from real observations. In the case of satellite measurements,

apart from the typical time coverage limitation, we find that the retrieval of the size distribution is just an intermediate-step for most of the traditional satellite operational algorithms (which are based on the look-up-tables). The quality of the derived size distributions is rarely analyzed; most main attention is paid to the outcome optical thickness and other aerosol optical properties (Dubovik et al., 2011). The new sophisticated multi-angular, multi-wavelength and polarimetric sensors and the progress in the performance of computer systems that will allow the operational use of new-generation retrieval algorithms

(based on statistically optimized search in a continuous space of solutions instead of look-up-tables) are expected to improve the reliability of the aerosol retrievals by giving a more detailed representation of the aerosol properties (Dubovik et al., 2019). Therefore, new generation of satellites will provide quality long data series of aerosol properties, including a better description about the aerosol size information, that will be used as the main tool for global aerosol monitoring and characterization.

Aerosol prediction models typically use ground-based radiometer measurements to complete the information coming from

satellite sensors (Randles et al., 2017; Rubin et al., 2017). It is rare to see examples of aerosol model (even in regional models) where the input data is exclusively coming from ground based measurements. The reason is based on the fact that ground-based systems do not represent, by themselves, the spatial variation in aerosol properties (Holben et al., 2018). However, the ground

based measurements are an essential tool for satellite and aerosol model validation purposes (to cite some: Chu et al., 2002; Liu et al., 2004; Remer et al., 2002, 2005; Kahn et al., 2005; Bréon et al., 2011; Sayer et al., 2013; Levy et al., 2013; Chen et al., 2018, 2020) since: a) the spectral aerosol optical extinction is obtained from direct observations which confers a high accuracy to the value; b) the aerosol properties are better described and characterized compared to satellite retrievals, given the larger information contained in their measurements (the fore-mentioned aerosol extinction plus aerosol scattering information in larger angular ranges).

The latter statement accounts especially for the representation of the size distribution. Several ground based operational retrievals use binned distribution (where the values of the particle concentration are defined for several radii) instead of using the superposition of log-normal functions typically preferred in satellite retrievals (Nakajima et al., 1996; Dubovik and King, 2000). For instance, the AERONET (AErosol RObotic NETwork Holben et al. (1998)) aerosol retrieval algorithm (Dubovik and King, 2000; Dubovik et al., 2000, 2002b, 2006; Sinyuk et al., 2020) uses 22 bins logarithmically equidistant (from $0.05\ \mu$m to $15\ \mu$m) to characterize the aerosol size distribution from aerosol optical depth measurements and cloud-free sky radiances. With such level of detail, the binned size distribution can represent nearly any possible shape of size distribution, and even very minor features in the size distribution shape have been successfully described by the AERONET aerosol retrieval algorithm. This ability allows to describe with great precision various aerosol related phenomena: coagulation, hygroscopy, aging, cloud processing, description of particular events such as volcanic plumes or dust storms etc. (Dubovik et al., 2002a; Eck et al., 2005, 2010). Nevertheless, the needs for global validation proposes (either satellite or aerosol model products) are typically restricted to a more basic description of the microphysical parameters (effective radius and/or total volume concentration). However, they demand for a better time resolution information.

Recent studies in the field of aerosol properties retrieval are conducted to satisfy this demand. The basis consists on reducing the high requirements of current ground-based operational retrievals (almost cloudless conditions and large solar zenith angles to assure full aerosol scattering information) to provide information about aerosol microphysical properties. One of the most recurrent attempt has been the analysis of using only spectral aerosol optical depth ($\tau$) for characterizing aerosol properties (to cite some: King et al., 1978; O'Neill et al., 2003; Schuster et al., 2006; Kazadzis et al., 2014; Pérez-Ramírez et al., 2015; Torres et al., 2017). These studies are encouraged by the relative high frequency of aerosol optical depth compare to the occurrence of full set of measurements (including radiances). For instance in AERONET, the number of valid clear-sky radiance retrievals can reach about sixteen per day (new instruments with Hybrid scans) while the number of $\tau$ measurements up to two hundred per day. Moreover, many AERONET sites are plagued by several months of partial cloudiness. In these situations, there are no angular measurements of sky-radiance suitable for the retrieval of detailed aerosol properties, and only a few direct Sun measurements are available at best. In addition, there are some other networks that only provide measurements of aerosol optical depth that could potentially make use of such techniques (Maritime Aerosol Network (Smirnov et al., 2009) or the the Global Atmospheric Watch GAW-PFR (Wehrli, 2005)). Another motivation to analyze the potential of using aerosol optical depth only is the development of night measurements (star-photometry Herber et al. (2002); Pérez-Ramírez et al. (2008, 2011); Baibakov et al. (2015), and lunar photometry Barreto et al. (2013, 2016)) where $\tau$ data are typically the only information

available. In polar regions, these night spectral aerosol optical depth are the main information that can be used to infer aerosol properties during winter months.

The studies analysing the spectral aerosol optical depth can be divided according to the information derived in the retrieval. Thus, the applications based on the linear estimation techniques (LET, for more information see Veselovskii et al., 2012; Kazadzis et al., 2014; Pérez-Ramírez et al., 2015) give a simplified description of the volume aerosol size distribution approximated by the effective radius and the total volume aerosol concentration. On the other hand, the Spectral Deconvolution Algorithm (SDA, O'Neill et al., 2003), which is part of the AERONET processing chain, successfully discriminates fine and coarse mode extinction at 500 nm assuming a bimodal particle size distribution, though it does not infer information related to the microphysical properties of the assumed bimodal column volume size distribution. Finally, the GRASP-AOD application (Torres et al., 2017) also assumes the volume size distribution as bimodal log-normal and retrieves as primary output the six parameters characterizing the function: volume median radii ($R_{Vi}[\mu m]$), geometric standard deviations ($\sigma_{Vi}$) and particle volume concentration ($C_{Vi}[\mu m^3/\mu m^2]$), with $i = f, c$ for the fine and coarse mode, respectively. Once this characterization is achieved, a set of secondary aerosol properties are estimated straightway including the total effective radius($R_{eff}[\mu m]$)), the total volume concentration ($C_{V_T}[\mu m^3/\mu m^2]$) and the discrimination between the fine and coarse mode aerosol optical depth at 500 nm ($\tau_f, \tau_c$). This strategy allows GRASP-AOD application to offer a more complete description, in terms of aerosol properties, compare to the other approaches.

The GRASP-AOD application has been identified as a promising advance to derive aerosol properties with enough frequency for model validation/forecasting, specifically to infer interesting information related to aerosol speciation (Benedetti et al., 2018). However, the application of the code has been restricted to specific studies so far (Boichu et al., 2016; Román et al., 2017; Popovici et al., 2018). For instance, Boichu et al. (2016) analyzed the volcanic plumes reaching France several times in September 2014, which were emitted by the Holuhraun eruption (a massive eruption in terms of sulfur emissions which has caused repeated episodes of air pollution on a continental scale) of the Icelandic volcano Bárðarbunga. Regarding size distribution properties, the description of these plumes was quite fragmented due to the persistent cloudy conditions that allowed only a few full AERONET aerosol retrieval algorithm in the whole month. The analyses of only direct Sun measurements with the GRASP-AOD application has been revealed quite relevant to complete the aerosol size information data set during the study.

Beyond the aforementioned specific studies, the purpose of the present work is to show that the GRASP-AOD application has the potential to be used for large scale datasets either for climate studies or for near real time modeler needs. In the study by Torres et al. (2017), GRASP-AOD application was deeply described and positioned within the development of the GRASP (Generalized Retrieval of Atmosphere and Surface Properties, see Dubovik et al., 2014) algorithm and software (more information and a free version of the code can be obtained in http://www.grasp-open.com/) in its entirety. To prove the robustness in the retrieval, the work by Torres et al. (2017) presented a validation of the application though it is mainly focused on simulated tests (multiple variation of the initial guess, effects of errors in the aerosol optical depth measurements and sensitivity to the refractive index assumptions). A first real data validation was also provided by comparing the aerosol properties obtained from 744 AERONET observations using GRASP-AOD application to those obtained through 165 AERONET aerosol retrieval

algorithm from almucantar measurements at 8 different AERONET sites from several pre-selected days. However, this first validation, based on daily averages, was considered not to be sufficient if GRASP-AOD application aims to be operationally applied. Here we present a larger real data validation based on 2.8 million GRASP-AOD retrievals using AERONET aerosol optical depth observations from 30 AERONET sites for 20 years (1997-2016).

This paper is organized as follows: Section 2 explains the methodology followed to do the validation (data selection and comparisons). Section 3 describes the results of the global validation. In Section 4, we discuss some of the main results obtained here, and we point out new retrieval ideas and possible improvements for future reprocessings. Finally, in Section 5 the main conclusions are presented.

## 2 Methodology and data

As commented in the introduction, the objective of this work is to offer a large-scale validation for GRASP-AOD application by using AERONET $\tau$ measurements and operational retrievals. Thus, the aerosol properties obtained by GRASP-AOD application (with only input AERONET aerosol optical depth measurements from 340-1020 nm) will be compared to those provided by AERONET retrievals, which come from: Spectral Deconvolution Algorithm (SDA, O'Neill et al., 2003, only input aerosol optical depth from 380-870 nm) and the AERONET aerosol retrieval algorithm (Dubovik and King, 2000; Dubovik et al., 2000, 2002b, 2006; Sinyuk et al., 2020, which uses both sky-radiances and sun-direct measurements from 440-1020 nm).

### 2.1 Data source

The only data source used in the analysis belongs to AERONET Level 2.0 from the recent Version 3 (Giles et al., 2019; Sinyuk et al., 2020), which can be found in the public AERONET database (http://aeronet.gsfc.nasa.gov):

1. The $\tau$ measurements used as inputs in the GRASP-AOD application, which have passed a cloud screening criteria to obtain, first, the AERONET Level 1.5 and automated quality control algorithms to achieve the Level 2.0 (Giles et al., 2019). The accuracy of the Level 2 spectral $\tau$ measurements is $\sim 0.01$ in the visible and NIR wavelenghts and $\sim 0.02$ in the UV (Eck et al., 1999).

2. The AERONET retrievals used for the comparisons: (a) $\tau_f(500)$ obtained from the SDA (O'Neill et al., 2003) (b) size distributions standard parameters and $\tau_f(500)$ computed from the AERONET aerosol retrieval algorithm (general description in Dubovik and King (2000); Dubovik et al. (2006) with some updates for Version 3 described in Sinyuk et al. (2020)).

As mentioned in the introduction, we have carried out the comparison with the data acquired at thirty AERONET sites for twenty years (1997-2016). The thirty AERONET sites have been selected based on the availability of an extensive data record (i.e., at least ten years of $\tau$ measurements in AERONET website) and accounting for the geographic distribution among the different aerosol source regions (see Figure 1). We have chosen to limit from 340 to 1020 nm the spectral range of GRASP-AOD data input, which is common to all the photometer in AERONET network (the 1640 nm was exclusive to extended

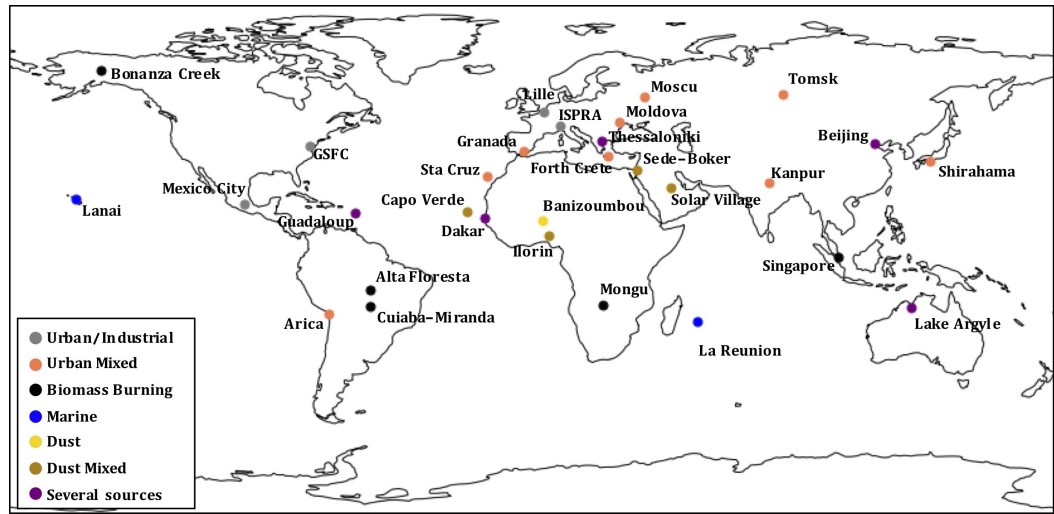

**Figure 1.** Geographical distribution of the AERONET sites considered in the analyses. The color of each site locator is assigned according to the aerosol type of the site (see sixth column in Table 1).

wavelength versions). The use of the 1640 nm channel did not suppose any substantial change in the study by Torres et al. (2017), and we have prioritized the use of a homogenized spectral range in this analysis, regardless of the photometer type[1].

Table 1 shows the information regarding the selected sites. The five first columns contain the name of the site, the period with measurements, the total number of $\tau$ measurements, the average aerosol optical depth values at 440 nm and the Ångström exponent (Ångström, 1961) registered between 1997-2016. The sixth column presents the dominant aerosol type. Note here that the purpose of the study is to carry out a validation of GRASP-AOD application and not to re-do aerosol climatologies for the different sites. In these regards, the average of $\tau(440)$, the Ångström exponent and the aerosol type labels are given here

just to briefly describe the site characteristics. The aerosol type labels are similar to those used in AERONET climatologies (Dubovik et al., 2002a; Giles et al., 2012), and the classification is based on the existing literature, which is indicated in the seventh column. Four categories represent the sites with a dominant aerosol type (despite episodic aerosol incursions outside of their classification category may have occurred at these sites during the analysis period): Biomass Burning, Urban/Industrial, Dust and Marine. We have also considered three mix aerosol categories to represent those sites presenting recurrently more

than one aerosol type: a) Urban mixed (Urban/Industrial predominance with some dust or biomass burning events along the year), b) Dust mixed (Desert dust predominance in urban backgrounds or with biomass burning episodes) and, finally, the category c) Several sources for the sites with presence of at least three different types of aerosols (ex. Beijing site in an urban

---

[1]Although new standard instrument of AERONET network, the Cimel-318T sun-photometer, includes the 1640 nm channel, the traditional standard version (which covers most of the period from 1997-2016) included only seven wavelengths in the spectral range from 340 to 1020 nm (340, 380, 440, 500, 670, 870 and 1020 nm).

background with seasonal episodes of desert dust and biomass burning). In Figure 1, the color of each site locator is assigned according to the aerosol type of the site.

The relative high average values of $\tau(440)$ in the third column are due to the fact that we have prioritized the selection of sites with significant aerosol load along the year in the study. The only exceptions are the two sites that we have categorized as influenced by marine aerosols: Reunion - St. Dennis and Lanai. The relative low average values of $\tau(440)$ for these two sites are in agreement with the studies by Smirnov et al. (2002a, 2009), which indicate that the values of $\tau(500)$ are typically less than 0.1 for pure maritime environments. As shown in Torres et al. (2017), the retrieval quality of some aerosol products

derived by GRASP-AOD do not depend on the aerosol load. This fact and the interest of including all aerosol species in the study justify the presence of these two sites in the analysis.

## 2.2    GRASP-AOD inversion

As commented in the introduction, GRASP-AOD application retrieves aerosol size properties from only $\tau$ measurements (Torres et al., 2017). The lack of scattering information, containing essential information to derive a detailed characterization

of aerosols, obliges to do a series of approximations and simplifications in order to adjust the aerosol model used in the retrieval to the actual information content. The GRASP algorithm has a highly flexible forward model that makes this possible. In these regards, the retrieve size distributions are approximated as bimodal log-normals which are described by only 6 parameters: volume median radius ($r_{V_i}$), standard deviation ($\sigma_{V_i}$) and volume concentration ($C_{V_i}$) for fine and coarse mode (instead of more detailed binned size distributions as in the case of AERONET standard inversion). The application assumes the complex

refractive index as known in the retrieval procedure. Full inversion details of GRASP-AOD and the consequences of the different assumptions can be gained in Torres et al. (2017) and references therein.

The purpose of this subsection is to describe the particular use made of GRASP-AOD in this validation study. As commented before, the primary input are the almost 2.8 millions $\tau$ measurements belonging to AERONET Level 2.0 of Version 3 between 1997-2016 in the 30 sites selected for the analysis (see Table 1). Regarding the assumption of the refractive index, we have

created a database of moving monthly means (2 adjacent months) for all sites using Version 3 AERONET aerosol retrieval algorithm (considering the entire historical database, beyond the period analyzed here). We have prioritized the use of Level 2.0 refractive index data (quality assured data, Holben et al., 2006). When there was not enough data for this calculation, we have increased the moving average to 5 or 7 months. When even the use of 7 months average was not enough to produce a climatological value, the Level 1.5 has been used. This happened for Lanai which does not have refractive index data in

Level 2.0 during the whole archive.

Given the speed of GRASP-AOD application (just a few seconds for an entire day), we have adopted a multiple choice strategy for the initial guess values. Thus, GRASP-AOD has been run with different initial guess and among the obtained results we have selected the one with the smallest fitting error. The different combinations for fine and coarse mode volume median radius initial guesses can be found in Table 2. They are inspired by the experience and the initial guess analysis proposed

in Section 3.3 of Torres et al. (2017). Depending on the Ångström exponent value, we give more options to the dominant mode

**Table 1.** General description of the data used in the validation study. First two columns present the name of the site, the period with $\tau$ measurements. Third column shows the total number of $\tau$ measurements. The two following columns contain the average values of $\tau(440)$ and the Ångström exponent. The last two columns illustrate the aerosol type and the main references analyzing the site characteristics.

| Site | Period | N° of $\tau$ meas. | $<\tau(440)>$ | $<\alpha>$ | Aerosol type | References |
|---|---|---|---|---|---|---|
| Alta Floresta | 28/01/1999 - 31/12/2016 | 71897 | 0.457 | 1.34 | Biomass Burning | Dubovik et al. (2002a); Eck et al. (2003) |
| Arica | 13/05/1998 - 23/12/2016 | 87116 | 0.262 | 1.055 | Urban mixed | Eck et al. (2012); Carn et al. (2007) |
| Banizoumbou | 01/01/1997 - 31/12/2016 | 152324 | 0.463 | 0.358 | Dust | Holben et al. (2001) |
| Beijing | 07/03/2001 - 31/12/2016 | 94768 | 0.771 | 1.132 | Several sources | Eck et al. (2005) |
| Bonanza Creek | 15/07/1997 - 01/11/2016 | 38016 | 0.267 | 1.356 | Biomass Burning | Eck et al. (2009) |
| Cuiaba Miranda | 22/03/2001 - 31/12/2016 | 57961 | 0.371 | 1.35 | Biomass Burning | Holben et al. (2001); Dubovik et al. (2002a) |
| Capo Verde | 01/01/1997 - 30/12/2016 | 85430 | 0.341 | 0.27 | Dust mixed | Holben et al. (2001); Tanré et al. (2001) |
| Dakar | 01/01/1997 - 31/12/2016 | 132155 | 0.442 | 0.35 | Several sources | Holben et al. (2001); Mortier et al. (2016) |
| Forth Crete | 04/01/2003 - 28/12/2016 | 80599 | 0.216 | 1.124 | Urban mixed | Bergamo et al. (2008) |
| GSFC | 04/01/1997 - 30/12/2016 | 143418 | 0.216 | 1.612 | Urban/Industrial | Dubovik et al. (2002a) |
| Granada | 29/12/2004 - 30/12/2016 | 115618 | 0.169 | 1.069 | Urban mixed | Lyamani et al. (2005, 2010) |
| Guadeloup | 19/02/1997 - 30/12/2016 | 41179 | 0.151 | 0.351 | Several sources | Prospero et al. (2014) |
| Ilorin | 25/04/1998 - 31/12/2016 | 76267 | 0.8 | 0.631 | Dust mixed | Eck et al. (2010) |
| Ispra | 28/06/1997 - 31/12/2016 | 112914 | 0.287 | 1.521 | Urban/Industrial | Mélin and Zibordi (2005) |
| Kanpur | 22/01/2001 - 31/12/2016 | 115651 | 0.719 | 0.98 | Urban mixed | Eck et al. (2012) |
| Lake Argyle | 28/10/2001 - 28/12/2016 | 132642 | 0.143 | 1.107 | Several Sources | Mitchell et al. (2013) |
| Lanai | 01/07/1997 - 04/02/2004 | 45850 | 0.078 | 0.445 | Marine | Dubovik et al. (2002a) |
| Lille | 01/01/1997 - 30/12/2016 | 57111 | 0.23 | 1.3 | Urban/Industrial | Mortier (2013) |
| Mexico City | 22/02/1999 - 05/12/2016 | 62132 | 0.386 | 1.562 | Urban/Industrial | Dubovik et al. (2002a) |
| Moldova | 03/09/1999 - 31/12/2016 | 92231 | 0.244 | 1.484 | Urban mixed | Kabashnikov et al. (2014) |
| Mongu | 02/01/1997 - 16/01/2010 | 103773 | 0.333 | 1.677 | Biomass Burning | Holben et al. (2001); Dubovik et al. (2002a) |
| Moscow | 28/08/2001 - 20/12/2016 | 60276 | 0.262 | 1.43 | Urban mixed | Chubarova et al. (2011a, b) |
| Reunion - St. Denis | 15/06/1997 - 28/12/2016 | 58768 | 0.074 | 0.65 | Marine | Mallet et al. (2018) |
| Sede Boker | 04/11/1997 - 31/12/2016 | 225289 | 0.199 | 0.931 | Dust mixed | Derimian et al. (2006) |
| St. Cruz de Tenerife | 22/07/2004 - 31/12/2016 | 117258 | 0.186 | 0.581 | Urban mixed | Gonzalez Ramos and Rodriguez (2013) |
| Shirahama | 19/10/2000 - 31/12/2016 | 99167 | 0.292 | 1.242 | Urban mixed | Eck et al. (2005) |
| Singapore | 14/11/2006 - 30/12/2016 | 35604 | 0.645 | 1.382 | Biomass Burning | Chew et al. (2013) |
| Solar Village | 22/02/1999 - 13/08/2015 | 182223 | 0.354 | 0.535 | Dust mixed | Dubovik et al. (2002a) |
| Thessaloniki | 01/06/2003 - 29/12/2016 | 86929 | 0.281 | 1.582 | Several sources | Giannakaki et al. (2010) |
| Tomsk | 24/10/2002 - 29/12/2016 | 27544 | 0.208 | 1.413 | Urban mixed | Panchenko et al. (2012) |
| Total | 01/01/1997 - 31/12/2016 | 2792110 | 0.329 | 1.017 | | |

**Table 2.** Multiple initial guess values for the volume mode radii used in the GRASP-AOD application. The choices depend on the Ångström exponent which characterizes the dominant mode.

| $\alpha$ | $r_{V_f}$ [$\mu$m] | $\sigma_{V_f}$ | $r_{V_c}$ [$\mu$m] | $\sigma_{V_c}$ | N° of inversion for retrieval |
|---|---|---|---|---|---|
| >1.2 | 0.15, 0.20, 0.25, 0.30 | 0.4 | $2.8 + 0.3\tau(440)$ | 0.7 | 4 |
| 0.9 – 1.2 | 0.14, 0.20, 0.26 | 0.4 | $2.3, 2.6 + 0.3\tau(440)$ | 0.6 | 6 |
| 0.6 – 0.9 | 0.13, 0.18, 0.23 | 0.4 | $2.15, 2.4 + 0.3\tau(440)$ | 0.6 | 6 |
| <0.6 | 0.12 | 0.4 | 1.9, 2.3 | 0.6 | 2 |

since we expect to have larger sensitivity to characterize its volume radius. The interval for the retrieved radii is independent to the Ångström exponent value and goes from 0.07 to 0.7 $\mu$m for the fine mode, and from 0.7 to 5 $\mu$m for the coarse mode.

Regarding the concentrations and standard deviations of both modes, only one choice has been used and the values are the same as in Table 4 of Torres et al. (2017). Due to the low sensitivity of GRASP-AOD to the shape of the modes (more details in Torres et al. (2017)), we have used strong a priori constraints on the actual values for the standard deviation of both modes (see Eq.1 in Torres et al. (2017)). Although the standard deviations are still retrieved by GRASP-AOD, in practice, their values are very similar to the given initial guess values. That is the reason why their retrieval will not be discussed in the comparison analysis with AERONET retrievals in Section 3.

Second column in Table 3 contains the total number of GRASP-AOD inversions for each site. We have added the data percentage with respect to the total $\tau$ measurements presented in Table 1. There is a high percentage with valid GRASP-AOD retrievals: 95% of the total $\tau$ measurements. The non one-to-one correspondence is due to the criteria defined to filter the GRASP-AOD retrievals. These criteria are based mostly on analyst's experience and are as follows:

1. At least four valid spectral $\tau$ measurements, i.e. four different wavelengths with $\tau$ measurements in Level 2.0 in the spectral range 340-1020 nm.

2. The set of $\tau$ measurements should contain at least one between 440 nm and 500 nm, and another between 870 nm and 1020 nm.

3. Value of $\tau(440)$ (or interpolated if does not exist) over 0.02.

4. Absolute total fitting under 0.015 for measurements with $\tau(440) < 0.5$, and under $\tau(440) \times 0.016 + 0.007$ for measurements with $\tau(440) > 0.5$. The fitting is obtained by the difference between the spectral $\tau$ measurements and the computed spectral $\tau$ values (which are estimated from the retrieved size distribution and the assumed refractive indices).

5. Absolute fitting of $\tau(500)$ (or interpolated if does not exist) under $0.01 + 0.005 \times \tau(500)$.

If we analyse now the data percentage by sites, all sites except Ilorin present more than 85% of GRASP-AOD valid retrievals with respect to the total number of $\tau$ measurements. In general, higher percentages are observed for sites with fine mode

**Table 3.** Total number of GRASP-AOD inversions, SDA retrievals and the AERONET aerosol retrievals for each site. The percentage with respect to the total number of $\tau$ measurements is indicated in parentheses.

| Site | N° of GRASP-AOD inversions | N° of SDA retrievals | N° of AERONET aerosol retrievals |
|---|---|---|---|
| Alta Floresta | 67251 (94%) | 62487 (87%) | 2795 (4%) |
| Arica | 85713 (98%) | 81929 (94%) | 4909 (6%) |
| Banizoumbou | 130543 (86%) | 0 (0%) | 8266 (5%) |
| Beijing | 84329 (89%) | 10068 (11%) | 4227 (4%) |
| Bonanza Creek | 36613 (96%) | 34462 (91%) | 937 (2%) |
| Capo Verde | 77221 (90%) | 5850 (7%) | 4519 (5%) |
| Cuiaba Miranda | 56066 (97%) | 50699 (87%) | 2127 (4%) |
| Dakar | 119676 (91%) | 55405 (42%) | 7278 (6%) |
| Forth Crete | 79645 (99%) | 77334 (96%) | 3982 (5%) |
| Granada | 115020 (99%) | 98422 (85%) | 7331 (6%) |
| Guadeloup | 38612 (94%) | 22116 (54%) | 958 (2%) |
| GSFC | 142839 (100%) | 139336 (97%) | 10840 (8%) |
| Ilorin | 57834 (76%) | 72340 (95%) | 3608 (5%) |
| Ispra | 111317 (99%) | 80218 (71%) | 4011 (4%) |
| Kanpur | 110699 (96%) | 99573 (86%) | 9548 (8%) |
| Lake Argyle | 125694 (95%) | 121098 (91%) | 7490 (6%) |
| Lanai | 44467 (97%) | 39763 (87%) | 1183 (3%) |
| Lille | 56605 (97%) | 38702 (68%) | 2971 (5%) |
| Mexico City | 60624 (98%) | 58284 (94%) | 2283 (4%) |
| Moldova | 92054 (100%) | 90280 (98%) | 5768 (6%) |
| Mongu | 90005 (87%) | 85869 (83%) | 4818 (5%) |
| Moscow | 59669 (99%) | 56839 (94%) | 2242 (4%) |
| Reunion - St. Denis | 57059 (97%) | 47977 (82%) | 2773 (5%) |
| Sede Boker | 222141 (99%) | 206055 (91%) | 15768 (7%) |
| St. Cruz Tenerife | 113376 (97%) | 111610 (95%) | 6413 (5%) |
| Shirahama | 96464 (97%) | 90266 (91%) | 4458 (4%) |
| Singapore | 33246 (93%) | 32922 (92%) | 559 (2%) |
| Solar Village | 172999 (95%) | 162326 (89%) | 14285 (8%) |
| Thessaloniki | 86772 (100%) | 63373 (73%) | 6097 (7%) |
| Tomsk | 27472 (100%) | 23850 (87%) | 747 (3%) |
| Total | 2651025 (95%) | 2119453 (76%) | 153191 (5%) |

predominance (between 94%-100%, with some exceptions) than for sites with coarse mode predominance (between 85%-95%). The relatively small number of valid GRASP-AOD retrievals in Ilorin site (76%) is related to the assumption of a bimodality in the size distribution. This issue will be deeply analyzed in section 4.1.1

## 2.3 AERONET retrievals

### 2.3.1 SDA algorithm

O'Neill et al. (2003) developed the Spectral Deconvolution Algorithm (SDA) to discriminate fine and coarse mode extinction at 500 nm ($\tau_f(500)$ and $\tau_c(500)$) with the only input of $\tau$ measurements between 380-870 nm. The algorithm is part of the AERONET processing chain: the value of the fine and coarse mode $\tau$ at 500 nm is retrieved from every measured $\tau$ spectrum and provided as a standard product of the network (full description in http://aeronet.gsfc.nasa.gov/new_web/PDF/tauf_tauc_technical_memo1.pdf). However and as previously indicated, only SDA Level 2.0 retrievals have been considered in the study. Note here that neither GRASP-AOD nor AERONET aerosol retrieval algorithm provide $\tau_f(500)$ and $\tau_c(500)$ as primary outputs. In both cases, the discrimination between fine and coarse mode extinction is estimated from their main outputs (more details can be gained in the following subsection).

Third column in Table 3 shows the number of SDA retrievals in Level 2.0. The percentage with respect to the total $\tau$ measurements in Level 2.0 is 76% for the 30 sites considered for the period 1997-2016. The non one-to-one correspondence is due to the criteria to reach SDA Level 2.0. Details on SDA Level 2.0 criteria can be gained in AERONET website (https://aeronet.gsfc.nasa.gov/new_web/data_description_AOD_V2.html). They are a little stricter than those of GRASP-AOD, which may justify the significantly lower number of retrievals with respect to GRASP-AOD application. Certainly, the most critical is that the spectral range must be bounded by 380 and 870 nm with at least two additional wavelengths between the bounds. Five of the 30 selected sites (Dakar, Capo Verde, Banizoumbo, Guadaloupe and Beijing) have installed polarized photometers for most of the years of this analysis. The polarized photometers only have four spectral channels from 440 to 1020 nm, and therefore, they do not provide $\tau$ measurements at 380 nm. This implies, for instance, that there are no SDA Level 2.0 data at Banizoumbou in the whole period, and that the SDA Level 2.0 data represents only 11% and 7% at sites of Capo Verde and Beijing, respectively.

### 2.3.2 AERONET aerosol retrieval algorithm

AERONET aerosol retrieval algorithm uses $\tau$ measurements combined with spectral sky radiances to obtain detailed aerosol volume size distribution (22 bins logarithmically equidistant between 0.05 and 15 $\mu$m), complex refractive index and the sphericity parameter as main outputs (Dubovik and King, 2000; Dubovik et al., 2006). Other aerosol properties such as the single scattering albedo (SSA), aerosol absorption or the asymmetry factor are estimated afterwards from the primary outputs.

In addition, the detailed 22-bin size distributions are approximated as bimodal log-normal distributions to derive their equivalent parameters: volume median radii, standard deviations and particle volume concentrations for fine and coarse mode (details and exact formulation can be gained in Dubovik et al. (2002a) and https://aeronet.gsfc.nasa.gov/new_web/Documents/

Inversion_products_for_V3.pdf). To perform these calculations, the contribution of fine and coarse mode in each 22-bin size distribution should be known beforehand. From AERONET Version 2, an automatic process finds the minimum of the volume size distribution within the size interval from 0.439 to 0.992 $\mu$m; this minimum is used to settle the separation point. This process has been kept in the current AERONET Version 3 (our data source). These so-called standard parameters of the volume size distributions can be directly compared with the GRASP-AOD retrievals since they are their primary outputs of this inversion. Furthermore, AERONET aerosol retrieval algorithm estimates the effective radius $R_{eff}$ and the total volume concentration $C_{V_T}$ for each mode as well as for the entire size distribution. Both parameters have been also computed for all GRASP-AOD retrievals.

The separation between fine/coarse mode in the detailed size distribution is used as well to estimate the optical thickness for fine and coarse mode at 440, 675, 870 and 1020 nm, from the AERONET aerosol retrieval algorithm outputs. The particular values at 500 nm, $\tau_f(500)$, have been interpolated for our validation study. Note that the way that the two modes are separated by the AERONET aerosol retrieval algorithm represents itself an inherent source of difference with other methods to estimate fine/coarse mode optical thickness. In fact, the distribution of fine and coarse particles are continuous entities which overlap between them and they spread beyond the border established by the separation point or cutoff. As explained by O'Neill et al. (2003), a simple analysis of Mie kernels would show that the optical depth due to coarse particles for radii smaller than the cutoff (wrongly included in $\tau_f$ calculations) is larger than the optical depth due to fine particles for radii larger than the cutoff (wrongly excluded from $\tau_f$ calculations). Therefore, the fine mode optical depth is generally overestimated while the coarse mode optical depth is generally underestimated. This effect is typically small, and it is more significant if the coarse mode dominates. Neither SDA nor GRASP-AOD application present this issue since the two modes can overlap in both algorithms. In the case of GRASP-AOD, the primary outputs are two independent log-normal functions which represent separately the fine and coarse mode as aforementioned. The values of $\tau_f(500)$ and $\tau_c(500)$ are derived from the aerosol optical depth values calculated individually for each log-normal function.

Last column in Table 3 contains the total number of AERONET aerosol retrievals for each site during the period 1997-2016 in Level 2.0. These numbers are much smaller than the number of $\tau$ measurements for several reasons. First, the AERONET standardised sequence of measurements includes around forty direct sun measurements per day (this number can vary depending on the site latitude and the type of instrument) but only about eight of these sequences are coincident with sky-radiance almucantar measurements[2] (suitable as input to the AERONET aerosol retrieval algorithm). Secondly, the AERONET aerosol retrieval algorithm requires that most sky radiances to be cloud-free and homogeneous in addition to the Sun being unobscured. Finally, the Level 2.0 criteria for size distribution parameters requires solar zenith angle greater than $50°$ and $\tau(440) > 0.02$ to assure the robustness of the retrievals[3].

---

[2]Recently, these numbers have been increased with the incorporation of new hybrid sky radiance measurements which are performed only by the newest instruments (Sinyuk et al., 2020). Nevertheless, we have limited our validation study to almucantar retrievals, since the results of the hybrid scans were still under validation at the time of this study, and its use is still relatively small in the AERONET network

[3]The use of the hybrid scans would allow to reduce the requirement in the solar zenith angles to only $25°$, since the scattering angular range provided by these measurements is the same as the one that almucantar provides at $50°$ (Sinyuk et al., 2020).

## 2.4 Match-up methodology

The data set in which GRASP-AOD and SDA can be applied is the same: every single $\tau$ measurement. Therefore, we can compare the results one by one between the two methods when both retrievals pass the criteria previously described. Comparisons of the results obtained for $\tau_f(500)$ (which will be the subject of the first analysis in Section 3) between GRASP-AOD and SDA correspond to same $\tau$ measurement as input.

Values of $\tau_f(500)$ computed using the retrieved parameters from the AERONET aerosol algorithm will be also compared with the results obtained by GRASP-AOD and SDA retrievals. However, the primary data set of AERONET aerosol retrieval algorithm is restricted to scenarios including sky radiance passing the aforementioned criteria. To homogenize the different data sets (spectral aerosol optical depth measurements with or without almucantar), we have performed averages of the $\tau_f(500)$ results obtained by GRASP-AOD and SDA within $\pm16$ min of each almucantar measurement. Note here that we have chosen that interval since it is the one used by AERONET aerosol retrieval algorithm to average the $\tau$ measurements around each almucantar to be used as input in the retrieval (this and more information can be found at https://aeronet.gsfc.nasa.gov/new_web/Documents/AERONETcriteria_final1_excerpt.pdf).

Results about aerosol size parameters can be only compared between GRASP-AOD and AERONET aerosol retrieval algorithm. As for the $\tau_f(500)$ comparison, we have averaged the aerosol size parameters retrieved by GRASP-AOD in a $\pm16$ interval centered in each almucantar measurement.

## 2.5 Considered metrics for comparison statistics

To evaluate the comparisons between GRASP-AOD and AERONET retrievals we make use of standard statistical parameters, including slope and offset of linear regression, Pearson's linear correlation coefficient (R), root mean square error (RMSE), root mean square relative error (RMSRE) and bias. The last three are defined as follows:

$$R = \frac{\sum_{i=1}^{N}(a_{i_{\text{AERONET}}} - \overline{a_{\text{AERONET}}})(a_{i_{\text{GRASP-AOD}}} - \overline{a_{\text{GRASP-AOD}}})}{\sqrt{\sum_{i=1}^{N}(a_{i_{\text{AERONET}}} - \overline{a_{\text{AERONET}}})^2 \sum_{i=1}^{N}(a_{i_{\text{GRASP-AOD}}} - \overline{a_{\text{GRASP-AOD}}})^2}} \tag{1}$$

$$\text{RMSE} = \sqrt{\frac{\sum_{i=1}^{N}(a_{i_{\text{AERONET}}} - a_{i_{\text{GRASP-AOD}}})^2}{N}} \tag{2}$$

$$\text{RMRSE} = \frac{\text{RMSE}}{\overline{a_{\text{AERONET}}}} = \frac{\sqrt{\frac{\sum_{i=1}^{N}(a_{i_{\text{AERONET}}} - a_{i_{\text{GRASP-AOD}}})^2}{N}}}{\overline{a_{\text{AERONET}}}} \tag{3}$$

$$\text{BIAS} = \frac{1}{N}\sum_{i=1}^{N}(a_{i_{\text{GRASP-AOD}}} - a_{i_{\text{AERONET}}}) \tag{4}$$

where N is the number of matched data points $i$; $a_{\mathrm{AERONET}}$ represents the value retrieved of a given parameter obtained by AERONET; $a_{\mathrm{GRASP-AOD}}$ represents the same value but obtained by GRASP-AOD; $\overline{a_{\mathrm{AERONET}}}$ and $\overline{a_{\mathrm{GRASP-AOD}}}$ are the mean values for AERONET and GRASP-AOD retrievals for a given parameter.

## 3 Results

The comparison between GRASP-AOD retrievals and AERONET retrievals has been divided in two main subsections in the analysis of the results. First, we compare the values of the fine mode aerosol optical depth $\tau_f(500)$ giving by: GRASP-AOD, SDA and the AERONET aerosol retrieval algorithm. Secondly, we compare the so-called standard parameters of the volume size distributions from the aerosol size distributions obtained by GRASP-AOD and the AERONET aerosol retrieval algorithm.

### 3.1 Separation fine/coarse mode: $\tau_f(500)$

Table 4 and Table 5 contain the most relevant parameters in the comparisons of the $\tau_f(500)$ values obtained by GRASP-AOD, SDA and the AERONET aerosol retrieval algorithm. Number of coincident data (following the criteria given in subsection 2.4), values of the correlation coefficients, root-mean-square errors (RMSE) and root-mean-square relative error (RMSRE, enclosed in parentheses) are represented in Table 4 for each site. Slopes and the intercepts of the linear regressions are shown in Table 5. In both tables, we have added the analysis for all sites in the last row. We have also interspersed a Sub-Total row with the general results but excluding from the analysis the 5 sites with less than $60\%$ of SDA retrievals respect to the total number of $\tau$ measurements (as explained in subsection 2.3.1 these are the sites with long periods of polarized photometers: Banizoumbou, Beijing, Capo Verde, Dakar and Guadeloup). Figure 2 illustrates the RMSE values from Table 4: red bars for comparisons between SDA and GRASP-AOD, green bars between AERONET aerosol retrieval algorithm and GRASP-AOD, and blue bars between AERONET aerosol retrieval algorithm and SDA. The sites have been ordered on the X-axis according to RMSE values obtained in the comparisons between AERONET aerosol retrieval algorithm and GRASP-AOD (common to all sites).

If we analyze Figure 2, we do not observe large differences between the three RMSE values for the same site. The lowest RMSE are typically obtained in the comparison between SDA and GRASP-AOD (red bars in Figure 2). This fact is confirmed in the comparison for all sites (last row Total from Table 4 or in the middle of Figure 2), where the RMSE for 2 million common retrievals between SDA and GRASP-AOD is the lowest 0.015. In the same row, we observe that the $\tau_f(500)$ computed by AERONET aerosol retrieval algorithm for all sites agrees slightly better with GRASP-AOD (RMSE=0.018, from 148526 comparisons) than with SDA (RMSE=0.022, from 127203 comparisons). This better agreement is more pronounced if we exclude from the analysis the sites with less than $60\%$ of SDA retrievals (row Sub-Total in Table 4): RMSE=0.016 between AERONET aerosol retrieval algorithm and GRASP-AOD, RMSE=0.022 for the comparison between AERONET aerosol retrieval algorithm and SDA. The analysis by sites shows that the largest RMSE between the different methods are obtained at: Beijing for the comparison between AERONET aerosol retrieval algorithm and GRASP-AOD (RMSE=0.054), Ilorin for the comparison between AERONET aerosol retrieval algorithm and SDA (RMSE=0.037), and Kanpur for the comparison between SDA and GRASP-AOD (RMSE=0.066). The smallest RMSE between the three methodologies are observed at La Reunion

**Table 4.** Comparisons of $\tau_f(500)$ values computed with three different algorithms (AERONET, GRASP-AOD, and SDA) for sites and periods indicated in Table 1. First column depicts the site name and the rest of the columns indicate the number of coincident data, values of the correlation coefficients, RMSE (and RMSRE enclosed in parentheses) of the comparisons between the methods.

| Sites | SDA vs GRASP | | | AERONET Std. vs GRASP | | | AERONET Std. vs SDA | | |
|---|---|---|---|---|---|---|---|---|---|
| | N° meas. | Coeff. -R- | RMSE | N° meas. | Coeff. -R- | RMSE | N° meas. | Coeff. -R- | RMSE |
| Alta Floresta | 61164 | 1.00 | 0.011 (5.8%) | 2682 | 1.00 | 0.023 (9.3%) | 2663 | 1.00 | 0.021 (7.8%) |
| Arica | 81319 | 0.99 | 0.017 (10.3%) | 4895 | 0.99 | 0.012 (6.9%) | 4778 | 0.99 | 0.019 (10.8%) |
| Bonanza Creek | 33970 | 1.00 | 0.011 (9.4%) | 889 | 1.00 | 0.025 (9.9%) | 890 | 1.00 | 0.024 (7.9%) |
| Cuiaba Miranda | 49842 | 1.00 | 0.012 (6.0%) | 2085 | 1.00 | 0.020 (7.6%) | 2095 | 1.00 | 0.020 (7.4%) |
| Forth Crete | 76560 | 0.99 | 0.011 (10.0%) | 3958 | 0.99 | 0.011 (8.7%) | 3968 | 0.99 | 0.013 (10.1%) |
| Granada | 97823 | 0.99 | 0.009 (11.7%) | 7308 | 0.98 | 0.011 (12.3%) | 7319 | 0.99 | 0.013 (14.8%) |
| GSFC | 138980 | 1.00 | 0.007 (4.7%) | 10834 | 1.00 | 0.008 (6.1%) | 10795 | 1.00 | 0.010 (7.3%) |
| Ilorin | 54661 | 0.99 | 0.033 (12.7%) | 2594 | 0.98 | 0.044 (12.0%) | 3591 | 0.95 | 0.066 (16.5%) |
| Ispra | 79606 | 1.00 | 0.011 (6.2%) | 3971 | 1.00 | 0.012 (5.2%) | 2650 | 1.00 | 0.013 (6.8%) |
| Kanpur | 98064 | 1.00 | 0.037 (8.8%) | 9477 | 1.00 | 0.023 (4.9%) | 9324 | 1.00 | 0.042 (9.1%) |
| Lake Argyle | 118086 | 1.00 | 0.008 (9.6%) | 7317 | 1.00 | 0.007 (7.3%) | 7245 | 1.00 | 0.009 (9.4%) |
| Lanai | 39175 | 0.99 | 0.007 (20.5%) | 1170 | 0.99 | 0.005 (14.9%) | 1166 | 0.99 | 0.008 (24.1%) |
| Lille | 38554 | 1.00 | 0.009 (6.8%) | 2969 | 1.00 | 0.010 (5.9%) | 2134 | 1.00 | 0.011 (6.9%) |
| Mexico City | 57503 | 1.00 | 0.018 (6.5%) | 2281 | 1.00 | 0.016 (6.4%) | 2251 | 1.00 | 0.016 (6.7%) |
| Moldova | 90041 | 1.00 | 0.01 (6.3%) | 5764 | 1.00 | 0.008 (5.0%) | 5709 | 1.00 | 0.013 (8.1%) |
| Mongu | 84482 | 1.00 | 0.01 (4.7%) | 4807 | 1.00 | 0.016 (6.0%) | 4714 | 1.00 | 0.013 (5.2%) |
| Moscow | 56267 | 1.00 | 0.011 (6.8%) | 2231 | 1.00 | 0.011 (5.7%) | 2228 | 1.00 | 0.014 (7.4%) |
| Reunion - St. Denis | 47693 | 0.99 | 0.005 (17.1%) | 2753 | 0.99 | 0.005 (13.7%) | 2407 | 0.99 | 0.007 (18.3%) |
| Sede Boker | 203620 | 0.98 | 0.010 (11.6%) | 15671 | 0.98 | 0.013 (13.7%) | 15575 | 0.98 | 0.015 (15.0%) |
| St. Cruz Tenerife | 108324 | 0.92 | 0.014 (26.8%) | 6261 | 0.92 | 0.019 (29.2%) | 6409 | 0.98 | 0.013 (18.9%) |
| Shirahama | 88379 | 1.00 | 0.013 (7.0%) | 4449 | 1.00 | 0.012 (6.1%) | 4442 | 1.00 | 0.016 (8.3%) |
| Singapore | 31382 | 1.00 | 0.020 (6.0%) | 553 | 1.00 | 0.029 (7.0%) | 549 | 1.00 | 0.028 (6.9%) |
| Solar Village | 155939 | 0.98 | 0.015 (14.8%) | 13755 | 0.97 | 0.019 (16.2%) | 13982 | 0.97 | 0.025 (20.3%) |
| Thessaloniki | 63299 | 1.00 | 0.010 (5.7%) | 6093 | 1.00 | 0.010 (5.3%) | 4688 | 1.00 | 0.012 (6.4%) |
| Tomsk | 23741 | 1.00 | 0.009 (6.7%) | 746 | 1.00 | 0.012 (7.3%) | 681 | 1.00 | 0.016 (9.0%) |
| Sub-Total | 1978474 | 1.00 | 0.015 (9.8%) | 125513 | 1.00 | 0.016 (9.2%) | 122253 | 1.00 | 0.022 (12.6%) |
| Banizoumbou | - | - | - | 7154 | 0.96 | 0.018 (15.0%) | - | - | - |
| Beijing | 9908 | 1.00 | 0.037 (11.3%) | 4010 | 1.00 | 0.054 (9.3%) | 681 | 1.00 | 0.050 (11.0%) |
| Capo Verde | 5140 | 0.91 | 0.018 (26.8%) | 4143 | 0.96 | 0.018 (19.9%) | 350 | 0.98 | 0.015 (16.4%) |
| Dakar | 49242 | 0.94 | 0.020 (20.1%) | 6496 | 0.95 | 0.022 (16.9%) | 3273 | 0.97 | 0.024 (17.1%) |
| Guadeloup | 21710 | 0.86 | 0.014 (40.5%) | 945 | 0.91 | 0.017 (38.0%) | 651 | 0.93 | 0.013 (29.3%) |
| Total | 2064377 | 1.00 | 0.015 (10.1%) | 148261 | 1.00 | 0.018 (10.4%) | 127203 | 1.00 | 0.022 (12.8%) |

**Table 5.** Continuation of Table 4, which describes the comparisons of $\tau_f(500)$. Here, we represent the slopes and the intercepts obtained by the linear regressions between the $\tau_f(500)$ values retrieved by the three different algorithms (AERONET, GRASP-AOD, and SDA).

| Sites | SDA vs GRASP | | AERONET Std. vs GRASP | | AERONET Std. vs SDA | |
|---|---|---|---|---|---|---|
| | Slope | Intercept | Slope | Intercept | Slope | Intercept |
| Alta Floresta | 1.003 | 0.005 | 1.044 | -0.006 | 1.039 | -0.009 |
| Arica | 0.928 | 0.020 | 0.986 | -0.003 | 1.057 | -0.021 |
| Bonanza Creek | 0.986 | 0.003 | 1.005 | -0.006 | 1.027 | -0.009 |
| Cuiaba Miranda | 1 | 0.005 | 1.041 | -0.007 | 1.044 | -0.011 |
| Forth Crete | 0.95 | 0.007 | 1.009 | -0.007 | 1.046 | -0.012 |
| Granada | 0.966 | 0.007 | 0.952 | -0.002 | 0.963 | -0.007 |
| GSFC | 0.989 | 0.004 | 1.036 | -0.005 | 1.042 | -0.008 |
| Ilorin | 1.058 | 0.006 | 1.032 | -0.004 | 0.922 | 0.014 |
| Ispra | 0.98 | 0.006 | 1.012 | -0.002 | 1.039 | -0.007 |
| Kanpur | 0.913 | 0.037 | 0.994 | -0.002 | 1.088 | -0.042 |
| Lake Argyle | 1.015 | 0.004 | 1.019 | -0.003 | 1.006 | -0.005 |
| Lanai | 0.986 | 0.004 | 0.999 | -0.003 | 0.999 | -0.006 |
| Lille | 0.971 | 0.006 | 1.001 | -0.004 | 1.039 | -0.010 |
| Mexico City | 1.002 | 0.011 | 1.036 | -0.004 | 1.033 | -0.013 |
| Moldova | 0.979 | 0.007 | 1.021 | -0.005 | 1.041 | -0.011 |
| Mongu | 1.009 | 0.004 | 1.045 | -0.007 | 1.035 | -0.01 |
| Moscow | 0.996 | 0.007 | 1.028 | -0.006 | 1.037 | -0.012 |
| Reunion - St. Denis | 1.032 | 0.002 | 1.012 | -0.003 | 0.986 | -0.004 |
| Sede Boker | 0.996 | 0.003 | 0.982 | -0.005 | 0.955 | -0.006 |
| St. Cruz Tenerife | 0.834 | 0.008 | 0.768 | 0.005 | 0.957 | -0.006 |
| Shirahama | 0.981 | 0.009 | 1.015 | -0.006 | 1.027 | -0.014 |
| Singapore | 0.989 | 0.014 | 1.028 | -0.008 | 1.03 | -0.015 |
| Solar Village | 1.087 | -0.002 | 0.937 | -0.003 | 0.82 | 0.004 |
| Thessaloniki | 1.001 | 0.003 | 1.027 | -0.006 | 1.032 | -0.007 |
| Tomsk | 1.001 | 0.002 | 1.025 | -0.006 | 1.013 | -0.009 |
| Sub-Total | 0.986 | 0.007 | 1.017 | -0.006 | 1.033 | -0.013 |
| Banizoumbou | - | - | 0.961 | -0.001 | - | - |
| Beijing | 0.938 | 0.016 | 0.951 | -0.002 | 1.047 | -0.031 |
| Capo Verde | 0.814 | 0.006 | 0.949 | -0.007 | 1.029 | -0.009 |
| Dakar | 1.031 | -0.002 | 0.967 | -0.006 | 0.975 | -0.008 |
| Guadeloup | 0.656 | 0.009 | 0.666 | 0.005 | 0.993 | -0.005 |
| Total | 0.985 | 0.006 | 0.996 | -0.004 | 1.032 | -0.013 |

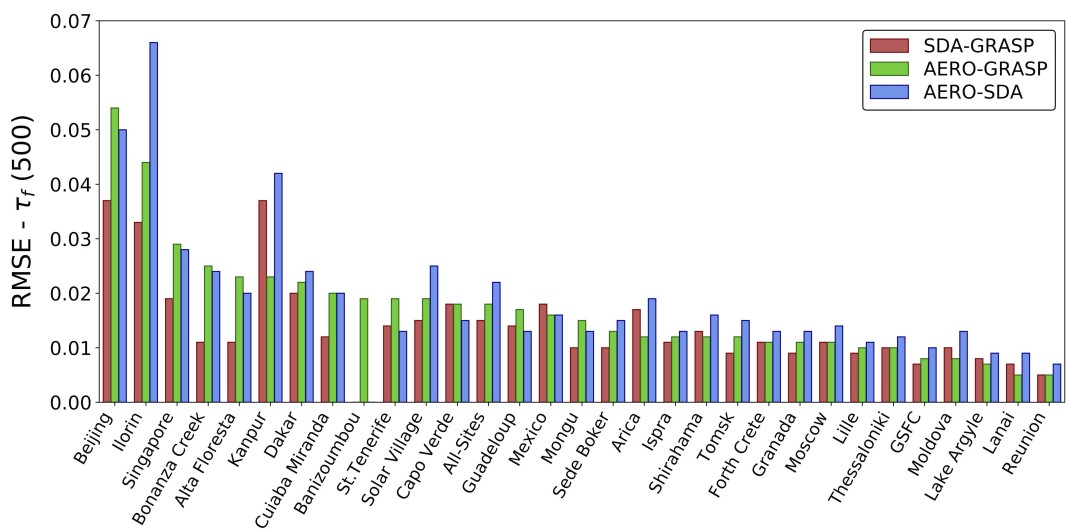

**Figure 2.** RMSE between $\tau_f(500)$ retrieved by SDA and GRASP-AOD (red bars), AERONET aerosol retrieval algorithm and GRASP-AOD (green bars) and AERONET aerosol retrieval algorithm and SDA (blue bars) (values can be found in Table 4) for all the sites considered in the analysis (Table 1) from 1997-2016. The sites have been ordered according to RMSE between AERONET aerosol retrieval algorithm and GRASP-AOD (green bars).

(RMSE values between 0.005 and 0.007), which could be expected since aerosol optical depth values were the lowest for this site.

In Figure 3 we show several examples of the $\tau_f(500)$ correlations retrieved by the different methodologies: a) The top panels
represent the comparisons for all sites for the period 1997-2016. b) The middle panels present the results for GSFC, which is the site with the largest number of $\tau$ measurement and comparisons from all the fine mode predominance sites. c) The bottom panels contain the comparisons for Solar Village, which is the site with the highest number of $\tau$ measurement and comparisons from all the coarse mode predominance sites. From left to right, we illustrate the $\tau_f(500)$ correlations between: SDA and GRASP-AOD, AERONET aerosol retrieval algorithm and GRASP-AOD, and AERONET aerosol retrieval algorithm
and SDA. In all the examples represented, we can observe that correlation coefficients are close to one (as for most of the sites in Tables 4). Regarding the slopes we observe that for GSFC they are almost 1 in all the correlations (values between $0.99-1.04$), while for Solar Village we observe small divergences with slopes ranging from $0.82-1.09$. Moreover, the analysis of the figures shows much greater data dispersion in Solar Village comparisons. Thus, RMSE values are twice as high for Solar Village (0.015-0.025) as for GSFC (0.007-0.010). These differences are even higher in relative terms: the RMSRE (from
Tables 4) are three/four times as large for Solar Village (15%-20%) as for GSFC (4%-7%).

The larger uncertainties observed for Solar Village compared to GSFC can be extrapolated to all sites with coarse mode predominance with respect to the sites with fine mode predominance. To better illustrate this idea, we have represented in Figure 4 the RMSRE (from Table 4) against the averaged Ångström exponent ($< \alpha >$) for each site (Table 1). We can observe

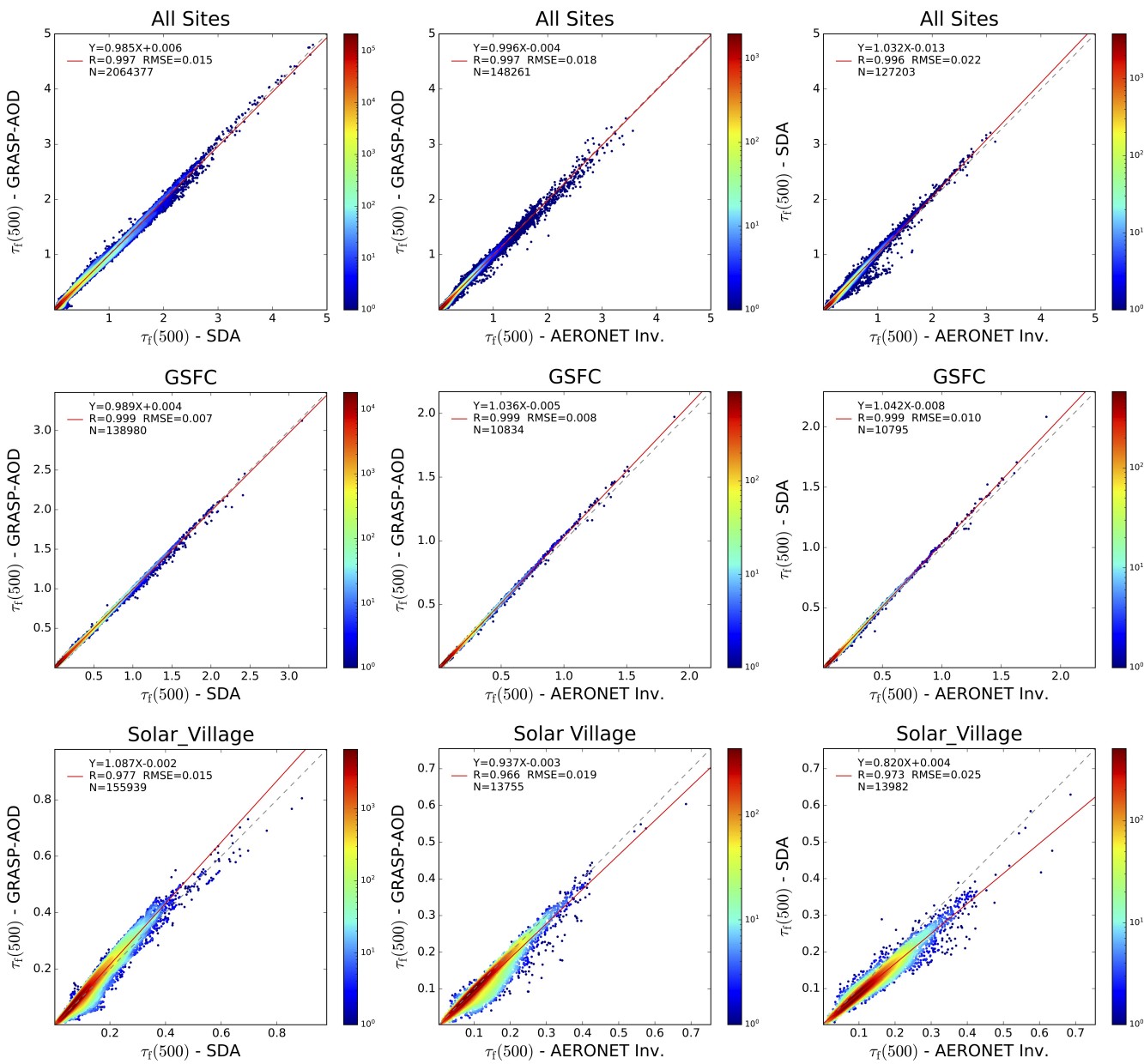

**Figure 3.** Comparisons of $\tau_f(500)$ retrieved from GRASP-AOD, SDA and AERONET: all sites (top subfigures), GSFC site (middle subfigures) and Solar Village site (bottom subfigures). From left to right the comparisons are done between: SDA and GRASP-AOD, AERONET aerosol retrieval algorithm and GRASP-AOD, and AERONET aerosol retrieval algorithm and SDA. Color bars represent data density in a $0.01 \times 0.01$ grid. Logarithmic scale has been chosen given the strong data density at low values.

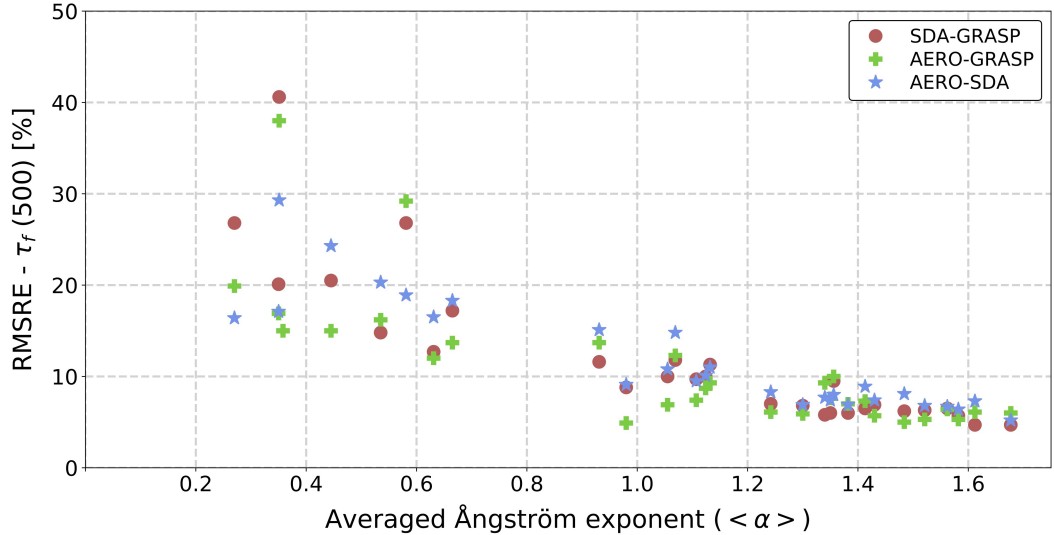

**Figure 4.** RMSRE from the comparisons of $\tau_f(500)$ retrieved by the different methods against the averaged Ångström exponent ($< \alpha >$) for all sites (values can be found in Table 4). RMSRE between SDA and GRASP-AOD are represented by red circles, between AERONET aerosol retrieval algorithm and GRASP-AOD by green crosses, and between AERONET aerosol retrieval algorithm and SDA by blue stars.

that the RMSRE increases as $< \alpha >$ decreases: RMSRE are between $5-10\%$ when $< \alpha >$ values are larger than $1.2$ (fine mode
predominant sites) and between $5-20\%$ for $< \alpha >$ values between $0.6-1.2$ (mixed cases). When $< \alpha >$ values are smaller than $0.6$ (coarse mode predominance sites), the RMSRE typically range between $10-30\%$. The only exception is Guadeloup site, that shows the largest RMSRE observed and they are between $30-40\%$. This site together with Dakar and Capo Verde has one of the lowest values of $< \alpha >$, but it also presents the smallest averaged fine mode optical depth at $500$ nm from all the AERONET aerosol retrievals: $< \tau_f(500) = 0.034 >$. This value is three times lower than that observed at Dakar or Capo
Verde, which may justify these extreme values of the RMSRE, even if the RMSE values only oscillate between $0.014-0.017$.

Figure 5 shows the correlations of $\tau_f(500)$ from all the retrievals but separately for different ranges of the Ångström exponent values: $\alpha < 0.6$ (top subfigures), $\alpha$ between $0.6$ and $1.2$ (middle subfigures) and $\alpha > 1.2$ (bottom subfigures). As in previous figures, the comparisons are presented from left to right for: SDA and GRASP-AOD, AERONET aerosol retrieval algorithm and GRASP-AOD, and AERONET aerosol retrieval algorithm and SDA. Figure 5 confirms that the correlation indices, RMSE
and slopes improve as the Ångström exponent increases. All panels for $\alpha > 0.6$ (middle panels $\alpha$ between $0.6$-$1.2$, and bottom panels $\alpha > 1.2$) show almost a perfect agreement between the different methods if we analyze both correlation coefficients and the slopes.

Significant discrepancies appear only for the cases with $\alpha < 0.6$ (top subfigures). The three subfigures show a much greater data dispersion compared to their equivalents for larger alpha values. The analysis of the figures shows that retrievals of
$\tau_f(500)$ from AERONET aerosol retrieval algorithm are higher on average than from SDA and GRASP-AOD. This result was also found in previous studies (O'Neill et al., 2003; Eck et al., 2010; Torres et al., 2017) and the main explanation is related

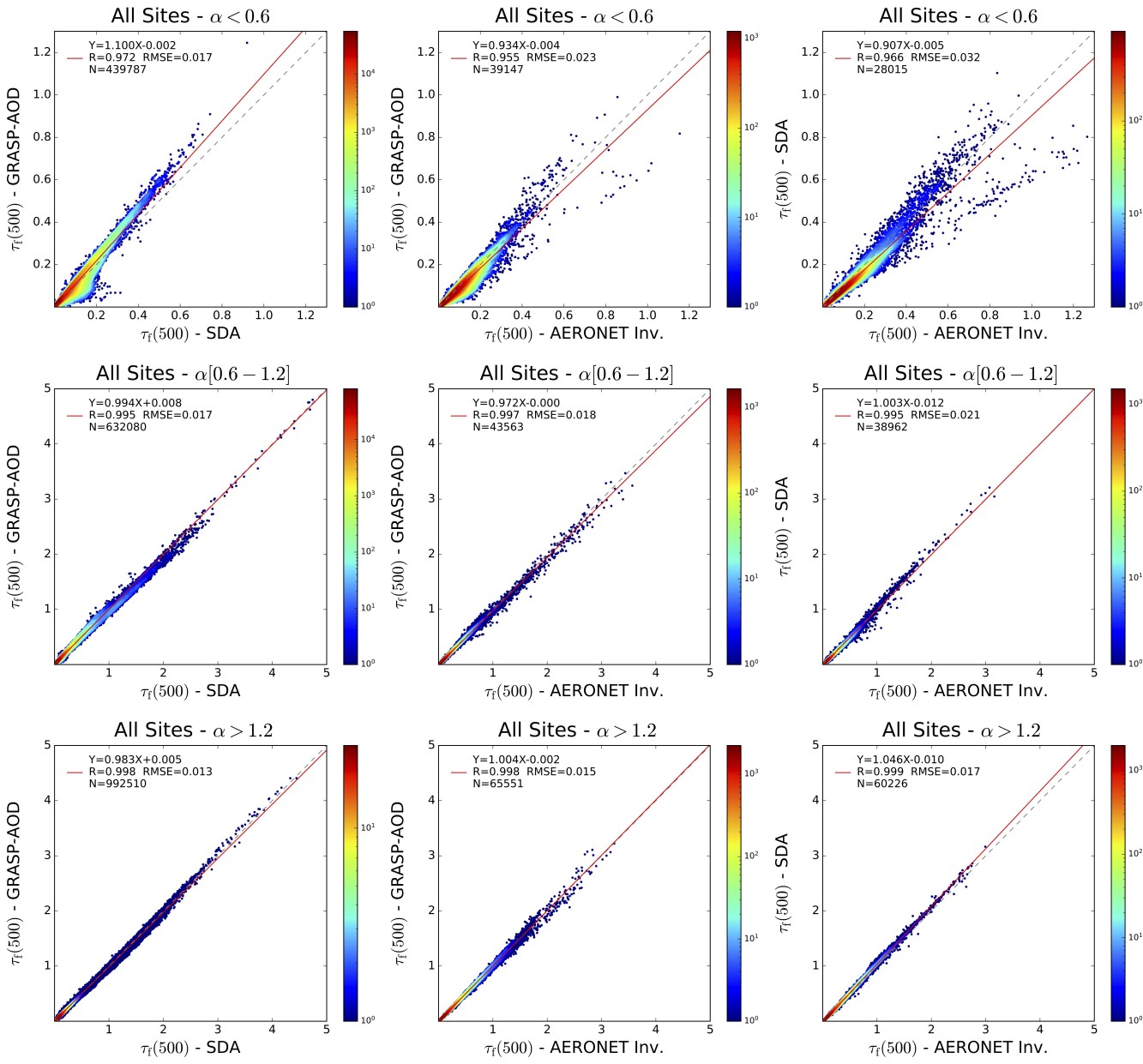

**Figure 5.** Comparisons of $\tau_f(500)$ retrieved from all sites for different ranges of the Ångström exponent values: $\alpha < 0.6$ (top subfigures), $\alpha$ between 0.6 and 1.2 (middle subfigures) and $\alpha > 1.2$ (bottom subfigures). From left to right the comparisons are done between: SDA and GRASP-AOD, AERONET aerosol retrieval algorithm and GRASP-AOD, and AERONET aerosol retrieval algorithm and SDA. Color bars represent data density in a $0.01 \times 0.01$ grid.

to the cutoff process used to define the two modes in AERONET aerosol retrieval algorithm (more details in subsection 2.3.2 or in O'Neill et al. (2003)). The uncertainties caused by the variations in the aerosol refractive index, which are not accounted by SDA and GRASP-AOD, could be also an error source and explain the discrepancies at some retrievals. Additionally, we

observe a second branch for comparisons of GRASP-AOD versus AERONET aerosol retrieval algorithm and SDA versus AERONET aerosol retrieval algorithm (more visible at high $\tau_f(500)$ of AERONET aerosol retrieval algorithm). The reason is related to the three mode structures observed in some desert dust retrievals. In some cases, the cutoff used by AERONET aerosol retrieval algorithm assigns the third midsize mode entirely to the fine mode, which originates high discrepancies with the other two methods, giving rise to this second branch. This effect will be further detailed in subsection 4.1.2.

The underestimation of GRASP-AOD with respect to AERONET aerosol retrieval is mainly located at low $\tau_f(500)$ values while the underestimation of SDA is smoother but is also presented at higher $\tau_f(500)$ values. This behaviour justifies that at coarse mode predominance sites presenting low averaged values of $\tau_f(500)$, the comparisons between GRASP-AOD and AERONET aerosol retrieval have larger RMSE than between SDA and AERONET aerosol retrieval (Guadeloup or St. Cruz de Tenerife). At sites with averaged values of $< \tau_f(500) >$ greater than 0.1, (Ilorin, Solar Village or Sede Boker) the trend is the

opposite and the RMSE values for the comparison between GRASP-AOD and AERONET aerosol retrieval are smaller than those found in the comparisons between SDA and AERONET aerosol retrieval.

Finally, we would like to comment on the results obtained at Banizoumbou - site without $\tau_f(500)$ retrievals from SDA-, and Beijing - site with only 11% of $\tau_f(500)$ retrievals from SDA respect to the total number of $\tau$ measurements (see the end of subsection 2.3.1 for more details). Regarding Banizoumbou, the comparison between GRASP-AOD and AERONET aerosol

retrieval show similar results to those obtained at other sites with coarse mode predominance: RMSE=0.019 which is equivalent to RMSRE=15% (from more than 7 thousand comparisons). For Beijing, the RMSE values obtained between GRASP-AOD and AERONET aerosol retrieval are the highest but is mainly due to the fact that Beijing site presents the highest averaged $\tau_f(500)$ values from all sites: $< \tau_f(500) >$=0.6. Thus, the RMSRE between GRASP-AOD and AERONET aerosol retrieval at Beijing is only 9.3%. This value is similar to those obtained from other sites with $< \alpha >$ values around 1 (see Figure 4). At the

same time, it is smaller than the one obtained from the existing comparisons between SDA and AERONET aerosol retrieval (11.0%). These results obtained at both sites, Beijing and Banizoumbou, indicate that the conditions required in subsection 2.2 assure a robust retrieval of $\tau_f(500)$ from GRASP-AOD, even if $\tau$ measurements at 380 nm and 500 nm are not available (polarized photometers).

## 3.2    Characterization of size parameters

The purpose of this subsection is to validate the volume size distribution parameters obtained from GRASP-AOD through comparisons with the retrievals from the AERONET aerosol algorithm. We will first analyze the standard parameters of the fine mode, after those of the coarse mode. At the end, we will show the comparisons for total volume concentration and effective radius.

### 3.2.1 Fine mode

One of the main conclusions from Torres et al. (2017) was the capacity of GRASP-AOD to accurately characterize the aerosol fine mode size properties, in particular for the cases with a predominant fine mode. Nevertheless, the characterization of fine mode size properties, especially for the fine mode median radius, would depend on a) a reliable a-priori information about the real refractive indices and b) accurate measurements of aerosol optical depths. The use of monthly climatological refractive index (described in subsection 2.2) seems a reasonable strategy, although it gives rise to errors in the retrieval of $R_{Vf}$, especially at sites with a strong variability in the aerosol characteristics (the use of standard refractive index values as an alternative of climatological values is discussed in subsection 4.2). On the other hand, simulation tests in Torres et al. (2017), including aerosol optical errors, showed that the uncertainty of the bimodal log-normal size distribution parameters dramatically increases as the aerosol load decreases. In this regard, a lower limit of $\tau(440)>0.2$ was suggested to assure quality retrievals of all aerosol size distribution parameters. It should be noted here that the lower limit of $\tau(440)>0.2$ was not identified as necessary in the retrieval of $\tau_f(500)$ in Torres et al. (2017). Further tests carried out during this validation study (partially shown in Subsection 3.1) have confirmed this result, indicating that the uncertainty of $\tau_f(500)$ is mainly associated with the Ångström exponent as illustrated in Figure 4. Therefore, the value $\tau(440)>0.02$, which is the general limit already established in subsection 2.2, stands also as a quality assurance threshold for $\tau_f(500)$.

Top panels on Figure 6 illustrate the comparisons between the fine mode volume median radius obtained by GRASP-AOD and AERONET aerosol retrieval algorithm using three different lower limits on the aerosol load, from left to right: $\tau(440) > 0.02$ (threshold established for all GRASP-AOD retrievals), $\tau(440) > 0.2$, and $\tau(440) > 0.4$. The analysis of the three panels indicates that correlation parameters improve as the $\tau(440)$ lower limit increases. The RMSE diminishes from 0.040 $\mu$m (RMSRE=25.6%), for the case with all retrievals, to 0.032 $\mu$m (19.8%) when we include $\tau(440) > 0.2$ as lower limit. The most restrictive limit $\tau(440) > 0.4$ hardly improves the RMSE (0.030 $\mu$m or RMSRE=18.3%), and the rest of the correlation parameters, while it does eliminate more than half of the data with respect to the limit $\tau(440) > 0.2$. Therefore, the lower limit of $\tau(440)=0.2$ suggested in Torres et al. (2017) seems a good compromise.

The bottom panels on Figure 6 represent the comparisons for the retrievals with $\tau(440) > 0.2$ separately for different ranges of the Ångström exponent, from left to right: retrievals with $\alpha > 1.2$, retrievals with $\alpha$ between 0.6 and 1.2, and finally, retrievals with $\alpha < 0.6$. We observe that the range with $\alpha > 1.2$ shows the best retrievals with a RMSE=0.023 $\mu$m (13.9%) and a correlation coefficient of 0.81. The comparison for the cases with $\alpha$ between 0.6-1.2 presents also a reasonable agreement with a RMSE=0.032 $\mu$m (18.7%) and a correlation coefficient of 0.787. The results for $\alpha < 0.6$ indicates a much lower sensitivity when there is a coarse mode predominance. In these conditions, the correlation coefficient is practically zero[4] and the RMSE=0.039 $\mu$m (29.8%).

As previously indicated, the study Torres et al. (2017) identified fine mode predominance as a key factor to accurately describe $R_{Vf}$ from GRASP-AOD inversion. However, the authors did not suggest any limits to assure the quality in the retrievals.

---

[4]We have largely underlined the low sensitivity in the retrieval of $R_{Vf}$ by using GRASP-AOD algorithm when the coarse mode dominates, which is certainly the main reason to explain the poor correlation found here when $\alpha < 0.6$. However, it should be also noted that in such conditions there is also a larger uncertainty in the retrieval of $R_{Vf}$ by the AERONET aerosol retrieval algorithm as pointed out in Sinyuk et al. (2020).

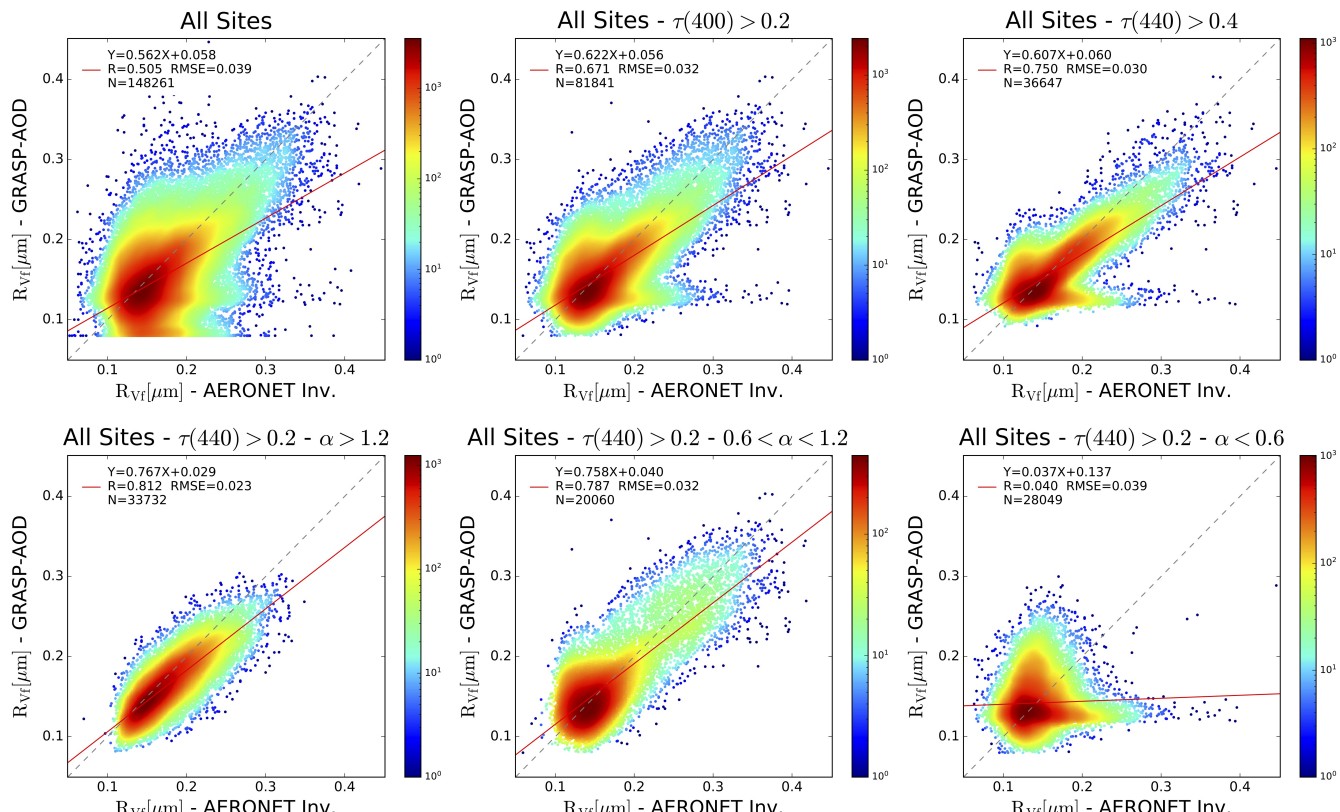

**Figure 6.** Comparisons between the fine mode volume median radius ($R_{Vf}[\mu m]$), obtained by GRASP-AOD and AERONET aerosol retrieval algorithm for all sites considered in the analysis (Table 1) for the period 1997-2016, using different thresholds for $\tau(440)$ and Ångström exponent values. The top subfigures analyze the effect of different lower limits of $\tau(440)$, from left to right: all retrievals ($\tau(440) > 0.02$), retrievals with $\tau(440) > 0.2$, and retrievals with $\tau(440) > 0.4$. The bottom subfigures analyse the results for the retrievals with $\tau(440) > 0.2$ and for different ranges of Ångström exponent, from left to right: retrievals with $\alpha > 1.2$, retrievals with $\alpha$ between 0.6 and 1.2, and finally, retrievals with $\alpha < 0.6$. Color bars represent data density in a $0.01 \times 0.01$ $\mu m$ grid. Logarithmic scale has been chosen given the strong data density between $0.12 - 0.16$ $\mu m$.

Here, we observe that if $\alpha > 1.2$ the characterization of $R_{Vf}$ becomes much more reliable. In such conditions, the uncertainty of $R_{Vf}$, with a relative error under 15%, is the lowest found for all size volume aerosol parameters. Note that for the other size parameters the relative errors typically range between 20%-30%, when $\tau(440) > 0.2$ (presented in the next subsections). If not conditions on $\alpha$ values are required, the characterization of $R_{Vf}$ presents similar results as the other size parameters (see top-middle panel on Figure 6 with a relative error of almost 20% when $\tau(440) > 0.2$).

Given the excellence of the results obtained in the characterization of $R_{Vf}$ when $\tau(440) > 0.2$ and $\alpha > 1.2$, we consider the interest of presenting the comparison results by sites in such conditions. Thus, first part of Table 6 depicts the parameters obtained from the comparison of $R_{Vf}$ between AERONET aerosol retrieval algorithm and GRASP-AOD. First two columns

**Table 6.** Comparison between fine mode size parameters obtained from AERONET standard inversion and GRASP-AOD. First column presents the site and second column the number of coincident retrievals accomplishing that $\tau(440) > 0.2$ and $\alpha > 1.2$. The percentage with respect to the total number of coincident retrievals is indicated in parentheses. Columns from three to six show the comparison results for volume median radius, while columns from seven to ten present the results for the fine volume concentration. In both cases, RMSE (and RMSRE enclosed in parentheses) correlation coefficients, slopes and intercepts from linear regressions are shown.

| Sites | N° meas. | $R_{Vf}$ | | | | $C_{Vf}$ | | | |
|---|---|---|---|---|---|---|---|---|---|
| | | RMSE | Coeff. -R- | Slope | Intercept | RMSE | Coeff. -R- | Slope | Intercept |
| Alta Floresta | 933 (35%) | 0.016 (10%) | 0.75 | 0.733 | 0.031 | 0.019 (20%) | 0.96 | 0.975 | 0.004 |
| Arica | 829 (18%) | 0.028 (14%) | 0.67 | 0.667 | 0.065 | 0.014 (30%) | 0.83 | 0.629 | 0.012 |
| Beijing | 1679 (42%) | 0.024 (13%) | 0.81 | 0.633 | 0.058 | 0.031 (29%) | 0.96 | 0.786 | 0.006 |
| Bonanza Creek | 271 (30%) | 0.024 (13%) | 0.82 | 0.813 | 0.02 | 0.016 (19%) | 0.99 | 0.896 | 0 |
| Cuiaba Miranda | 898 (43%) | 0.018 (12%) | 0.73 | 0.792 | 0.019 | 0.017 (20%) | 0.96 | 0.942 | 0.007 |
| Forth Crete | 1134 (29%) | 0.034 (22%) | 0.48 | 0.496 | 0.058 | 0.011 (28%) | 0.69 | 0.493 | 0.019 |
| GSFC | 2806 (26%) | 0.024 (14%) | 0.84 | 0.798 | 0.022 | 0.011 (21%) | 0.94 | 0.855 | 0.009 |
| Granada | 601 (8%) | 0.030 (19%) | 0.73 | 0.663 | 0.034 | 0.011 (35%) | 0.43 | 0.260 | 0.023 |
| Ilorin | 214 (8%) | 0.014 (10%) | 0.79 | 0.613 | 0.060 | 0.029 (29%) | 0.85 | 0.849 | 0.024 |
| Ispra | 2150 (54%) | 0.024 (13%) | 0.81 | 0.789 | 0.031 | 0.015 (25%) | 0.92 | 0.769 | 0.012 |
| Kanpur | 4054 (43%) | 0.021 (11%) | 0.80 | 0.628 | 0.059 | 0.016 (19%) | 0.95 | 0.782 | 0.012 |
| Lake Argyle | 1486 (20%) | 0.015 (12%) | 0.45 | 0.596 | 0.048 | 0.019 (34%) | 0.76 | 0.578 | 0.019 |
| Lille | 1217 (41%) | 0.028 (14%) | 0.74 | 0.623 | 0.066 | 0.011 (24%) | 0.91 | 0.718 | 0.011 |
| Mexico City | 1617 (71%) | 0.022 (13%) | 0.78 | 0.682 | 0.063 | 0.017 (32%) | 0.84 | 0.738 | 0.008 |
| Moldova | 2512 (44%) | 0.022 (13%) | 0.77 | 0.682 | 0.042 | 0.012 (29%) | 0.83 | 0.649 | 0.013 |
| Mongu | 2893 (60%) | 0.017 (13%) | 0.67 | 0.715 | 0.028 | 0.013 (21%) | 0.93 | 0.883 | 0.008 |
| Moscow | 1062 (48%) | 0.019 (12%) | 0.75 | 0.663 | 0.048 | 0.015 (31%) | 0.91 | 0.872 | 0.001 |
| Sede Boker | 1214 (8%) | 0.045 (25%) | 0.67 | 0.457 | 0.065 | 0.006 (22%) | 0.75 | 0.592 | 0.011 |
| Shirahama | 1736 (39%) | 0.023 (12%) | 0.80 | 0.71 | 0.046 | 0.016 (31%) | 0.85 | 0.67 | 0.012 |
| Singapore | 432 (78%) | 0.017 (9%) | 0.80 | 0.813 | 0.037 | 0.018 (23%) | 0.96 | 0.934 | -0.001 |
| Solar Village | 165 (1%) | 0.034 (22%) | 0.71 | 0.461 | 0.059 | 0.008 (28%) | 0.84 | 0.560 | 0.014 |
| Thessaloniki | 3401 (56%) | 0.017 (11%) | 0.75 | 0.827 | 0.023 | 0.015 (28%) | 0.82 | 0.610 | 0.017 |
| Tomsk | 258 (35%) | 0.029 (19%) | 0.80 | 0.593 | 0.047 | 0.015 (30%) | 0.93 | 0.841 | 0.003 |
| All Sites | 33732 (23%) | 0.023 (14%) | 0.81 | 0.767 | 0.029 | 0.016 (26%) | 0.94 | 0.818 | 0.008 |

contain the name of the site[5] and the number of coincident measurement accomplishing the quality assured conditions (the percentage with respect to the total number of coincident retrievals is indicated in parentheses). Third column shows the RMSE obtained from the two retrievals with the RMSRE in parentheses. Columns from four to six present the values of correlation coefficients, slopes and intercepts. Apart from the results by sites, we include a last row summarizing the results for all sites together. Note that the results of this last row, all sites when $\tau(440) > 0.2$ and $\alpha > 1.2$, were illustrated at the bottom left panel in Figure 6.

Analysing the results of Table 6, sites with fine mode domination and significant aerosol load along the year present the largest data percentage (over $40\%$) accomplishing the fore-mentioned criteria (e.g. Kanpur, Lille, Ispra, Mexico City or Mongu), as expected. These sites show generally the lowest RMSRE values (between 9-13%) for the comparison of $R_{Vf}$ obtained by GRASP-AOD and AERONET aerosol retrieval algorithm. The correlation coefficients are typically larger than $0.75$ and the slopes are between 0.6-0.85. Top panels on Figure 7 illustrate the comparisons for three sites with these characteristics (Ispra, Lille and Kanpur).

Lower data percentage (between $18 - 40\%$) are presented at sites with fine mode predominance but with lower aerosol load throughout the year than at the earlier sites (e.g. Arica, GSFC or Tomsk). The RMSRE of $R_{Vf}$ comparisons at this group are a bit higher (between 12-19%) than at the previous group. However, correlations coefficients and slopes show similar values (0.6-0.8). Three examples of this group (GSFC, Arica and Shirahama) are shown at middle panels of Figure 7.

The lowest data percentages (under $20\%$) are obtained for those sites regularly affected by desert dust episodes: Granada, Ilorin, Lake Argyle, Sede Boker and Solar Village. The only exception is Forth Crete, which should be included in this group even though it shows a higher data percentage (29%). The correlations and slopes values are significantly lower compared to the precedent groups (from $0.45$ to $0.75$). Correlations for Ilorin, Granada and Sede Boker are shown at the bottom panels of Figure 7. Note that data variation of $R_{Vf}$ is considerably narrower at these last sites compared to the previous group (as can be gained from Figure 7) and values of $R_{Vf}$ rarely exceed $0.25 \ \mu m$. This fact justifies that, even if correlation coefficient are quite small (for instance 0.45 in Lake Argyle), the RMSRE values are still quite low (between $12\%$ and $25\%$). On the other hand, the use of climatological values for the refractive index may induce a larger error in the retrieval of $R_{Vf}$ at this group compared to previous groups. Thus, the monthly averages of real refractive index, estimated from Level 2.0 of Version 3 AERONET aerosol retrieval algorithm, are dominated by the frequent desert dust episodes occurring at these sites. These values may significantly differ from typical real refractive index values of the data selected here ($\tau(440) > 0.2$ and $\alpha > 1.2$). Future reprocessings using more developed climatologies (e.g. considering different values for different Ångström exponents) may improve the results obtained in this study.

Finally, we would like to mention that there is a certain bias between GRASP-AOD and AERONET in $R_{Vf}$ retrievals. The total bias is $-0.011 \ \mu m$ ($-5.7\%$ in relative terms) though we observe that it increases for higher values of $R_{Vf}$: from only $-0.002 \ \mu m$ (or $-1.4\%$) when $R_{Vf} < 0.14 \ \mu m$ to $-0.029 \ \mu m$ (or $-12.7\%$) when $R_{Vf} > 0.26 \ \mu m$. That explains why all the slopes in Table 6 (or in Figure 7) are under 1. A possible explanation could be related to a general loss of sensitivity in the

---

[5]The results for the sites with less than fifty points (which includes several with no points at all) are not considered in the table: Banizoumbou, Capo Verde, Dakar, Guadeloup , Lanai, Reunion St. Denis and Sta. Cruz de Tenerife.

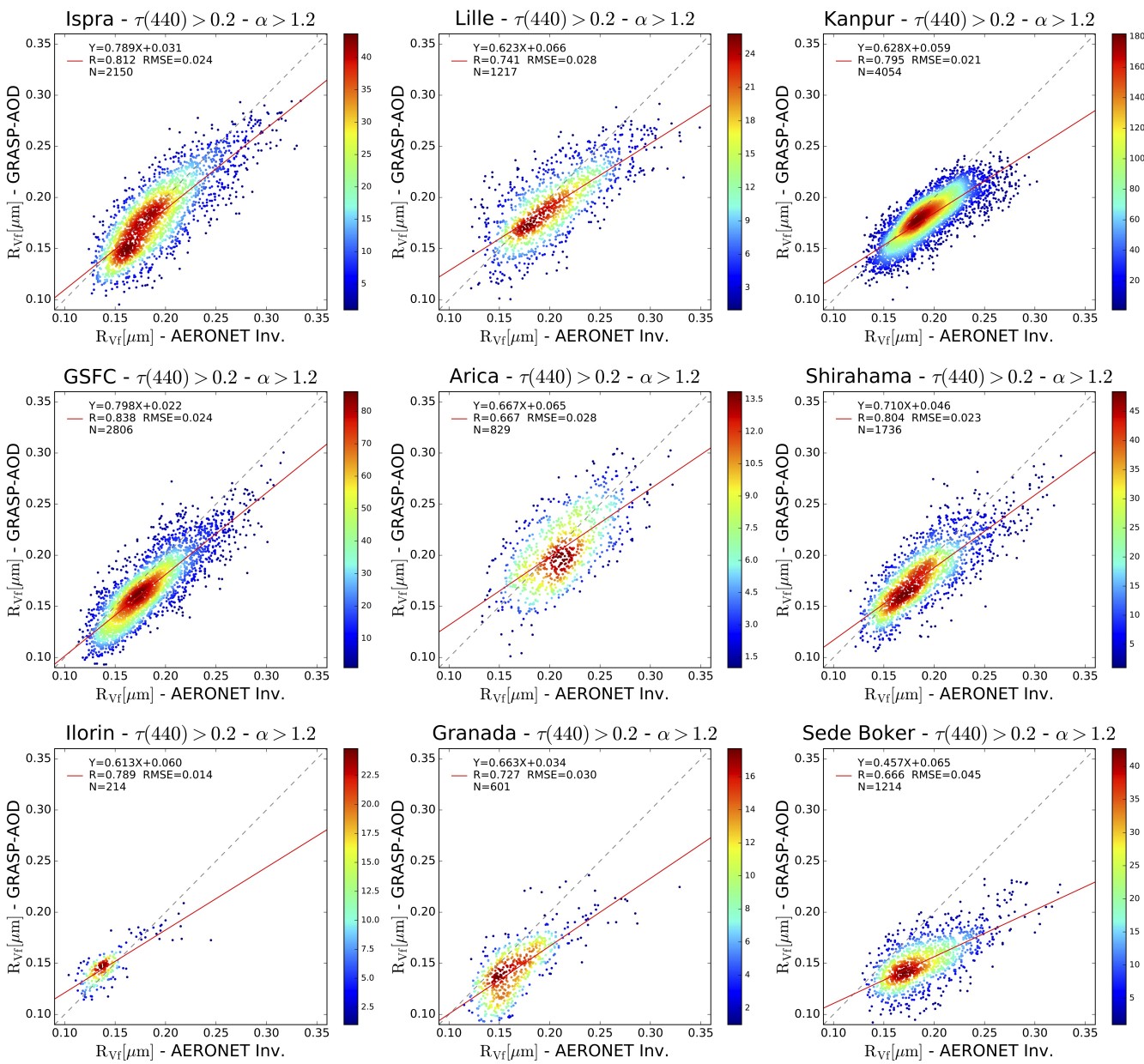

**Figure 7.** Comparisons between the fine mode volume median radius ($R_{Vf}[\mu m]$), obtained by GRASP-AOD and AERONET aerosol retrieval algorithm during the period 1997-2016 for some selected sites (from top to bottom and from left to right: Ispra, Lille, Kanpur, GSFC, Arica, Shirahama, Ilorin, Granada and Sede Boker). Note that comparisons include only the data accomplishing the thresholds $\tau(440) > 0.2$ and $\alpha > 1.2$ (same as in Table 6). Color bars represent data density in a $0.01 \times 0.01 \mu m$ grid. We have intentionally kept the same X-Y scale in all the figures.

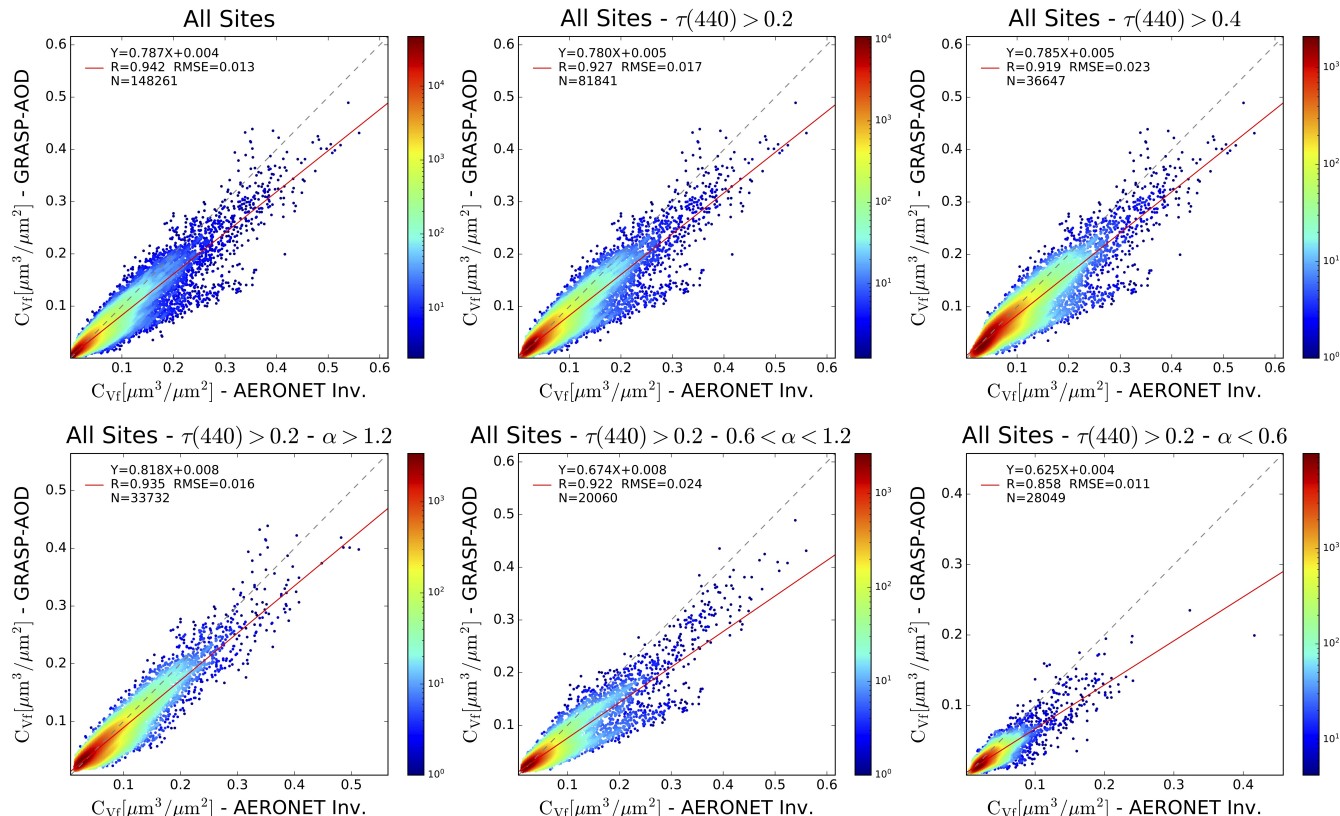

**Figure 8.** Comparisons between the fine mode volume concentration ($C_{Vf}[\mu m^3/\mu m^2]$), obtained by GRASP-AOD and AERONET aerosol retrieval algorithm for all the sites considered in the analysis (Table 1) for the period 1997-2016, using different thresholds for $\tau(440)$ and Ångström exponent values. Top subfigures analyze the effect of different lower limits of $\tau(440)$, from left to right: all retrievals ($\tau(440) > 0.02$), retrievals with $\tau(440) > 0.2$, and retrievals with $\tau(440) > 0.4$. Bottom subfigures analyse the results for the retrievals with $\tau(440) > 0.2$ and for different ranges of Ångström exponent, from left to right: retrievals with $\alpha > 1.2$, retrievals with $\alpha$ between 0.6 and 1.2, and finally, retrievals with $\alpha < 0.6$. Color bars represent data density in a $0.01 \times 0.01 \mu m^3/\mu m^2$ grid. Logarithmic scale has been chosen given the strong data density at low values.

GRASP-AOD retrieval as $R_{Vf}$ increases. Thus, if we analyze the variation of the extinction coefficient in function of the size parameter ($\chi = 2\pi R/\lambda$), we observe that there is a strong variation from $\chi = 0.5 - 2.5$ which becomes smoother for $\chi > 2.5$ since the extinction coefficient arrives to its maximum (see for instance Figure 3 from Tonna et al. (1995) or Figure 2.10 from Lenoble et al. (2013)). For radii around $0.14$ $\mu m$, the size parameters for all the considered wavelengths at this study are between 0.6 and 2.2. At $R_{Vf} = 0.23$ $\mu m$ half of the channels are already out of the so-called maximum sensitivity interval ($\chi(\lambda=500$ nm$) = 2.9$). Nevertheless, the retrieval is still quite sensitive even if we limit the analysis to $R_{Vf} > 0.23$ $\mu m$; in these conditions, the RMSE value is $0.039$ $\mu m$ (or $17\%$ in relative terms) and the correlation coefficient is larger than 0.6.

Figure 8 shows the comparison of the fine mode volume concentration ($C_{Vf}$) obtained by GRASP-AOD and AERONET aerosol retrieval algorithm. Similarly as on Figure 6, three different lower limits on the aerosol load are used at the top panels, from left to right: $\tau(440) > 0.02$, $\tau(440) > 0.2$, and $\tau(440) > 0.4$. Correlation coefficients are over $0.91$ in the three graphics, which is significantly better than for $R_{Vf}$ comparisons. This is mainly due to the much larger variability for the concentration values. The slopes (between 0.78-0.79) and intercepts values (0.004-0.005) are similar between the three cases regardless the $\tau(440)$ limit. Significant differences can be observed only in the RMSE value which increases as the lower limit rises: $0.013$ $\mu m^3/\mu m^2$ for all the retrievals, $0.017$ $\mu m^3/\mu m^2$ when $\tau(440) > 0.2$ and $0.023$ $\mu m^3/\mu m^2$ if $\tau(440) > 0.4$. However, the relative value (RMSRE) decreases as the lower limit increases: $42.6\%$ for all the retrievals, $36.8\%$ for $\tau(440) > 0.2$ and $34.5\%$ when $\tau(440) > 0.4$. Once again, we observe that the most restrictive limit $\tau(440) > 0.4$ hardly improves the RMSRE with respect to the limit $\tau(440) > 0.2$ while it eliminates half of the data. Therefore, the threshold $\tau(440)=0.2$ proposed by Torres et al. (2017) seems to be adequate also here.

Bottom panels on Figure 8 represent the $C_{Vf}$ comparisons when $\tau(440) > 0.2$ for different ranges of the Ångström exponent: retrievals with $\alpha > 1.2$, retrievals with $\alpha$ between 0.6 and 1.2, and retrievals with $\alpha < 0.6$. The best results are obtained for the case $\alpha > 1.2$ with a slope of $0.82$, a correlation coefficient of $0.94$ and RMSE=$0.016$ $\mu m^3/\mu m^2$, which is equivalent to RMSRE=$26\%$. Although the lowest RMSE is observed for the case $\alpha < 0.6$, the RMSRE=$42\%$ is the largest in relative terms. The comparison for the cases with $\alpha$ between 0.6-1.2 presents a RMSE=$0.024$ $\mu m^3/\mu m^2$ (RMSRE=$40\%$).

Second part of Table 6 presents the comparisons by sites for $C_{Vf}$, when $\tau(440) > 0.2$ and $\alpha > 1.2$. The correlation coefficients and the slopes are between $0.8 - 1.0$ for most of the sites which indicates a good correlation by sites in general terms. In addition, all RMSRE values are between $19 - 35\%$. The lowest values (around $20\%$) are mainly obtained for the sites with a predominant fine mode (e.g. Kanpur, Bonanza Cree, Mongu or GSFC). On the other hand, sites with regular presence of desert dust depict the highest RMSRE values (see Granada or Lake Argyle). Nevertheless, there are some exceptions to both statements, see for instance the relatively low RMSRE value of $22\%$ found at Sede Boker, or the relatively high found at Moscow and Shirahama (RMSRE=$31\%$ in both cases).

### 3.2.2 Coarse mode

The study by Torres et al. (2017) pointed out that the characterization of coarse mode size properties by GRASP-AOD is less accurate compared to the characterization of fine mode. This is mainly due the much lower sensitivity of the spectral $\tau$ measurements (in the spectral range between 340-1020 nm) to the coarse mode size distribution. In this regards, the study by Torres et al. (2017) recommended the use of moderate a priori information about coarse mode parameters to significantly improved the characterization. The values of the multiple initial guess approach (values in Table 2) used in this first validation analysis are certainly inspired by typical AERONET climatological values, for example $R_{Vc}$=1.9-2.3 $\mu m$ for desert cases (typically $\alpha <0.6$). However, they do not account for possible peculiarities of a particular site. A discussion with ideas about how to improve the coarse mode characterization is presented in Section 4.3. Here, we limit the analysis to the general results based in the methodology described in the subsection 2.2 (which includes the multiple initial guess approach shown in Table 2).

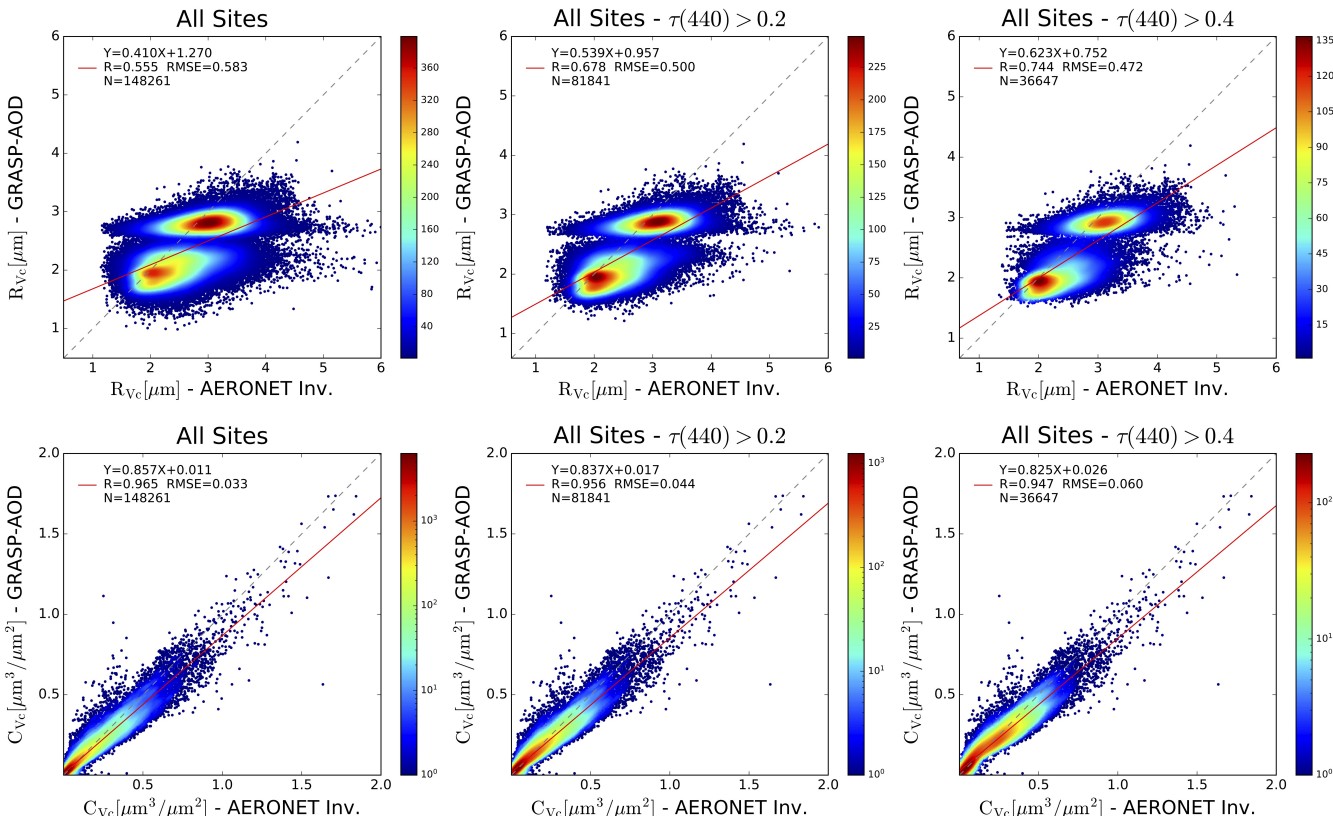

**Figure 9.** Comparisons between the coarse mode volume properties obtained by GRASP-AOD and AERONET aerosol retrieval algorithm for all the sites considered in the analysis (Table 1) during the period 1997-2016. Comparisons for the coarse mode volume median radius ($R_{Vc}[\mu m]$) are represented at the top panels, while comparisons for the coarse mode volume concentration ($C_{Vc}[\mu m^3/\mu m^2]$) are shown at the bottom panels. Different thresholds for $\tau(440)$ have been applied in the comparisons, from left to right: all the retrievals ($\tau(440) > 0.02$), retrievals with $\tau(440) > 0.2$, and retrievals with $\tau(440) > 0.4$. Color bars represent data density in a $0.05 \times 0.05$ $\mu m$ grid for $R_{Vc}$ and in $0.01 \times 0.01 \mu m^3/\mu m^2$ grid for $C_{Vc}$. For the volume concentration, a logarithmic scale has been chosen given the strong data density at low values.

Top panels of Figure 9 show the comparisons between the coarse mode volume median radius obtained by GRASP-AOD and AERONET aerosol retrieval algorithm for all the sites considered in the analysis, during the period 1997-2016 and using three different lower limits on the aerosol load; from left to right: $\tau(440) > 0.02$ (threshold established for all GRASP-AOD retrievals), $\tau(440) > 0.2$, and $\tau(440) > 0.4$. We can observe how the correlation coefficients and the slopes improve as the $\tau(440)$ lower limit increases. The same happens with RMSE and RMSRE: 0.583 $\mu m$ (23.2%) when $\tau(440) > 0.02$, 0.500 $\mu m$ (20%) when $\tau(440) > 0.2$ and 0.472 $\mu m$ (18.8%) when $\tau(440) > 0.4$. Analysing those values, the threshold of $\tau(440) > 0.2$ suggested by Torres et al. (2017) to derive aerosol size properties seems a good compromise for the retrieval of $R_{Vc}$ as well. Unlike the retrieval of $R_{Vf}$, filtering the retrievals by the Ångström exponent do not present any improvements in the

characterization of $R_{Vc}$: the analysis results in similar RMSRE values at different Ångström exponent ranges (not shown in the figure).

On the other hand, we observe the presence of two main clusters in the comparison between the retrievals of $R_{Vc}$ for the three different thresholds. The main reason for the appearance of these clusters is related to the limited sensitivity of GRASP-AOD to coarse mode retrieval. As a matter of fact, the retrieved $R_{Vc}$ does not typically present strong variation respect to the considered initial guess value (given in Table 2). In these regards, the first cluster around 1.7-2.1 $\mu$m is associated to the initial guess election for low $\alpha$ values. The second cluster, which is centered in 3.0 $\mu$m, correspond to the election for the GRASP-AOD retrievals with larger $\alpha$ values. As pointed out in the introduction, a good election of these initial guess values is key for a correct characterization if only optical depth values are considered. This aspect is revisited in subsection 4.3.

First part of Table 7 shows the main parameters of the comparison between $R_{Vc}$ retrievals obtained from AERONET aerosol algorithm and GRASP-AOD when $\tau(440) > 0.2$ by sites. We notice that the RMSRE for almost all the sites are between $15 - 25\%$, without a clear tendency regarding the aerosol type of the sites. This result could be expected since, as previously commented, we have not observed a clear dependence of the errors on the value of the Ångström exponent. On the other hand, the values of the correlation coefficients by sites are mostly between 0.25-0.5, which is significantly lower than the value of 0.68 found when analyzing the retrievals from all the sites together (last row in Table 7 or top-middle panel at Figure 9). A similar result is obtained in the characterization of the slopes. This can be partly explained by the fact that $R_{Vc}$ does not present strong variations for a same aerosol type that typically predominates in a given site. However, there is a significant variation when all sites are considered. The result indicates that GRASP-AOD is not sensitive to the small oscillations of $R_{Vc}$ occurred for individual aerosol types at the different sites, but it gives a reasonable characterization overall, which is mainly due to an optimal election of the initial guess.

Bottom panels at Figure 9 present the comparison for the coarse mode volume concentration ($C_{Vc}$), for three different thresholds of the aerosol load; from left to right: $\tau(440) > 0.02$, $\tau(440) > 0.2$, and $\tau(440) > 0.4$. As in the characterization of $C_{Vf}$, the three correlation coefficients are greater than 0.9, and the three slopes are close to 0.8. We observe that as the $\tau(440)$ lower limit increases the RMSE value increases. However, it decreases in relative terms: 0.033 $\mu$m (35.7%) when $\tau(440) > 0.02$, 0.044 $\mu$m (30.5%) when $\tau(440) > 0.2$ and 0.060 $\mu$m (28.8%) when $\tau(440) > 0.4$. The $\tau(440) > 0.4$ threshold eliminate half of the data but it only improves 1.7% the RMSRE with respect to the limit $\tau(440) > 0.2$. In these regards, the latter seems a good compromise also for the retrieval of $C_{Vc}$. If we filter the retrievals by the Ångström exponent, the RMSRE diminishes for lower $\alpha$ values. For instance, if we consider the threshold $\tau(440) > 0.2$, we obtain: RMSRE=24% (RMSE=0.06 $\mu$m$^3/\mu$m$^2$) when $\alpha < 0.6$, RMSRE=28% (RMSE=0.04 $\mu$m$^3/\mu$m$^2$) when $0.6 < \alpha < 1.2$ and RMSRE=46% (RMSE=0.03 $\mu$m$^3/\mu$m$^2$) when $\alpha > 1.2$. The main reason for this result is the much higher $C_{Vc}$ values, as a consequence of the larger coarse mode contribution when Ångström exponent values are smaller.

The comparison results by sites can be found in the second part of Table 7. The analysis of RMSRE values shows lower relative errors for sites with a predominant coarse mode, which is in line with the result obtained filtering by Ångström exponent values. Thus, the RMSRE values for the sites with a predominance of coarse mode go from 13% to 26%, while the values for the rest of the sites go from 30% up to 70%. The analysis of the correlation coefficients shows values between 0.8 and 1.0

**Table 7.** Comparison between coarse mode size parameters obtained from AERONET standard inversion and GRASP-AOD. First column presents the site and second column the number of coincident retrievals accomplishing that $\tau(440) > 0.2$. The percentage with respect to the total number of coincident retrievals is indicated in parentheses. Columns from three to six show the comparison results for coarse mode volume median radius, while columns from seven to ten present the results for the coarse volume concentration. In both cases, RMSE (and RMSRE enclosed in parentheses) correlation coefficients, slopes and intercepts from linear regressions are shown.

| Sites | N° meas. | $R_{Vc}$ | | | | $C_{Vc}$ | | | |
|---|---|---|---|---|---|---|---|---|---|
| | | RMSE | Coeff. -R- | Slope | Intercept | RMSE | Coeff. -R- | Slope | Intercept |
| Alta Floresta | 939 (35%) | 0.52 (16.6%) | 0.58 | 0.226 | 2.299 | 0.029 (53.8%) | 0.62 | 0.856 | 0.005 |
| Arica | 2953 (60%) | 0.856 (31%) | 0.24 | 0.135 | 2.816 | 0.016 (28%) | 0.83 | 0.748 | 0.016 |
| Banizoumbou | 5809 (81%) | 0.414 (20%) | 0.17 | 0.096 | 1.724 | 0.051 (20.8%) | 0.97 | 0.865 | 0.008 |
| Beijing | 3220 (80%) | 0.542 (19.4%) | 0.40 | 0.39 | 1.492 | 0.078 (45.8%) | 0.83 | 0.983 | 0.037 |
| Bonanza Creek | 277 (31%) | 0.748 (24.6%) | 0.35 | 0.108 | 2.635 | 0.04 (70.5%) | 0.68 | 1.278 | 0.01 |
| Cuiaba Miranda | 901 (43%) | 0.465 (14.9%) | 0.41 | 0.212 | 2.283 | 0.026 (42.2%) | 0.64 | 0.663 | 0.014 |
| Capo Verde | 2894 (70%) | 0.23 (12.1%) | 0.24 | 0.208 | 1.48 | 0.03 (13.7%) | 0.97 | 0.941 | 0.008 |
| Dakar | 5715 (88%) | 0.323 (15.8%) | 0.30 | 0.193 | 1.559 | 0.04 (17.8%) | 0.96 | 0.881 | 0.013 |
| Forth Crete | 2043 (52%) | 0.37 (15.2%) | 0.65 | 0.609 | 0.965 | 0.022 (26.2%) | 0.96 | 0.817 | 0.023 |
| GSFC | 2881 (27%) | 0.503 (17%) | 0.56 | 0.173 | 2.372 | 0.015 (50.1%) | 0.81 | 1.154 | 0.002 |
| Granada | 2090 (29%) | 0.461 (20.1%) | 0.73 | 0.546 | 0.827 | 0.025 (22.4%) | 0.96 | 0.778 | 0.022 |
| Guadeloup | 249 (26%) | 0.266 (13.8%) | 0.34 | 0.288 | 1.352 | 0.032 (15.1%) | 0.93 | 0.806 | 0.033 |
| Ilorin | 2565 (99%) | 0.42 (18.1%) | 0.30 | 0.254 | 1.606 | 0.073 (22.7%) | 0.94 | 0.814 | 0.046 |
| Ispra | 2317 (58%) | 0.541 (18.4%) | 0.30 | 0.105 | 2.513 | 0.017 (44.1%) | 0.87 | 1.032 | 0.002 |
| Kanpur | 9391 (99%) | 0.546 (20.6%) | 0.50 | 0.558 | 0.882 | 0.068 (33.9%) | 0.94 | 0.712 | 0.046 |
| Lake Argyle | 1642 (22%) | 0.474 (16.9%) | 0.30 | 0.168 | 2.25 | 0.02 (33.3%) | 0.89 | 0.785 | 0.019 |
| Lille | 1392 (47%) | 0.546 (19.6%) | 0.42 | 0.164 | 2.28 | 0.019 (46.9%) | 0.88 | 1.01 | 0.011 |
| Mexico City | 1658 (73%) | 0.761 (23.7%) | 0.18 | 0.15 | 2.892 | 0.02 (44%) | 0.69 | 0.765 | 0.002 |
| Moldova | 2854 (50%) | 0.53 (18.3%) | 0.52 | 0.265 | 1.945 | 0.014 (28.6%) | 0.92 | 0.934 | 0.009 |
| Mongu | 2896 (60%) | 0.619 (19.3%) | 0.28 | 0.076 | 2.692 | 0.016 (49.5%) | 0.63 | 0.781 | 0.004 |
| Moscow | 1129 (51%) | 0.467 (15.8%) | 0.39 | 0.175 | 2.274 | 0.018 (37.8%) | 0.82 | 0.931 | 0.01 |
| Reunion - St. Denis | 66 (2%) | 0.371 (14.1%) | 0.31 | 0.145 | 2.355 | 0.013 (28.4%) | 0.92 | 0.881 | 0.016 |
| Sede Boker | 6204 (40%) | 0.422 (18.4%) | 0.61 | 0.486 | 1.007 | 0.028 (22.8%) | 0.96 | 0.778 | 0.022 |
| Santa Cruz Tenerife | 1905 (30%) | 0.228 (12.1%) | 0.25 | 0.273 | 1.396 | 0.028 (15.3%) | 0.97 | 0.799 | 0.032 |
| Shirahama | 2525 (57%) | 0.564 (22.8%) | 0.35 | 0.212 | 2.108 | 0.029 (47%) | 0.92 | 0.915 | 0.023 |
| Singapore | 504 (91%) | 0.545 (19.4%) | 0.26 | 0.115 | 2.495 | 0.033 (55%) | 0.82 | 1.196 | 0.002 |
| Solar Village | 10537 (77%) | 0.523 (24%) | 0.24 | 0.153 | 1.605 | 0.055 (26.4%) | 0.95 | 0.836 | 0.003 |
| Thessaloniki | 3975 (65%) | 0.374 (13.4%) | 0.61 | 0.325 | 1.859 | 0.019 (35.4%) | 0.93 | 1.058 | 0.006 |
| Tomsk | 283 (38%) | 0.619 (21.8%) | 0.10 | 0.037 | 2.689 | 0.033 (62.2%) | 0.91 | 1.34 | 0 |
| All Sites | 81841 (55%) | 0.5 (20%) | 0.68 | 0.539 | 0.957 | 0.044 (30.5%) | 0.956 | 0.837 | 0.017 |

for the sites with a predominant coarse mode, while lower values are found for the rest of sites (down to 0.6). Similar results are obtained for the analysis of the slopes. All these results may suggest the possibility to add the threshold $\alpha < 1.2$ to assure quality in $C_{Vc}$ retrievals. In such conditions, $\tau(440) > 0.2$ and $\alpha < 1.2$, the RMRSE=25.7% and R=0.95 for a total of 48109 retrievals.

### 3.2.3   Effective radius and total volume concentration

Finally, we will comment on the comparison results obtained between GRASP-AOD and AERONET aerosol retrieval algorithm for the effective radius ($R_{eff}$) and the total volume concentration ($C_{V_T}$). It should be recalled here that neither of the two parameters are primary outputs of the two codes. They are computed from the retrieved values of the bimodal log-normal size distribution for GRASP-AOD and from the 22 bins detailed size distribution for AERONET aerosol retrieval algorithm (more information at http://aeronet.gsfc.nasa.gov/new_web/Documents/Inversion_products_V2.pdf). Therefore, their accuracy
is conditioned by the accuracy of the retrieved parameters.

    The comparison of the effective radius for all the sites can be found at the top panels of Figure 10. As in previous figures, we have imposed three different thresholds for aerosol load at 440 nm, from left to right: $\tau(440) > 0.02$, $\tau(440) > 0.2$, and $\tau(440) > 0.4$. We can see how all the relevant parameters in the comparison improve as the lower limit increases, though, the greatest improvement occurs between the first two thresholds. Thus, the correlation coefficient and the slope are around 0.7
for all points, and they are around 0.8 when $\tau(440) > 0.2$. For the case $\tau(440) > 0.4$, the slope is 0.82 and the correlation coefficient is 0.85. The same applies to the values of RMSE and RMSRE: 0.185 $\mu$m (38%) when $\tau(440) > 0.02$, 0.160 $\mu$m (31%) when $\tau(440) > 0.2$ and 0.151 $\mu$m (29%) when $\tau(440) > 0.4$.

    First part of Table 8 presents the results by sites for the effective radius when $\tau(440) > 0.2$. The correlation coefficients and the slopes are significantly worse for most of the sites than when computing all sites together, with values typically between
0.5-0.7. The larger variation in the effective radius when all sites are analyzed together with respect to performing the analysis one by one is the main reason for this result. Regarding the values of RMSE, we observe that they are the highest for the sites with a coarse mode predominance. At these sites, the differences are between 0.18-0.24 $\mu$m. Coarse mode sites present also the largest differences in relative terms with RMSRE values between 30-40%. On the other hand, the sites with a fine mode predominance present RMSRE values between 20-30%.

Bottom panels of Figure 10 illustrate the comparison for the total volume concentration. The correlation coefficients, the slopes and RMSE are slightly better for the study including all the retrievals (left panel) compare to the other two analyses with $\tau(440) > 0.2$, and $\tau(440) > 0.4$. The only parameter that improves as the lower limit increases is the RMSRE: 29% when $\tau(440) > 0.02$, 25% when $\tau(440) > 0.2$ and 23% when $\tau(440) > 0.4$.

    The analysis by sites with $\tau(440) > 0.2$, shown in the second part of Table 8, exhibits the second best results from the size
parameters analyzed in the present study (just after the characterization of $R_{Vf}$). The correlation coefficients are larger than 0.85 for all the sites, and larger than 0.92 for most of the sites. The slopes are between 0.7 and 1.1, with most of the sites between 0.8 and 1.0. The relative differences do not depend on the aerosol type of the site, with most of the values around 25% ($\pm$5%), which is the averaged value found in the analysis of all the sites together.

**Table 8.** Comparison between effective radius and total concentration obtained from AERONET standard inversion and GRASP-AOD. First column presents the site and second column the number of coincident retrievals accomplishing that $\tau(440) > 0.2$. The percentage with respect to the total number of coincident retrievals is indicated in parentheses. Columns from three to six show the comparison results for effective radius, while columns from seven to ten present the results for the total volume concentration. In both cases, RMSE (and RMSRE enclosed in parentheses) correlation coefficients, slopes and intercepts from linear regressions are shown.

| Sites | N° meas. | $R_{eff}$ | | | | $C_{V_T}$ | | | |
|---|---|---|---|---|---|---|---|---|---|
| | | RMSE | Coeff. -R- | Slope | Intercept | RMSE | Coeff. -R- | Slope | Intercept |
| Alta Floresta | 939 (35%) | 0.047 (20.1%) | 0.74 | 0.782 | 0.038 | 0.034 (22.5%) | 0.92 | 0.838 | 0.023 |
| Arica | 2953 (60%) | 0.153 (33.2%) | 0.41 | 0.322 | 0.355 | 0.018 (18.9%) | 0.85 | 0.808 | 0.014 |
| Banizoumbou | 5809 (81%) | 0.191 (26.4%) | 0.70 | 0.541 | 0.353 | 0.055 (20.1%) | 0.97 | 0.867 | 0.007 |
| Beijing | 3220 (80%) | 0.142 (31.9%) | 0.71 | 0.664 | 0.209 | 0.062 (23.0%) | 0.94 | 1.054 | 0.002 |
| Bonanza Creek | 277 (31%) | 0.085 (33.3%) | 0.63 | 1.108 | 0.029 | 0.041 (32.1%) | 0.91 | 1.103 | 0.009 |
| Cuiaba Miranda | 901 (43%) | 0.065 (25.6%) | 0.72 | 0.48 | 0.102 | 0.03 (20.6%) | 0.93 | 0.779 | 0.027 |
| Capo Verde | 2894 (70%) | 0.183 (22.6%) | 0.55 | 0.341 | 0.547 | 0.033 (13.6%) | 0.97 | 0.939 | 0.007 |
| Dakar | 5715 (88%) | 0.192 (26.5%) | 0.68 | 0.532 | 0.402 | 0.046 (18.0%) | 0.95 | 0.877 | 0.011 |
| Forth Crete | 2043 (52%) | 0.145 (34.8%) | 0.73 | 0.912 | 0.061 | 0.026 (22.4%) | 0.94 | 0.761 | 0.035 |
| GSFC | 2881 (27%) | 0.058 (22.6%) | 0.68 | 0.88 | 0.03 | 0.018 (22.2%) | 0.90 | 0.808 | 0.023 |
| Granada | 2090 (29%) | 0.205 (39.9%) | 0.39 | 0.478 | 0.299 | 0.029 (20.6%) | 0.96 | 0.736 | 0.031 |
| Guadeloup | 249 (26%) | 0.243 (32.6%) | 0.35 | 0.274 | 0.480 | 0.038 (16.5%) | 0.92 | 0.788 | 0.033 |
| Ilorin | 2565 (99%) | 0.201 (37.7%) | 0.56 | 0.431 | 0.389 | 0.09 (22.2%) | 0.95 | 0.806 | 0.042 |
| Ispra | 2317 (58%) | 0.073 (26.2%) | 0.62 | 0.75 | 0.069 | 0.021 (21.0%) | 0.93 | 0.856 | 0.015 |
| Kanpur | 9391 (99%) | 0.139 (27.7%) | 0.75 | 0.619 | 0.201 | 0.071 (26.3%) | 0.94 | 0.723 | 0.055 |
| Lake Argyle | 1642 (22%) | 0.081 (30.1%) | 0.81 | 0.709 | 0.086 | 0.026 (23.5%) | 0.85 | 0.734 | 0.032 |
| Lille | 1392 (47%) | 0.095 (28.4%) | 0.65 | 0.861 | 0.079 | 0.019 (23.2%) | 0.91 | 0.914 | 0.016 |
| Mexico City | 1658 (73%) | 0.076 (25.4%) | 0.58 | 0.697 | 0.096 | 0.027 (27.7%) | 0.85 | 0.736 | 0.012 |
| Moldova | 2854 (50%) | 0.077 (24.8%) | 0.72 | 0.782 | 0.071 | 0.018 (19.5%) | 0.90 | 0.838 | 0.018 |
| Mongu | 2896 (60%) | 0.047 (23.2%) | 0.49 | 0.484 | 0.085 | 0.019 (20.6%) | 0.91 | 0.807 | 0.016 |
| Moscow | 1129 (51%) | 0.077 (25.2%) | 0.68 | 0.689 | 0.117 | 0.021 (21.8%) | 0.91 | 0.859 | 0.015 |
| Reunion - St. Denis | 66 (2%) | 0.083 (24.0%) | 0.52 | 0.462 | 0.211 | 0.013 (18.9%) | 0.85 | 0.712 | 0.033 |
| Sede Boker | 6204 (40%) | 0.19 (30.5%) | 0.54 | 0.637 | 0.208 | 0.03 (20.9%) | 0.96 | 0.753 | 0.029 |
| Santa Cruz Tenerife | 1905 (30%) | 0.236 (33.4%) | 0.41 | 0.461 | 0.489 | 0.036 (17.6%) | 0.97 | 0.761 | 0.036 |
| Shirahama | 2525 (57%) | 0.109 (31.6%) | 0.71 | 0.795 | 0.132 | 0.029 (27.1%) | 0.91 | 0.853 | 0.029 |
| Singapore | 504 (91%) | 0.097 (29.4%) | 0.76 | 0.991 | 0.062 | 0.034 (26.0%) | 0.92 | 0.989 | 0.007 |
| Solar Village | 10537 (77%) | 0.212 (29.8%) | 0.49 | 0.509 | 0.324 | 0.059 (25.7%) | 0.95 | 0.81 | 0.009 |
| Thessaloniki | 3975 (65%) | 0.091 (29.7%) | 0.72 | 0.952 | 0.04 | 0.023 (22.3%) | 0.88 | 0.902 | 0.015 |
| Tomsk | 283 (38%) | 0.088 (31.0%) | 0.65 | 0.683 | 0.099 | 0.031 (30.4%) | 0.93 | 1.083 | -0.005 |
| All Sites | 81841 (55%) | 0.16 (31.2%) | 0.80 | 0.789 | 0.129 | 0.047 (24.8%) | 0.96 | 0.834 | 0.02 |

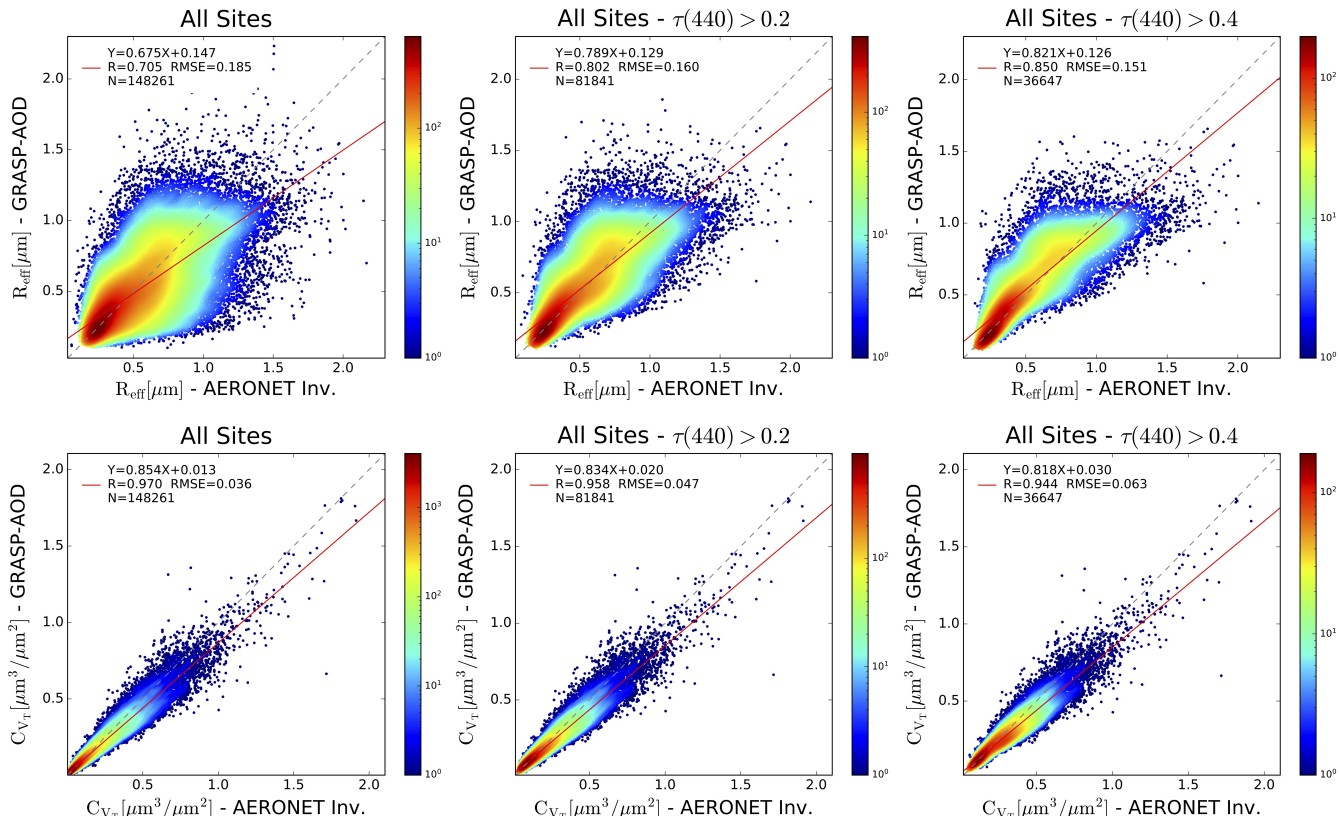

**Figure 10.** Comparisons between the effective radius and the total volume concentration obtained by GRASP-AOD and AERONET aerosol retrieval algorithm for all the sites considered in the analysis (Table 1) for the period 1997-2016. Comparisons for the effective radius ($R_{eff}[\mu m]$) are represented at the top panels, while comparisons for the coarse mode volume concentration ($C_{V_T}[\mu m^3/\mu m^2]$) are shown at the bottom panels. Different thresholds for $\tau(440)$ have been applied in the comparisons, from left to right: all the retrievals ($\tau(440) > 0.02$), retrievals with $\tau(440) > 0.2$, and retrievals with $\tau(440) > 0.4$. Color bars represent data density in a $0.02 \times 0.02$ $\mu m$ grid for $R_{eff}$ and in $0.01 \times 0.01 \mu m^3/\mu m^2$ grid for $C_{V_T}$. For the both parameters, a logarithmic scale has been chosen given the strong data density at low values.

As mentioned in the introduction, the study by Pérez-Ramírez et al. (2015) (based on linear estimation techniques (LET) described by Veselovskii et al., 2012) proposed to derived the effective radius and the total volume aerosol concentration from only spectral $\tau$ measurements. In the same work, the authors proposed a validation study using AERONET $\tau$ measurements as input from 18 sites during one year (around 75.000 $\tau$ measurements). Afterwards, they compared LET retrievals of $R_{eff}$ and $C_{V_T}$ to the coincident values obtained by AERONET aerosol retrieval algorithm, similarly as here. The characterization obtained for effective radius is comparable to the one obtained here, with relative errors respect to AERONET around 30% in both cases. On the other hand, the characterization of the total volume concentration computed for GRASP-AOD agrees better with AERONET aerosol retrieval algorithm (25% RMSRE) compared to LET retrievals (40% relative differences).

## 4    Discussion

### 4.1    Bimodal assumption and three mode size distributions

During the analysis of Table 3 in sub-section 2.2 we indicated that all sites except Ilorin presented more than $85\%$ of GRASP-AOD valid retrievals with respect to the total number of $\tau$ measurements. The relatively small number of valid GRASP-AOD retrievals at Ilorin site ($76\%$) is under analysis in this section. Particularly, we are confident that the main reason of the low valid retrievals is related to the bimodal log-normal assumption regarding the size distribution. This assumption, which is one of the main bases of the GRASP-AOD application, would not be true for many aerosol retrievals found at Ilorin site, which originates a high residual fitting in those retrievals.

### 4.1.1    Low data percentage at Ilorin site

The study by Eck et al. (2010) pointed out that a midsize aerosol mode at $0.6$ $\mu$m was recurrently present in the dust and mixed fine/coarse mode aerosol retrievals at Ilorin. The origin of this mode is related to the desert dust from Bodélé Depression of central Chad (in the southern Saharan desert) typically transported over Ilorin during the winter/spring period (Washington et al., 2006). The dust from the Bodélé Depression, which is a unique source for aerosols and it is sometimes described as the single largest individual desert dust source on Earth, was deeply analyzed during the Bodélé Dust Experiment (BoDEx) in 2005. The study from Todd et al. (2007) showed that the dust consists predominantly of fragments of diatomite sediment. The particle size distribution of this diatomite dust estimated from AERONET aerosol retrievals indicated a dominant coarse mode (radius centered on 1–2 $\mu$m) similar to other Saharan dust observations. However, they observed also a minor but noticeable presence of particles with radii < 1 $\mu$m, which is unusual for desert dust, that gives rise to the aforementioned midsize mode.

It is precisely this midsize mode the origin of the high percentage of retrievals at Ilorin site that do not pass the criteria of the GRASP-AOD application. To support this idea, we illustrate at Figure 11a the average of the normalized size distributions (normalization done by the maximum value) retrieved by the AERONET aerosol algorithm at Ilorin site for the whole analyzed period (in the case of Ilorin between 1998-2016). The retrievals have been divided in two groups depending on whether the coincident GRASP-AOD retrievals (at least one in the 32 minute interval around each almucantar measurement defined at subsection 2.4) meet the quality criteria defined at subsection 2.2: gray dashed line when the coincident GRASP-AOD retrievals pass the quality criteria (2594 inversions), black solid line for the cases when the coincident GRASP-AOD retrievals do not pass the quality criteria (1014 inversions). On the one hand, we observe a clearly defined bimodal structure for the size distributions with coincident GRASP-AOD valid retrievals. On the other hand, a third mode centered at $0.6$ $\mu$m appears in the average of the size distributions without a corresponding GRASP-AOD valid retrieval.

The averages shown at Figure 11a represent tendencies in the two types of retrievals. However, the GRASP-AOD filter criteria can not be considered as a perfect detector of three mode structures. In fact, analyzing one by one the size distribution retrieved by AERONET, there are several with a noticeable third mode and with a corresponding GRASP-AOD valid retrieval. At the same time, there are some perfectly bimodal AERONET size distributions without a valid GRASP-AOD retrieval. Nevertheless, it should be noted here that the cross section of extinction at $0.6$ $\mu$m (or kernels for the extinction, see Equation

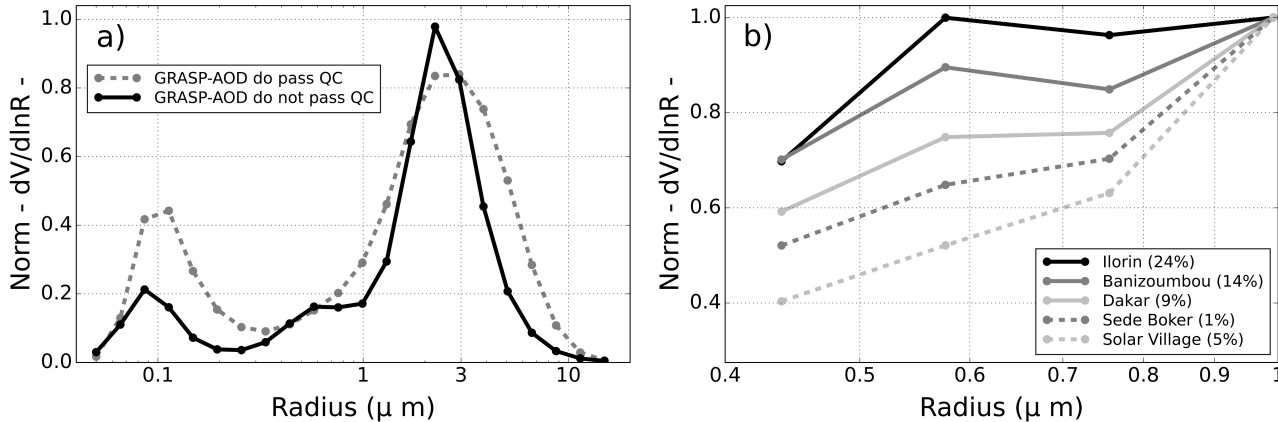

**Figure 11.** a) Average of normalized (by maximum value) size distributions retrieved by AERONET aerosol algorithm at Ilorin site. They have been divided in two groups depending if the coincident GRASP-AOD retrievals meet the quality criteria defined at subsection 2.2: in gray dashed line when the coincident GRASP-AOD retrievals pass the quality criteria (2594 inversions), black solid line for the cases when the coincident GRASP-AOD retrievals do not pass the quality criteria (1014 inversions). b) Averages of normalized (by the value at $0.992\ \mu m$) size distributions at several AERONET sites when the coincident GRASP-AOD retrievals do not meet the quality criteria: Ilorin (black solid line), Banizoumbou (grey solid line), Dakar (silver solid line), Sede Boker (grey dashed line) and Solar Village (silver dashed line). Only the interval with radii from $0.439\ \mu m$ to $0.992\ \mu m$ is plotted.

2 and 3 of Torres et al. (2017)) is quite high for all the wavelengths considered at the present study. The fact of neglecting the third mode (since a bimodal structure is assumed) is a significant source of error in the estimation of the spectral aerosol optical depth. Indeed, errors associated to a deficient aerosol model can be treated as other error sources (for instance the intrinsic to the measurements, see Dubovik (2004)) to estimate the uncertainty of the retrieval. Therefore, the recurrently third mode structure at Ilorin produces a systematic error that affects the retrieval fitting or residual for GRASP-AOD. It may not

be determinant but is added to the rest of the errors. The fact that some of the quality filters used for GRASP-AOD retrievals refers to the fitting or residual (specifically the last two at subsection 2.2) justifies that at Ilorin site the percentage of valid GRASP-AOD retrievals is the lowest.

To put the results at Ilorin in perspective, Figure 11b analyzes the averages of the normalized size distributions when the coincident GRASP-AOD retrievals do not meet the filtering criteria, at several AERONET sites with a desert dust predomi-

nance: Ilorin (black solid line), Banizoumbou (grey solid line), Dakar (silver solid line), Sede Boker (grey dashed line) and Solar Village (silver dashed line). Since we are interested in the presence of the third mode at $0.6\ \mu m$, the size distributions are normalized to the value at $0.992\ \mu m$, and in the figure only the section of radii from $0.439\ \mu m$ to $0.992\ \mu m$ is plotted. We observe that Ilorin has the highest values for the size distribution at $0.576\ \mu m$ and $0.756\ \mu m$, with both values similar to one (i.e. to the size distribution value at $0.992\ \mu m$). The other two Sub-Saharan sites present also high values at $0.576\ \mu m$ and

$0.756\ \mu m$, specially at Banizoumbou, though significantly lower than at Ilorin. The two Middle East sites present the lowest value at $0.576\ \mu m$ and $0.756\ \mu m$, with the size distributions perfectly decreasing from $0.992\ \mu m$ to lower radii.

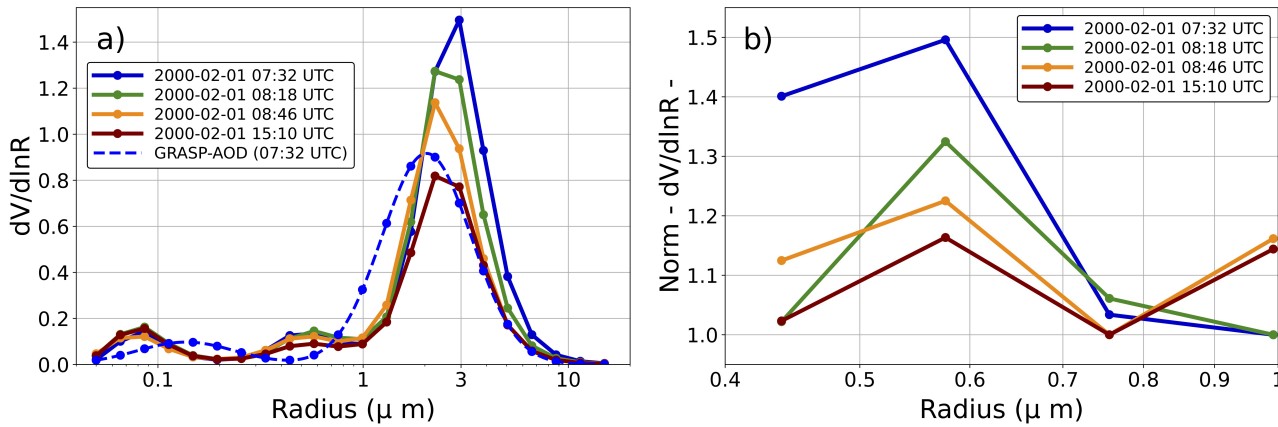

**Figure 12.** a) Volume size distributions retrieved by AERONET aerosol algorithm at Ilorin site on 1 February 2000. The only valid retrieval from GRASP-AOD, which corresponds to the $\tau$ measurement at 07:32 UTC is also plotted with a dash blue line. b) The same volume size distributions retrieved by AERONET but only from $0.439\ \mu$m to $0.992\ \mu$m (cutoff radius range used to separate modes in AERONET) and normalized by the minimum value at this interval

In fact, at Ilorin site the recurrently third mode is reported from climatologies, but this third mode is not present at climatologies of the other four dust affected sites analyzed here (Dubovik et al., 2002a; Eck et al., 2008). Note at this point that we have only averaged the size distributions without a valid GRASP-AOD retrieval, and the percentage of the no-valid retrieval is shown in the legend for each site. So even if at Banizoumbou or Dakar we can observe an incipient third mode, the size distributions illustrated here only represent the $14\%$ and the $9\%$ of the retrievals, while at Ilorin they represent the $24\%$. It should be also highlighted that the percentage of GRASP-AOD retrievals that do not meet the criteria at Sede Boker and Solar Village is $1\%$ and $5\%$, respectively. These values are on the same order as those found at the sites with a predominant fine mode.

### 4.1.2   Large discrepancies in the estimation of $\tau_f(500)$

We suggested at the end of subsection 3.1 that the three mode structures analyzed at this section were the reason of the second branch observed at Figure 5 while comparing $\tau_f(500)$ retrieved by AERONET aerosol retrieval algorithm versus GRASP-AOD and SDA (when $\alpha > 0.6$). Thus, for some retrievals there is a significant overestimation of the $\tau_f(500)$ from AERONET aerosol retrieval algorithm compared to the other two methods. We have observed that most of these retrievals are at Ilorin site (though a few examples can be found at Banizoumbou and Dakar) and for all of them the midsize mode is relatively high. This makes that the minimum value of the volume size distribution in the cutoff radius range (from $0.439\ \mu$m to $0.992\ \mu$m) is found either at $0.992\ \mu$m or at $0.756\ \mu$m. That is why the AERONET retrieval assigns the midsize mode completely (or mostly) to the fine mode while the other two methods do not. This fact creates the aforementioned overestimation.

**Table 9.** Values of $\tau_f(500)$ retrieved at Ilorin site on 1 February 2000. The first two columns contain the values obtained by AERONET aerosol retrieval algorithm and SDA. The rest of the columns correspond to simulated $\tau_f(500)$ values estimated by GRASP forward code using as input the aerosol properties retrieved by AERONET and considering different cutoffs.

| Date - Time | $\tau_f(500)$ | | Simulated $\tau_f(500)$ at different cutoffs (GRASP forward code) | | | | | |
| --- | --- | --- | --- | --- | --- | --- | --- | --- |
| | AERONET std. | SDA | - 0.992 $\mu$m - | - 0.756 $\mu$m - | - 0.576 $\mu$m - | - 0.439 $\mu$m - | - 0.335 $\mu$m - | - 0.255 $\mu$m - |
| 01/02/2000 07:32 | 1.005 | 0.599 | 1.009 | 0.957 | 0.892 | 0.751 | 0.533 | 0.387 |
| 01/02/2000 08:18 | 0.991 | 0.590 | 0.992 | 0.933 | 0.851 | 0.685 | 0.485 | 0.366 |
| 01/02/2000 08:46 | 0.904 | 0.565 | 0.965 | 0.907 | 0.839 | 0.721 | 0.532 | 0.385 |
| 01/02/2000 15:10 | 0.785 | 0.523 | 0.831 | 0.787 | 0.735 | 0.645 | 0.508 | 0.401 |

To present an example of these retrievals, left panel of Figure 12 illustrates with solid lines the four aerosol volume size distributions retrieved by AERONET (Level 2) at Ilorin site on 1 February 2000. Right panel of Figure 12 contains these same four size distributions but only from $0.439\ \mu$m to $0.992\ \mu$m (cutoff radius range) and normalized by the minimum value at this interval. We can see that the first two size distributions (07:32 UTC and 08:18 UTC plotted with blue and green line respectively) present their minimum value at $0.992\ \mu$m while the other two (08:46 UTC and 15:10 UTC plotted with orange and red line respectively) at $0.756\ \mu$m.

The values of $\tau_f(500)$ estimated by AERONET aerosol retrieval algorithm and SDA are shown in Table 9. The first two retrievals (with AERONET cutoff at $0.992\ \mu$m) present differences around 0.4 between the two methods. The last two (with AERONET cutoff at $0.756\ \mu$m) present differences of 0.34 and 0.26. To better interpret these differences, second part of Table 9 depicts $\tau_f(500)$ values estimated by GRASP forward code using as input the aerosol properties retrieved by AERONET considering different cutoffs. We note that the values obtained at $0.992\ \mu$m for the first two retrievals and at $0.756\ \mu$m for the last two are almost identical to those given by AERONET aerosol retrieval. This result was expected since the radiative transfer codes used by GRASP and AERONET are quite similar and the cutoff established by AERONET correspond to these radii. We observe that to find the values retrieved by SDA the cutoff should be established between $0.335\ \mu$m to $0.439\ \mu$m for the four retrievals. Nevertheless, the mode separation in three-mode structures is not immediate neither for the SDA algorithm nor for GRASP-AOD, which base their functioning on the existence of two well-defined aerosol modes. Therefore, the values obtained by these two algorithms can not be considered as the truth in these circumstances. Actually, if the midsize mode, which is originated by dust particles, was completely assigned to the coarse mode (see values for cutoff at $0.255\ \mu$m) the values of $\tau_f(500)$ would be much smaller than the values obtained by SDA.

Finally, we would like to point out that there is only one GRASP-AOD valid retrieval from the twenty-five $\tau$ measurements in AERONET level 2 available at that day. The valid retrieval corresponds to the first $\tau$ measurement in the morning which was taken just before the almucantar at 07:32 UTC. The corresponding size distribution retrieved by GRASP-AOD has been represented with a dashed blue line in Figure 12a. It can be observed how the midsize mode is not detected by GRASP-AOD

since a bimodal size distribution is imposed. As commented before, the midsize mode is situated in a radius range with relative high values of the extinction cross sections and presents also a quite defined spectral dependency. Therefore, neglecting this mode can not be correctly compensated by adding extra particles in the two assumed modes. As a matter of fact, the valid retrieval has an absolute residual fitting of $0.029$ (relative $1.7\%$) which is quite high compared to typical residual fitting values that are lower than $0.01$. The other twenty-four retrievals presented residual fittings higher than $0.04\%$ and they did not fulfill the filtering criteria defined at subsection 2.2 (point 4 though most of them did not fulfill neither the point 5). The $\tau_f(500)$ obtained by the GRASP-AOD valid retrieval was $0.642$ which is much closer to the value given by SDA than the one obtained by AERONET aerosol retrieval.

## 4.2  Use of standard refractive index values

As largely described in Torres et al. (2017), the information contained exclusively in the spectral aerosol optical depth measurements is not enough to retrieve the aerosol refractive indices. Consequently, this parameter needs to be assumed to run GRASP-AOD application. The use of monthly climatological values has been proposed in the present validation study as explained in subsection 2.2. These values have been obtained by averaging the retrievals of AERONET aerosol standard algorithm (which includes full sky radiances and $\tau$ measurements) available for each site at AERONET webpage (https://aeronet.gsfc.nasa.gov/cgi-bin/webtool_inv_v3). It should be noted here, that all the sites chosen for this study were selected based on the availability of an extensive data record of at least ten years of $\tau$ measurements in AERONET website. Although, this requirement was primarily settled to have a large number of $\tau$ measurements at each site, our strategy regarding the refractive index has certainly benefited from this fact. Thus, all the analyzed sites counted on robust datasets of aerosol optical properties, which have been used to generate the refractive indices for the GRASP-AOD retrievals (more details at subsection 2.2).

At this point, we wish to discuss the methodology to run GRASP-AOD in new sites or in sites with only few years of existing data. A reasonable strategy would be the use of standard refractive index values of the dominant aerosol type expected at the new site. For example, in sites with a predominant fine mode (mainly urban sites) the values suggested would be around $1.45$ - $0.005i$ (spectrally independent). In sites with frequent desert dust episodes the real part would be higher (up to $1.56$) and the imaginary part would count for a larger absorption in UV channels. It should be noted that we do not intend here to detail the rules about how the election of refractive index values for new sites should be made. Our proposes are just to underline that in new sites assumptions would need to be made, and to estimate the impact that these assumptions may have on the retrieved parameters by comparing these results to the optimum case when a robust dataset is available.

Before beginning the description of the tests performed in this study, we would like to briefly summarize the main outcomes obtained in previous works concerning this topic, to avoid repeating previous analysis. Thus, one of the main results derived from the study by Torres et al. (2017) was that the retrieved parameters by GRASP-AOD application were much more sensitive to a variation in real refractive index than to a variation in the imaginary part. The same result was also found by earlier studies of only aerosol optical depth retrievals such as King et al. (1978) and Yamamoto and Tanaka (1969). Moreover, from all the retrieved parameters, the mean radius and the volume concentration of the fine mode were the most affected by a variation in

the real part of the refractive index. This result is illustrated in Figure 9 of the study by Torres et al. (2017). The figure shows a decrease in the mean radius and the volume concentration of the modes when the real part of the refractive index increases, while for negative variations both parameters increase their values. A similar result was previously obtained in King et al. (1978), where it was pointed out that the shape of the size distribution remains the same but shifts with a varying real part.

Both results are derived from the anomalous diffraction theory by Van de Hulst (Van de Hulst, 1957). The other main interesting result from Torres et al. (2017) was that the separation fine/coarse of the aerosol optical depth as well as the characterization of the coarse mode size parameters were practically unaffected by variations of the refractive index.

Considering these previous results, we have reprocessed the data from Mongu site (same $\tau$ measurements as described in Table 1), though instead of using the climatological values as in the general analysis of section 3, we have assumed standard

values of $1.45 - 0.005i$. Mongu site has been chosen principally for two reasons: a) it has one of the largest dataset from all the predominant fine mode sites for the period 1997-2016. The interest here for fine mode dominant sites is due to the fact that we expect $R_{Vf}$ and $C_{Vf}$ to be the most affected parameters by the variation of the refractive index. At the same time the characterization of $R_{Vf}$ showed the lowest differences with AERONET from all the size volume aerosol parameters analyzed in the subsection 3.2, specially if $\tau(440) > 0.2$ and $\alpha > 1.2$. Therefore, the main analysis of this section would be focus on

describe how the characterization of $R_{Vf}$ is affected by the election of standard refractive index in the aforementioned best retrieval conditions. b) The monthly climatological refractive index values at Mongu are around $1.51 - 0.021i$ (similar to the results found at Dubovik et al. (2002a)). These values are one of the most different compared to the standard values proposed here ($1.45 - 0.005i$), from all the sites with a predominant fine mode. Actually, the monthly climatological averages found at several sites (to cite some GSFC, Ispra, Lille or Shirahama) are quite close to $1.45 - 0.005i$, and logically, the assumption of

this so-called standard value would have a little impact on the retrievals.

Figure 13 illustrates the comparisons of fine mode volume median radius ($R_{Vf}[\mu m]$, at left panels), fine mode volume concentration ($C_{Vf}[\mu m^3/\mu m^2]$, at central panels), and fine mode optical depth ($\tau_f(500)$, at right panels) retrieved by GRASP code and AERONET aerosol retrieval algorithm for Mongu site during the whole analyzed period (measurements from 1997-2010 in the case of Mongu site). At top panels, GRASP-AOD retrievals have been processed assuming a standard values for the

refractive index of $1.45 - 0.005i$, while at bottom panels, the retrievals correspond to the general processing (section3) where monthly climatological values were used. The first thing that we can observe is that the number of data in the comparisons, both for size parameters and $\tau_f(500)$, is almost the same regardless the refractive index used. This is due to the fact that the number of $\tau$ retrievals that passes the quality criteria with standard values of refractive index (90129) is almost the same that the one obtained with climatological values (90005).

Concerning the retrieval of the size parameters, we observe that GRASP-AOD retrievals of $R_{Vf}$ and $C_{Vf}$ show larger values with the use of standard refractive index values. This result was expected, since as previously pointed out, for negative variations of the real refractive index the fine mode median radius and the volume concentration increase their values (Figure 9 of Torres et al. (2017)). Thus, the mean value of GRASP-AOD retrievals of fine mode median radius ($< R_{Vf} >$) at Figure 13 is 0.131 $\mu m$ when refractive index from climatological values are used, while $< R_{Vf} >$ is 0.151 $\mu m$ when the standard values are used.

Note here that Torres et al. (2017) pointed out that the variation of $R_{Vf}$ due to a variation of real refractive index ($\Delta n$)

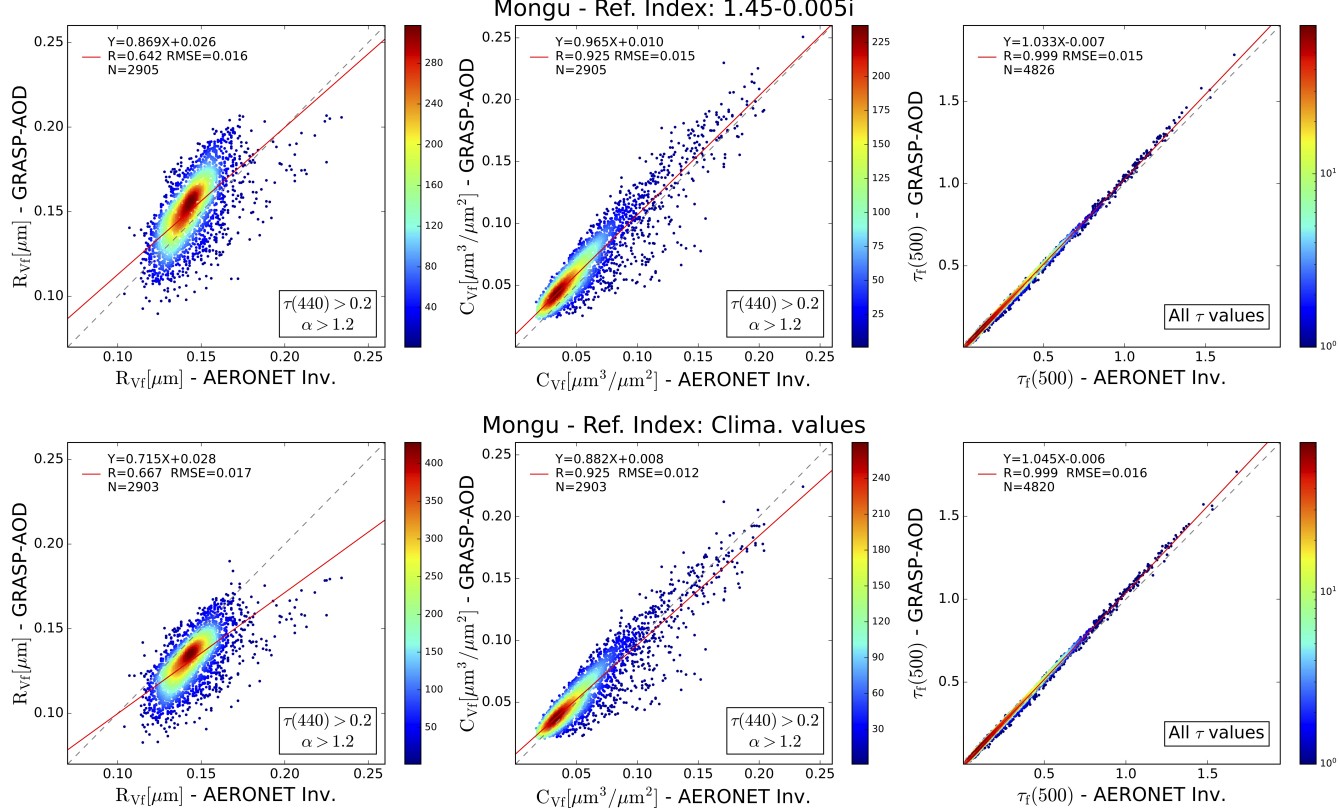

**Figure 13.** Comparisons between the fine mode volume median radius ($R_{Vf}[\mu m]$, at left panels), the fine mode volume concentration ($C_{Vf}[\mu m^3/\mu m^2]$, at central panels) and $\tau_f(500)$ (at right panels) obtained by GRASP code and AERONET aerosol retrieval algorithm for Mongu site during the whole analyzed period (measurements from 1997-2010 in the case of Mongu site). In retrievals at the top panels, we have assumed standard values of the refractive index (1.45 - 0.005$i$) to run GRASP-AOD . The retrievals at the bottom panel corresponds to the results of general analysis from section 3, where monthly climatological values were used for the assumption of the refractive index.

do not depend on the aerosol load, and that this variation could by roughly approximated by $\Delta R_{Vf} \sim -0.4 \times \Delta n$. In these regards, the variation of 0.02 $\mu m$ (14% in relative terms) obtained here fits this estimate since the $\Delta n$ (standard values minus climatological values) is in average $-0.06$. On the other hand, the $< R_{Vf} >$ obtained by AERONET retrievals corresponds to 0.143 $\mu m$, which is an intermediate value between the two averages obtained by the two GRASP-AOD processings. This

fact justifies that the RMSE of $R_{Vf}$ comparison with AERONET retrieval does not vary when using standard values for the refractive index (0.016-0.017 $\mu m$ or between 10-12% in relative terms). This is a particular result for Mongu processings, though overall, we might expect the RMSE with respect to AERONET retrievals be affected to some extent. Other correlation parameters for $R_{Vf}$ comparisons present some variations but they are not quite significant either. For instance, the correlation coefficient is a bit better with the use of climatological values (0.67 versus 0.64) though the slope is a bit worse (0.72 versus

780 0.87).

We observe similar patterns for the fine volume concentration comparisons. The mean values of GRASP-AOD retrievals of fine mode volume concentration ($< C_{Vf} >$) are 0.059 with the use of climatological values and 0.066 with standard values of the refractive index. This variation of -0.007 fits into the approximation given by Torres et al. (2017) for the fine mode concentration, $\Delta C_{Vf} \sim -0.27 \times \Delta n \times \tau(440)$, which would foresee a variation of $-0.008$ considering that $< \tau(440) > = 0.499$. In this case, $< C_{Vf} >$ of AERONET retrievals is 0.056 which may justify that the RMSE is better, 0.012 (20%) versus 0.015 (25%), when climatological values are used. On the other hand, the correlation coefficients are the same for both processings and the slope improves (0.97 versus 0.88) with the use of standard values for the refractive index.

Finally, right panels at Figure 13 represent the comparisons of $\tau_f(500)$ for the two reprocessings with respect to AERONET. As commented before, one of the main conclusions from Torres et al. (2017) established that the characterization of this parameter was independent from the assumption of the refractive index. This result is confirmed by the analysis of the correlation parameters of both subfigures. We observe that the correlation coefficients, slopes and RMSE do present negligible differences between the two processings. The average values ($< \tau_f(500) >$) are also very close: 0.259 obtained with climatological values and 0.262 with the proposed standard values (relative difference of 1%). For AERONET retrievals $< \tau_f(500) >$ is 0.257 which is also similar to both GRASP-AOD averaged values.

## 4.3 Advanced characterization of the aerosol coarse mode

The low sensitivity to the coarse mode properties of $\tau$ measurements in the spectral range between 340-1020 nm was one of the main conclusions in the study by Torres et al. (2017). In this respect, an optimal selection of the initial guess was pointed out as a key factor to improve the characterization of coarse mode. These results have been confirmed throughout the Subsection 3.2.2 of this work. Specifically, we have indicated that GRASP-AOD was not sensitive to the small oscillations of coarse mode median volume radius occurred for individual aerosol types at the different sites, though we have obtained reasonable characterization overall. This has been possible mainly due to an optimal election of the initial guess. Nevertheless, there is still a wide scope for improving this choice. For instance, the use of climatological values by sites, or even the use of the retrieved value from the nearest AERONET aerosol retrieval, will be certainly attempted in future reprocessings. Note that the latter approach would probably show the best results. Moreover, it would improve not only the characterization of coarse mode but also the effective radius and the total volume concentration, specially in those sites with a predominant coarse mode. However, the current study is limited to GRASP-AOD retrievals with a close AERONET aerosol retrieval whose products are used for validation proposes. The use of the AERONET aerosol retrieval at the same time as initial guess and to validate the GRASP-AOD retrievals would show an excellent characterization that might be biased from the real performance in a global processing.

Another idea to improve the characterization of the coarse mode would be to complement the $\tau$ measurements with aureole measurements. In fact, the angular distribution of scattered light is known to be strongly dependent on the coarse mode particles, specially at the aureole region ($\Theta$=3-10°, see for instance Tonna et al., 1995). The main interest of this approach would be to obtain better aerosol information in the common situations of partial cloudiness. In such occasions, sky-radiance measurements are not suitable for the retrieval of detailed aerosol properties (from almucantar or hybrid scenarios). However, the sky region

around the Sun could be cloud-free, which would allow us to use as input the available $\tau$ and aureole measurements. It should be noted that aureole measurements do not provide the necessary information to retrieve the aerosol optical properties. In this regard, the refractive index values, which are necessary to run GRASP-AOD, would continue to be taken from the site's climatologies. The contribution of aureole measurements is restricted to improve the coarse mode characterization and the derived products such as effective radius or total volume concentration.

To check this idea, we have selected the aureole measurements, between $3.5°$ to $10°$ azimuth angle, belonging to existing al-mucantar measurements at Granada site from the period 2011-2012. These aureole measurements are at only four wavelengths: 440, 670, 870 and 1020 nm. We have run GRASP code adding these aureole measurements to the coincident $\tau$ measurements (7 wavelengths in the range 340-1020 nm), using the same configuration as in GRASP-AOD application (bimodal log-normal size distribution, refractive index pre-fixed, etc.). To distinguish this new use from the classic GRASP-AOD application from

now on we will refer to it as GRASP-AUR. Some comparisons between the aerosol properties obtained by GRASP-AUR and those obtained by AERONET aerosol retrieval algorithm (with the full almucantar) are presented at Figure 14. From left to right: coarse mode volume median radius ($R_{Vc}[\mu m]$) , effective radius ($R_{eff}[\mu m]$) and the total volume concentration ($C_{V_T}[\mu m^3/\mu m^2]$). To put these results in perspective, at bottom panels we present the comparison in the same period at Granada site for the GRASP-AOD retrievals (only $\tau$ measurements in the input) and AERONET aerosol retrieval algorithm. In

all the comparisons, we have selected the data with $\tau(440) > 0.2$ since this was the threshold identified in section 3 to assure the quality in the retrievals.

The first thing we observe in Figure 14 is that there are less common retrievals between AERONET aerosol retrieval algorithm and GRASP-AUR (when we add the aureole measurements) than between AERONET and GRASP-AOD (without the aureole measurements). This small discrepancy in the common retrievals (16 retrievals out of almost 500) is due to the general

increase in the retrieval fitting of $\tau$ measurements when we consider the aureole measurements[6]. The lost of several valid retrievals of GRASP-AUR with respect to GRASP-AOD is justified since we have kept the same quality criteria regarding the fitting of $\tau$ measurements (defined at subsection 2.2).

If we analyse first the results for $R_{Vc}$ (left panels), we observe that all the parameters in the comparison with AERONET aerosol retrieval algorithm are improved when the aureole measurements are added. Thus, the correlation coefficient passes

from 0.75 to 0.91 with aureole measurements. The slope rises from 0.66 to 1.1 and the intercept is reduced from 0.54 to only 0.04. The RMSE also decreases strongly from 0.438 $\mu m$ to 0.299 $\mu m$, which in relative terms means a reduction from 19% to 13%. Visually, we observe a continuous correlation for all radii, beyond the two clusters obtained for GRASP-AOD: the overall reasonable characterization obtained by the smart election of initial guess evolves to an excellent correlation of $R_{Vc}$ when aureole measurements are added.

The characterization of the derived properties, total volume concentration and effective radius, also improves when adding aureole measurements. Thus, the slopes passes from 0.6-0.7 to be around 1 in both characterizations. In the case of $C_{V_T}$, the rest of parameters improve even thought some of them were already excellent. For instance, the correlation coefficient passes

---

[6]This result is logical from the retrieval point of view. Generally, the fact of adding new measures, which should also be fitted, degrades the fitting of existing measurements.

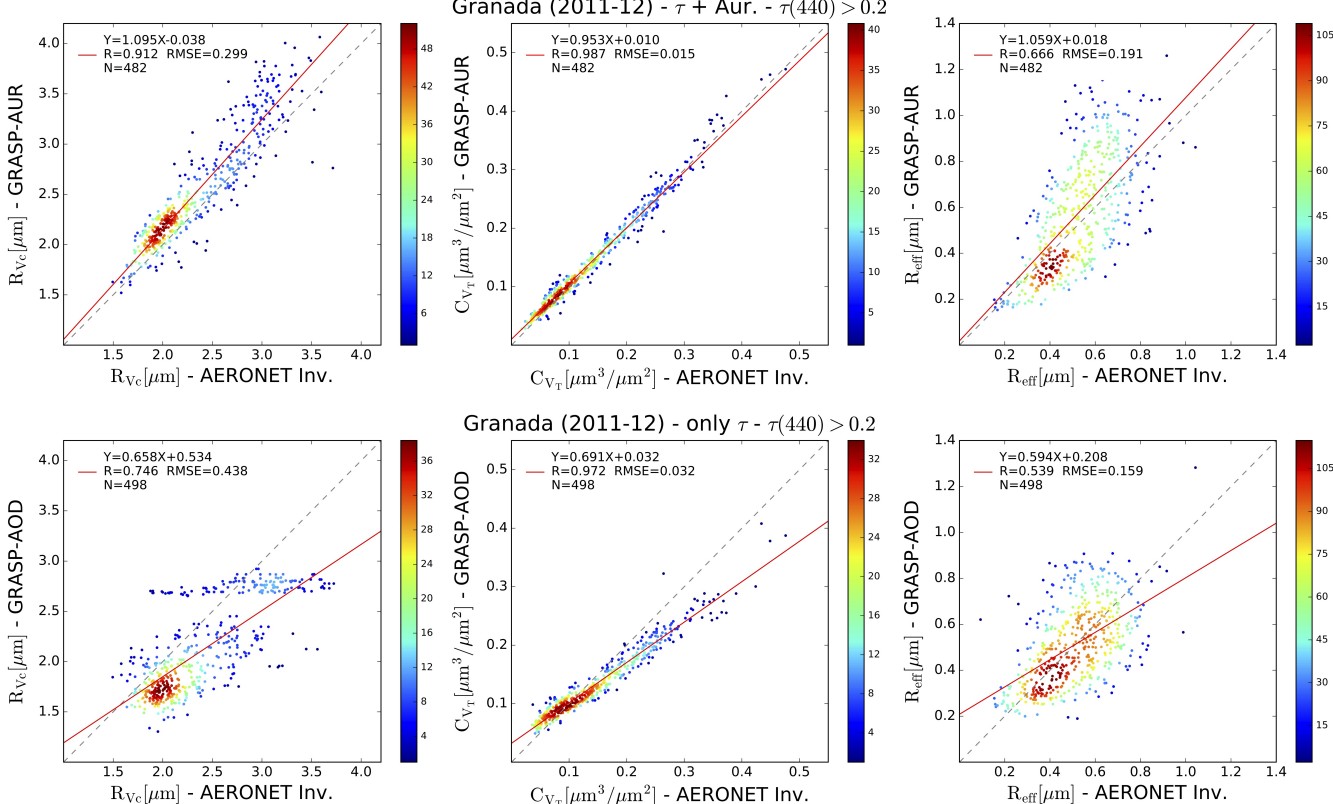

**Figure 14.** Comparisons between the coarse mode volume median radius ($R_{Vc}[\mu m]$, at left panels), the total volume concentration ($C_{V_T}[\mu m^3/\mu m^2]$, at central panels) and effective radius ($R_{eff}[\mu m]$, at right panels) obtained by GRASP code and AERONET aerosol retrieval algorithm for Granada site for the biennium 2011-2012. In retrievals at the top panels, we have used $\tau$ measurements and aureole measurements between $3.5°$ to $10°$ of azimuth angle as input (GRASP-AUR application). The retrievals at the bottom panel only contain $\tau$ measurements as input (GRASP-AOD retrievals). In all the retrievals, we have added the filter $\tau(440) > 0.2$. Color bars represent data density in a $0.2 \times 0.2\ \mu m$ grid for $R_{Vc}$ and $R_{eff}$ and in $0.02 \times 0.02\mu m^3/\mu m^2$ grid for $C_{V_T}$.

from 0.97 to 0.99 and the RMSE is divided by two from $0.032\mu m^3/\mu m^2$ (22.7%, in relative terms) to $0.015\mu m^3/\mu m^2$ (10.6%). For $R_{eff}$, we observe a better correlation when we introduce the aureole measurements, however, the RMSE increases from 0.159 $\mu m$ to 0.191 $\mu m$, (or from 32.1% to 37.5% in relative terms). This increase is justified by the general overestimation of $R_{Vc}$, and specially at largest values ($3.5 - 4\ \mu m$).

It should be noted that a comprehensive study on the use of aureole measures to improve the GRASP-AOD application is outside the scope of the present validation study. These first results presented here indicate the direction that may be taken in future analysis, specially to improve the characterization of the coarse mode. Finally, we would like to highlight the versatility of GRASP code that allows easily to integrate the aureole measurements in a predefined inversion scheme such as GRASP-AOD. Moreover, the GRASP multi-pixel approach, which is successfully used in satellite retrievals (Dubovik et al., 2014; Chen

et al., 2020), will be certainly explored in future tests. Although some preliminary promising results have been obtained (not presented here), the choice of optimal constraints and a detailed analysis of the benefits (or disadvantages) of its use deserves to be the main subject of further studies.

 ## 5   Conclusions

The work presented here aimed to compliment the study of Torres et al. (2017) by the demonstration of the applicability of GRASP-AOD approach to large datasets aerosol optical depth observations. In these regards, the study has proposed a real data validation based on 2.8 million GRASP-AOD retrievals using AERONET aerosol optical depth observations from 30 AERONET sites for 20 years (1997-2016) in the range between 340-1020 nm. The study has also provided (see subsection 2.2) several recommendations for applying GRASP-AOD in operational processings. Specifically, we have proposed the assumption of climatological values for the refractive index in the retrieval and the criteria for the quality control of the results.

The validation study has had two main points. First, the values of $\tau_f(500)$ obtained by GRASP-AOD have been compared with the results obtained by AERONET aerosol retrieval algorithm and SDA retrievals. We have also compared the retrievals between AERONET aerosol retrieval algorithm and SDA. Second, the results of aerosol size parameter retrieval by GRASP-AOD have been compared with AERONET aerosol retrieval algorithm.

The analysis of $\tau_f(500)$ has shown the robustness of the GRASP-AOD algorithm to discriminate between fine and coarse mode extinction with a performance comparable to that obtained by SDA. The comparison, with more than 2 millions of common retrievals between both methods, has shown a correlation coefficient of 0.997, a slope of 0.985, and an intercept of 0.006. The RMSE found was equal to 0.015 or $10\%$ in relative terms. The comparisons of SDA and GRASP-AOD results with those obtained by AERONET aerosol retrieval algorithm showed similar tendencies. The comparisons showed an excellent agreement for sites dominated by the fine mode aerosol with the slopes and correlation coefficient very close to 1. For those sites, the RMSRE among the three retrievals is between $5-10\%$. Larger discrepancies for the retrievals of $\tau_f(500)$ appeared for sites with a predominant coarse mode, typically ranging between 10 and $30\%$. Moreover, filtering the retrievals by $\alpha<0.6$, we have found that values of $\tau_f(500)$ estimated from AERONET aerosol retrievals are higher on average than from SDA and GRASP-AOD, as observed in previous studies (see Eck et al. (2010) and Torres et al. (2017)). This result is mainly related to the cutoff process used to define the two modes in AERONET aerosol retrieval algorithm as explained by O'Neill et al. (2003), or at this study in subsection 2.3.2. Furthermore, we have observe a second branch for comparisons of GRASP-AOD versus AERONET aerosol retrieval algorithm and SDA versus AERONET aerosol retrieval algorithm when $\alpha<0.6$, where values of AERONET aerosol retrieval algorithm are much higher than for the other two retrievals. The reason has been associated to the three mode structures observed in some desert dust retrievals at Sub-Saharan sites as described in subsection 4.1.2.

The validation of the aerosol size parameters has been carried out through the comparison of almost 150 thousand common retrievals from GRASP-AOD and AERONET aerosol algorithm. The analysis has confirmed the good capacity of GRASP-AOD to accurately characterize the aerosol fine mode size properties as indicated earlier by Torres et al. (2017). The utilization a lower limit of $\tau(440)>0.2$ was suggested for GRASP-AOD application to assure the quality in the retrievals at Torres et al.

(2017) and this limit has been confirmed here. A higher limit of $\tau(440)>0.4$ hardly improved the results obtained, while it erased an enormous number of data. In agreement with study by Torres et al. (2017), the characterization of fine mode size properties was better when fine mode was predominant. The threshold of $\alpha > 1.2$ has been identified for assuring the highest quality of the retrieval. In such conditions ($\tau(440)>0.2$ and $\alpha > 1.2$), the comparison between GRASP-AOD and AERONET aerosol retrieval algorithm showed a RMSE=0.023 $\mu$m (equivalent to $13.9\%$ in relative terms) for $R_{Vf}$ and a

RMSE=0.016 $\mu$m$^3$/$\mu$m$^2$ (equivalent to $26\%$ in relative terms) for $C_{Vf}$. Evidently, the characterisation of fine mode would improve if radiance measurements (containing scattering information) were added in the retrieval process. For instance, the study by Sinyuk et al. (2020) has recently shown that the uncertainty in $R_{Vf}$ by the AERONET aerosol retrieval algorithm is less than 0.006 $\mu$m when similar thresholds as the ones imposed here are applied ($\tau(440)>0.2$ and fine mode dominated sites). In these regards, the characterization of fine mode properties by GRASP-AOD application becomes useful when only aerosol

optical depth observations are available.

In agreement with the analysis of Torres et al. (2017), GRASP-AOD retrievals of the coarse mode size distribution were less accurate. The analysis has shown very low sensitivity of GRASP-AOD results to the small oscillations of $R_{Vc}$ occurred for individual observations of different aerosol types in different locations. Nonetheless, the general characterization of coarse mode size distribution parameters was reasonable. The latter has been achieved using multiple initial guesses (based on cli-

matological values) and choosing the results with the best fitting. Thus, the achieve agreement of GRASP-AOD retrieval with AERONET aerosol retrieval algorithm for coarse mode showed a RMSE=0.500 $\mu$m (RMSRE=$20\%$) when $\tau(440) > 0.2$. No improvement was found for lower values of Ångström exponent. Therefore, the site by site analysis has showed the RMSRE values mostly between $15 - 25\%$ regardless the dominant type of aerosol over the site. On the other hand, the comparison of $C_{Vc}$ has given a RMSE=0.044 $\mu$m (RMSRE=$30.5\%$) when $\tau(440) > 0.2$. A clear decrease of the RMSRE to $24\%$ was

observed in situation with Ångström exponent under 0.6 (coarse mode dominant).

The effective radius and total volume concentration computed from the GRASP-AOD retrievals have well agreed with the values provided by AERONET aerosol retrieval algorithm. The RMSRE values for the effective radius and for the total volume concentration were $30\%$ and $25\%$, respectively, when $\tau(440) > 0.2$. The analysis for different sites showed quite similar values of the relative errors around $25\%$ for the total volume concentration. On the other hand, the characterization of the effective

radius at coarse mode sites has presented slightly higher RMSRE values (between 30-40$\%$), than at sites with dominant fine mode (RMSRE values between 20-30$\%$).

Thus, the conducted studies showed that GRASP-AOD performs similarly and somewhat better compare to established codes conventionally used for the analysis of only $\tau$ measurements. For example, comparisons to AERONET aerosol retrieval algorithm show similar results for $\tau_f(500)$ that the ones exhibited by the SDA. The characterization of effective radius by

GRASP-AOD approach is comparable at that obtained by linear estimation techniques (relative errors around $30\%$ in both cases). However, the characterization of the total volume concentration by GRASP-AOD approach is significantly better with a relative error of only $25\%$ for GRASP-AOD compare to the $40\%$ obtained by linear estimation techniques (see Pérez-Ramírez et al., 2015). Moreover, GRASP-AOD has provided an excellent characterization of the fine mode size properties, specially in those cases when there is a sufficient aerosol load ($\tau(440)>0.2$) and the fine mode is dominant ($\alpha > 1.2$). In

addition, the GRASP-AOD application retrieves all the parameters at the same time which can be considered as an additional strength. Therefore, the description of the GRASP-AOD retrieval and all comparisons with other approaches discussed above demonstrate both the efficiency of the proposed methodology and an important novelty compared to previous algorithms.

Finally, the subsection 4.3 has showed a promising perspective of improving the characterization of the coarse mode by adding available aureole measurements. In should be noted that a straightforward integration of such measurements into GRASP-AOD established scheme is only possible given the flexibility of GRASP code. The detailed consideration of adding aureole data into GRASP-AOD retrieval as well as other innovative retrieval configuration possible with GRASP algorithm, such as the utilization of the multi-pixel approach, are to be explored in future studies.

*Code and data availability.* More detailed information and a free version of GRASP code can be gained at http://www.grasp-open.com/. The website also provides access to all products derived by GRASP activities. This includes all the data processed by GRASP-AOD application which goes beyond the data used in this work. Specifically, the GRASP-AOD data products analyzed in this study (30 AERONET sites in the period 1997-2016) have been saved and stored at https://doi.org/10.5281/zenodo.4010385. These data have been processed with version v1.0.0 of GRASP algorithm.

*Author contributions.* BT and DF carried out this study and the analysis. The results were discussed with other GRASP team members specially with Oleg Dubovik. The manuscript was mainly written by BT with contributions of DF.

*Competing interests.* The authors declare that no competing interests are present.

*Acknowledgements.* The authors are grateful to Oleg Dubovik for valuable discussions and his help in improving the manuscript. Additional thanks to the rest of GRASP team, specially to Tatsiana Lapionak, for her constant help concerning the use of GRASP code. We would like to thank the Cloud Flight GmbH staff involved in GRASP code activities, and specially Daniel Marth for his guiding related to data processing.

This research has mainly been performed within the ESA funded project DIVA (Demonstration of an Integrated approach for the Validation and exploitation of Atmospheric missions). The authors also acknowledge the funding provided by the European Union (H2020-INFRAIA-2014-2015) under Grant Agreement No. 654109 (ACTRIS-2). Acknowledgement are addressed to the project Labex CaPPA: the CaPPA project (Chemical and Physical Properties of the Atmosphere) is funded by the French National Research Agency (ANR) through the PIA (Programme d'Investissement d'Avenir) under contract "ANR-11-LABX-0005-01" and by the Regional Council "Nord Pas de Calais - Picardie" and the European Funds for Regional Economic Development (FEDER).

We thank the AERONET, Service National d'Observation PHOTONS/AERONET, INSU/CNRS, RIMA and WRC staff for their scientific and technical support. We acknowledge AERONET team members for calibrating and maintaining instrumentation and processing data. We would also like to thank the following principal investigators and their staff for maintaining the following sites: Brent Holben (Alta

Floresta, Arica, GSFC, Kanpur, Lanai, Mexico City, Moldova, Mongu, Moscow, Solar Village), Didier Tanre (Banizoumbou, Dakar), Hong-Bin Chen (Beijing), John Vande Castle (Bonanza Creek), Paulo Artaxo (Cuiaba Miranda), Philippe Goloub (Beijing, Capo Verde, Lille), Andrew Clive Banks (Forth Crete), Lucas Alados-Arboledas (Granada), Jack Molinie (Guadelopu), Rachel T. Pinker (Ilorin), Giuseppe Zibordi (Ispra), Sachchida Nand Tripathi (Kanpur), Ian Lau (Lake Argyle), Robert Frouin (Lanai) Valentin Duflot (Reunion - St. Denis), Arnon Karnieli (Sede Boker), Emilio Cuevas-Agullo (St. Cruz de Tenerife), Itaru Sano (Shirahama), Soo-Chin Liew (Singapore), Alkiviadis Bais (Thessaloniki) and Mikhail Panchenko (Tomsk).

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
