# Peer review of "Characterisation of aerosol size properties from measurements of spectral optical depth: a global validation of the GRASP-AOD code using long-term AERONET data."

_Atmospheric Measurement Techniques, 2020_

## Referee Comment (RC1) · Anonymous Referee #1 · 25 Nov 2020

General comments:

Overall, this is a well-written paper describing the large scale validation of the GRASP-AOD product. Considering this product is able to retrieve total column size distribution and some optical properties without the need of having sky radiance measurements, GRASP-AOD will provide valuable information for atmospheric research and will certainly be widely used. I consider that this manuscript fits perfectly into the scope of AMT. I recommend publishing the manuscript, but there are some minor/technical details that I would like to be addressed in this discussion process.

As general comment, this paper presents a very comprehensive and compelling study on the validation of the GRASP-AOD product. A similar study was already performed by Torres et al. (2017). However this new paper is approached as a large-scale validation using AERONET as the most widespread operational network for ground-based aerosol observation. The use of thirty sites and some million of observations worldwide provides robustness to this analysis. However, the results have been listed in this work as a pure sequence of 20 pages with numbers and some partial conclusions that are very difficult for a reader to follow. I therefore suggest that the authors make a synthesis effort so that the results are clearer for the reader.

Specific and technical comments:

Page 4, line 120: This is not the first time that GRASP has been mentioned in the text. Therefore, I recommend including the acronym once GRASP-AOD product is referred.

Page 5, line 153: Please, correct the typo "teen".

Page 6, footer line: Please correct the Cimel version. It is not CE-310 but CE-318.

Page 8, first paragraph: In this part of the text the authors stated that the priority was the selection of sites with high aerosol loads. However, some lines below, they stated that the GRASP-AOD products do not depend on aerosol load. This sentence seems confusing for the reader. It is also confusing the fact that, if your aim is including sites with predominantly clean conditions, why selecting only two among some hundred stations? Please clarify.

Page 11, line 255: Are you using the AERONET Version 2 instead of Version 3? Is this specific process you are talking about in this paragraph not provided in Version 3?

Page 19, second paragraph: The reason for having higher on-average AERONET retrievals in comparison to SDA and GRASP-AOD is attributed by the authors to the radius cut-off used in AERONET to define the two modes. I suggest the authors to describe briefly the differences between the three compared techniques. This description AMTD
would be more enlightening than attributing beforehand the problem to the AERONET's cut-off.

Page 23, last line: Is there a typo or a lost sentence within the text? Please, correct.

Page 24, line 424: "the interest of presenting"

Page 36, second paragraph: In this part of the text is stated that the main interest of having aureole measurements is adding extra information for improving the coarse mode characterization in situations of partial cloudiness. However, I consider that this improvement cannot be linked only to conditions of partial cloudiness. Furthermore, there are other possible and important applications in the use of this type of measurements, such as quality control, cloud screening, among others, that should be acknowledge. Regarding the use of aureole measurements to improve the aerosol characterization, there are published papers that have also followed this philosophy, such as the work published by Román et al. (2017). These authors proposed the use of an all sky camera to add aureole information into the GRASP code. Please acknowledge in this Section.

Page 36, line 645: Refractive indices are necessary to run the GRASP-AOD, even when aureole measurements are performed. But, taking into account that aureole measurements are relatively insensitive to chemical composition, do the authors consider is still relevant the use of climatological data, or the effect of the uncertainty on the refractive index in this case is less important?

Pages 38-40, lines 704, 711, 714 and 744: The statement about the excellent agreement for fine mode is repeated throughout the conclusion section. Please avoid using redundant conclusions in this section.

Page 39, line 739: Spectral Deconvolution Algorithm is written here without the acronym, as the first time in the conclusion section, despite "SDA" has been mentioned in previous lines. Please homogenize the use of acronyms in the text.

---

## Referee Comment (RC2) · Anonymous Referee #2 · 26 Nov 2020

Review for Atmospheric Measurement Techniques

Title: Characterisation of aerosol size properties from measurements of spectral optical depth: a global validation of the GRASP-AOD code using long-term AERONET data

Authors: Benjamin Torres and David Fuertes

General Comments:

This paper presents a lengthy evaluation of the GRASP-AOD retrieval algorithm performance in comparison to both SDA and the Dubovik almucantar retrievals in AERONET.

[Figure]

Comparisons of fine mode AOD and also both fine and coarse size distribution parameters are made. Although these comparisons are comprehensive in some respects there is also a lack of analysis of why there are some biases in some of the results presented (see details below). Additionally, it should be noted that the author's suggested threshold of AOD(440) > 0.2 for retrieval of radii and other size distribution parameters results in the exclusion of most of the measurements in the global AERONET database. See Sinyuk et al. (2020) for the small errors in fine mode radius from the Dubovik retrievals for even very low values of AOD. Figures 26 and 27 in Sinyuk et al. (2020) show that the uncertainty in fine mode radius for fine mode dominated sites is less than 0.01 micron for AOD>0.10. This is much more accurate than the GRASP-AOD retrievals of fine mode radius (as expected when adding sky radiance information) and needs to be emphasized in this paper and included in discussions. The authors need to note that the percentage of cases excluded by the AOD(440)>0.2 is much larger for the entire AERONET database than for the 30 sites they have analyzed in this paper since they did not include many sites that have persistently low AOD (in Table 4).

One issue that requires additional discussion in the GRASP-AOD Inversion section is the selection of the refractive indices. Please write a few sentences about how the complex refractive index is selected for each site (so that readers do not have to go to your 2017 paper). Also state what the radius limits are for the two modes in the bimodal assumption of GRASP-AOD. A discussion on the effect of errors/uncertainty in refractive index is also warranted in the paper. Additionally, please be clear here that you create a climatology of the complex refractive index for each site based on the full sky scan retrievals (that include spectral AOD) in the AERONET database. Therefore this retrieval is not independent and it also cannot be done for a new site since a 'climatology' of the retrievals for that site are required first. How many retrievals over how many seasons would be required to declare that a sufficient climatology exits to run the GRASP-AOD algorithm for a given site? Also for low AOD sites there will never be a robust refractive index climatology therefore it seems that GRASP-AOD retrievals would never be possible for such sites. It would be very useful to provide

[Figure]

some information on the impact of the refractive indices on the retrieved parameters in this current paper or summarize the results from the 2017 paper. For example, what would the results be if the Real part was assumed to be 1.45 for all wavelengths and the imaginary part of 0.005 for all wavelengths? There needs to be some expanded discussion about the differences in the definition of fine versus coarse modes for the different retrieval algorithms in this paper. For the Dubovik retrieval (Dubovik et al., 2006) which you call the AERONET aerosol algorithm (a confusing choice of terms in my opinion), there is a variable radius cutoff from 0.44 to 0.99 micron depending on the minimum between modes in the retrieved size distribution, while for the SDA algorithm the fine mode includes the influence of the tails of the log-normal distributions. This results in some bias in the retrievals (see O'Neill et al. (2003) and Eck et al. (2010)) between these two independent retrieval methods. You should be clear about how the separation of fine and coarse modes are defined in the GRASP-AOD algorithm.

Figure 2: This plot is quite highly correlated with the AOD magnitudes at each site, as expected. Therefore, it is of relatively limited usefulness and should probably be eliminated. A much more informative comparison would have been the fine mode fraction (FMF) of AOD at 500 nm for these retrievals, as this would be less dependent in magnitude on the AOD levels at each site.

Please discuss the systematic underestimation by GRASP (Figure 7) of fine radius which gets significantly worse as fine radius increases, even for the best conditions of high AOD and high AE. It is surprising that the authors did not investigate this bias that occurred in multiple sites. Provide some analysis or at least speculation on the reasons for the GRASP-AOD underestimation of fine mode radius versus the Dubovik almucantar retrievals and why this error increases for the largest fine radius cases.

Also it is necessary to provide some analysis and discussion of the two distinct populations of the coarse mode radii in the top row plots in Figure 9. I suspect that the larger radii population is from fine mode dominated cases and the lower radii cluster from dust dominated cases, but this needs to be analyzed. If this is the case then the

claim for higher accuracy that you imply is somewhat suspect since the accuracy of the coarse mode radii when fine mode dominates the signal is VERY low due to very low coarse mode AOD resulting in very little coarse mode information content in the spectra of total extinction AOD. Additionally you have again neglected to include information from the study of Sinyuk et al. (2020) that shows that the accuracy of the retrieval of coarse mode radii is much less than that for fine mode aerosol.

Explain why the effective radius of both modes combined are analyzed at all in this paper. I have never seen a published peer-reviewed paper that shows the value or justification in combining the information from both modes into a total effective radius and total volume concentration value. If you have information that shows the value of these combined mode parameters then please discuss it in the text plus provide references in order to convince the reader of their value. The separate fine and coarse mode parameters on the other hand have much value and have been utilized in numerous published papers in the scientific literature.

Please quantify what you refer to as 'good capacity' of the GRASP-AOD retrieval of fine mode radius in the Conclusions section. For the Rvf the uncertainty of GRASP-AOD is ∼0.023 micron for fine mode dominated data while for the AERONET Dubovik algorithm almucantar retrievals the accuracy is ∼0.006 for AOD(440)>0.2 for the fine mode observations (large AE). You lack references to the values of Rvf and Rvc from Sinyuk et al. (2020) as a way to compare the accuracy of these retrievals (see Fig 27 for example for the fine mode sites Rvf uncertainty).

On a positive note: You should note that with the newer Cimel instruments the cross scan in the solar aureole is taken with every AOD spectra measurement sequence as a cloud screening data set for the detection of cirrus. This in effect provides aureole sky radiance values for every AOD measurement made with these newer Cimel instruments. This could provide a potentially powerful addition to your retrievals and should be explored for even the fine mode dominated cases to assess any impact of this added aerosol information.

Specific Comments:

Line 9: Misspelling of 'diverse'

Line 20: What about for low AOD cases? Sinyuk et al. (2020) show that the fine mode radius is retrieved very accurately down to very low AOD.

Line 21: Should be AE>1.2. Seems like this is a bit careless to get such a basic statement backwards in the Abstract.

Line 23: This is an odd choice of words here: oscillations implies somewhat periodic variability between two states, not sure the authors really mean that here.

Line 27-28: Strange terminology for presenting statistics. What exactly is the RMSE values of a correlation? Please be clearer and more precise.

Line 50: Should be 'continuous' instead of 'continued'.

Line 54-55: This sentence has some very awkward English and should be re-written. Hard to know the exact meaning as it is now.

Line 61: High accuracy is even more important than the high precision of the sun photometer measurements.

Line 73: "cloud processing" would be much more appropriate here than "cloud formation"

Line 73: 'plums' should be "plumes"

Line 77: Large solar zenith angles are no longer required with the Hybrid scan in AERONET, see a description of the hybrid scan in Sinyuk et al. (2020).

Line 110 & line 118: 'punctual studies': this is awkward English, better to choose a different word, perhaps 'specific studies'? However, not really sure what you are trying to say here.

Line 145-146: This is a very strange and misleading statement. The only cloud screen-

ing check from Smirnov et al. (2000) that is also utilized in the V3 cloud screening is the triplet variability check and even then the magnitude of this triplet threshold has been changed plus spectrally limited to longer wavelengths in V3 (see Giles et al. 2019). Other checks are unique to V3 and also V3 is completely automatic, while the V2 cloud screening of Smirnov et al. required an analyst to remove numerous cloud contaminated observations. This sentence needs to be re-written to be more factual and informative.

Line 146: You need to state that the accuracy of the Level 2 spectral AOD is ∼0.01 and ∼0.02 in the UV (Eck et al. 1999) since highly accurate data is the key to the applicability of the GRASP-AOD retrievals you are discussing.

Line 148-149: You should state here that the fine mode AOD from the Dubovik retrieval is given at 440 and 675 nm, not 500 nm. Since you are describing the data sources in this section you should be more accurate as there is no 500 nm fine mode AOD directly provided by the Dubovik retrieval. Please write how you computed the fine mode AOD at 500 nm from the Dubovik retrievals.

Line 153: 'teen' should be 'ten'

Line 155: It is common to most Cimels in the network, but the older PHOTONS group polarized Cimel model do not have the 340, 380 or 500 nm channels. Instead they have three polarized 870 nm channels. Five of your 30 selected sites Dakar, Capo Verde, Banizoumbo, Guadaloupe and Beijing do not have the 340, 380 and 500 nm channels for most or all years of this analysis. For Dakar 1997-2008 plus 2010 do not have the 340, 380 or 500 nm channels and for the Capo Verde site most of the record you analyzed 1997-mid 2016 lack these key channels. Additionally the Beijing site has spectral AOD only from 440, 675, 870 and 1020 nm for all the years 2002 through 2015. Guadaloupe lacks the 340, 380 and 500 nm AOD for 1999 through 2008. Banizoumbo lacks the AOD at 340, 380 and 500 nm for the entire measurement record. The spectral AOD information content of these instruments is much reduced

compared to the full wavelength range, therefore it is very important that you mention this and address this issue in the analyses of these sites. You should compare your algorithm with and without the 340, 380 and 500 nm channels for a few sites that have the full wavelength suite of channels. Note that the AERONET group did a full analysis of comparisons of the SDA algorithm with various wavelength combinations in order to determine the wavelengths necessary for Level 2 quality retrievals. The SDA algorithm excludes the 340 and 1020 nm channels since the uncertainties in AOD are higher for these wavelengths. The 340 nm filters have been the least stable (temporal degradation) of all the other wavelength filters plus have out-of-band blockage issues in many 340 nm filter batches. At 1020 nm the silicon detector has a large temperature sensitivity and must be corrected using the sensor head temperature, plus there is significant water vapor absorption at 1020 nm that is accounted for from the retrievals made at 945 nm. These two factors increase the uncertainty at 1020 nm relative to the other wavelengths. The lack of discussion of these issues in this GRASP-AOD paper should be corrected.

Line 162: Are these multi-year averages computed from daily averages or from all individual instantaneous vales weighted equally? Averaging daily first and then monthly gives a more representative values of the monthly and annual aerosol loading. It is important to clearly write in the paper how you computed these averages.

Line 197-198: This is not really true. The Lanai site does not have any L2 retrievals for refractive index since AOD(440)<0.4, but it does have very many L2 retrievals for the size distributions.

Line 203-204: Please provide a sentence or two to describe how the options for the dominant mode radii initial guesses change as a function of Angstrom Exponent. I do not see this for the coarse mode as for coarse mode dominated cases AE<0.6 in Table 2 as there are only 2 static choices of coarse radius while for mixed modes 0.6<AE<1.2 there is one static and one dynamic coarse mode radius.

[Figure]

Line 205-206: If the standard deviation (width) of each mode is fixed, then you need to give these values here instead of forcing the reader to look them up in another paper.

Line 220-222: Please explain the fitting here in more detail. I assume you compute spectral AOD based on the retrieved size distribution plus the assumed refractive indexes and then compare this to the measured spectral AOD. A written discussion in the text is needed.

Line 244-245: You need to be more precise here in your explanation for the lack of SDA retrievals at L2 for these sites that had old style polarization Cimels with only 4 wavelengths of measured AOD data. The reason for no L2 SDA retrievals is the lack of 380 and 500 nm AOD values for the instrument types deployed at these sites. You need to prove that the GRASP-AOD retrievals give the same values for 4 channel AOD input versus 7 channel AOD input. This should be especially important at the Beijing site which is fine mode dominated and therefore has much greater non-linearity in the AOD spectra in logarithmic space. For coarse mode desert dust sites this will not matter nearly as much as the AOD spectra is relatively flat with little non-linearity in logarithmic coordinates.

Line 255: It should be noted that the fine/coarse mode radius separation value is the same for Version 3 as it was in Version 2.

Line 258-259: Please add "for each mode as well as for the entire size distribution".

Line 260: This is the wrong vocabulary word ('mechanical') here. I suggest that this word can be eliminated and the sentence will be clearer. I suggest: "The separation between fine/coarse mode..."

Line 262: How do you make this interpolation? In log-log space by Angstrom Exponent relationship, or by 2nd order fit of AOD in log-log space which is the most accurate methodology.

Line 267: This is just way too simplistic an estimate for this paper. The number of

[Figure]

AOD spectra measured per day in AERONET depends on site latitude and day of year, resulting in differing number of day-length hours. In addition, the newer instruments are set to take 5-minute sampling interval data versus 15-minute sampling intervals in the old Cimels for direct sun AOD observations. More details on the variable number of AOD measurements per day in AERONET are required in a paper that utilizes AOD spectra as the primary input parameter.

Line 273: This is an inaccurate statement since some sites only have the 440, 675, 870 and 1020 nm AOD while most other sites add the 340, 380 and 500 nm channels to those.

Line 275-276: Except as you noted that the SDA does not make a retrieval when the 380 nm AOD are missing.

Line 322-327: No real surprise here as these 3 sites have the highest AOD levels in the entire AERONET network. I suggest adding the average AOD values in the table and plotting the RMSE versus this average AOD. For the La Reunion site you should add the phrase: "...because the AOD were lowest for this site."

Line 350-351: Please include an investigation and explanation of some cases in the two branches of the Fig 5 plots for AE<0.6 of GRASP-AOD versus AERONET and SDA versus AERONET (Dubovik). An attempt should be made to explain these two data populations and why they diverge as fine AOD increases.

Line 371: Please mention that this is a quality control issue for SDA due to insufficient AOD wavelengths for highest accuracy of the retrievals.

Line 381: It is not just 500 nm but also 340 and 380 nm that are not available in the old Polarized Cimels. Please add this to the text. To prove the level of robustness you have claimed, for Beijing you need to run the GRASP-AOD retrievals for the full 7 channels (340-1020 nm) for years when this type of Cimel was operating there and then subsequently run the GRASP retrievals with only the 440 , 675, 870 and 1020 nm

data as input for these same exact measurement scans. Only this direct comparison of the same AOD spectra and almucantars but with different spectral channels used as input can really determine just how robust the 4 channel GRASP-AOD retrievals are.

Line 413-415: It should be noted that the retrieval of the fine mode radius when the coarse mode dominates (AE<0.6) also has a large uncertainty in the Dubovik retrieval with sky radiance information, see Sinyuk et al. (2020). Therefore the lack of correlation with GRASP is also due largely to very weak information and thus large uncertainty for fine radius in the AERONET almucantar retrievals for coarse mode cases.

Line 423: This is an incomplete sentence here should probably be deleted.

Line 431: Should change 'column' to 'row' here.

Line 437: Please discuss the reasons for this systematic underestimation by GRASP which gets worse as fine mode radius increases in Figure 7 for all sites shown, even for the best conditions of high AOD and high AE.

Line 467: Please discuss the reason for the 2 populations that are obvious in most of the plots of Figure 8.

Line 482: It is interesting that you mention 1640 nm here since the GRASP-AOD retrieval does not use this wavelength of AOD data. Theoretically inclusion of the 1640 nm AOD should indeed provide more information on the radius of the coarse mode, so you should discuss that here.

Line 485: Please be clear here that these are AERONET climatological values.

Line 695: This is the wrong word choice ('axes') here. Although the writing is in general relatively good from the English grammar and vocabulary aspects, please have a native English speaker review the manuscript to catch the various instances of awkward phrasings and/or poor vocabulary choices.

Line 701: Nothing involving real data is ever a perfect correlation. Please give the exact

value of correlation here even if it is very close to 1.

---

## Referee Comment (RC3) · Anonymous Referee #3 · 16 Dec 2020

General comments:

The aim of this paper is to show that the GRASP-AOD code has the potential to be used for large scale datasets either for aerosol climate studies or for near real time modeler needs. The validation based on 2.8 million GRASP-AOD retrievals using AERONET AOD observations from 30 sites during 20 years makes the work robust enough to reach appropriate conclusions. The paper is to long taking into account the methodology used, the results and the prior knowledge published about this type of AOD inversion codes. I suggest to make a synthesis relying on the bibliography already

published, including the new considerations used that can improve this type of AOD inversion codes (comparative and differences with other papers already published). The paper is well written and into the scope of AMT. I recommend the publication of this paper, but there are some issues should be addressed prior to publication. The Editor will judge.

The AOD inversion codes have used in different papers from many years. These type of inversion codes are based on the aerosol scattering equation that express the dependence of the spectral variation of AOD on the aerosol size distribution, and also depend of the Qext parameters (particle extinction efficiency factors), which in turn depend on the wavelength, the refractive index and particle radius. As example, King et al. (1978) already pointed out that the definition of the particle radius interval on which the inversion method can be correctly used, and the assumption of realistic refractive index values are the most crucial points in any rigorous application of inversion methods applied to spectral series of the AOD. On the other hand, the independent information content on the optical characteristics of columnar aerosols is contained primarily in the particle radius interval from 0.1 to 2 microms, approximately, for AOD measurements covering spectral range 340-1020 nm. On the other hand, the iterative procedures modified the radius interval within the prescribed ranges, and the best results were obtained for reduced radius range. In this sense, with this type of codes the results are limited to the accumulation mode. On the other hand, some AOD inversion algorithms use a single refractive index, while the true is dependent on wavelength. The assumption of an a priori defined refractive index in the AOD inversion procedures may lead to very different derive size distributions, but other authors (e.g., Yamamoto and Tanaka, 1969; King et al., 1978; González and Ogren, 1996) show that the shape of the retrieved aerosol size distribution is not substantially altered as a result of using such assumptions. In this sense, this paper should take into account previous work and show the improvements that can be made. Taking into account previous results, obviously these type of inversion algorithms would not work well for coarse particle modes just considering only the AOD spectral values. Spectral aureole data (sky radiances)

are required to achieve good results in coarse mode.

Specific comments:

-Lines 85-95. To motive the importance of this work, the authors comments that many AERONET sites are plagued by several months of partial cloudiness (no sky radiance measurements) . . . but later they use climatological values for refractive index and information about radius modes. How it is possible for this type of AERONET stations, and how representative are these values? also for future applications to night measurements. The columnar aerosol properties change from day to night, depend on sources, the air masses transport, the planetary boundary layer high ... Also, a study of the GRASP-AOD sensitivity to the refractive index is needed.

-Line 185. The GRASP-AOD code assumes the refractive index as known. Which one has been chosen for each AERONET station and aerosol type? Can be Included in Table-1? On the other hand, the aerosol type selected for each station (Table 1) can be the more frequent (climatology), but not all ways are the same. As example, the Saharan dust outbreaks. How these facts affect the inversion products?

Lines 190-195. If the refractive indices are assumed, what happens, as example, with stations where there are many clouds and cannot be computed with the sky radiance data? There are no data? Do you use the climatological value? How much data have you used to obtain this climatological value, and how is it distributed throughout the year? In order to these results will be realistic, an extensive database should be available and the appropriate refractive index value used for each atmospheric condition. The purpose of this work is to show that the GRASP-AOD application has the potential to be used for large scale datasets.

Lines 480-525. Obviously, the algorithm does not work well for coarse particle mode just taking into account only the AOD spectral parameters and a climatological value of the refractive index. But we already knew these results from the papers published related with these type of inversion codes. The sky radiance data is needed to achieve

good results in coarse mode. I think this section should be shortened or removed from the paper. Also, the last sentence of the abstract is a well-known result and it is not new.

Lines 200-225. The criteria are based mostly on analyst's experience. The authors show "Due to the low sensitivity of GRASP-AOD to the shape of the modes... we have used strong a priori constraints on the actual values for the standard deviation of both modes... in practice, their values are very similar to the given initial guess values". On the other hand, in Line 340 the authors show: "The larger uncertainties observed for Solar Village compared to GSFC can be extrapolated to all sites with coarse mode predominance with respect to the sites with fine mode predominance", and the following lines. Taking into account the papers published so far, it is clear that this methodology can only be applied to places where the fine mode predominates.

In my opinion, this work should be drastically reduced, showing only those aspects that can improve the results of the works already published. On the other hand, the usefulness of using climatological values in the a priori assumptions should be better discussed, taking into account the sensitivity of the GRASP-AOD model to the parameters.

---

## Author Comment (AC1) · 26 Feb 2021

**General comments:**

Overall, this is a well-written paper describing the large scale validation of the GRASP-AOD product. Considering this product is able to retrieve total column size distribution and some optical properties without the need of having sky radiance measurements, GRASP-AOD will provide valuable information for atmospheric research and will certainly be widely used. I consider that this manuscript fits perfectly into the scope of AMT. I recommend publishing the manuscript, but there are some minor/technical details that I would like to be addressed in this discussion process.

We would like to thank the anonymous referee 1 for reviewing the manuscript. We are glad for the overall positive assessment regarding the manuscript.

As general comment, this paper presents a very comprehensive and compelling study on the validation of the GRASP-AOD product. A similar study was already performed by Torres et al. (2017). However this new paper is approached as a large-scale validation using AERONET as the most widespread operational network for ground-based aerosol observation. The use of thirty sites and some million of observations world-wide provides robustness to this analysis. However, the results have been listed in this work as a pure sequence of 20 pages with numbers and some partial conclusions that are very difficult for a reader to follow. I therefore suggest that the authors make a synthesis effort so that the results are clearer for the reader.

Thank you for your comment. We agree that this study may provide more robustness to GRASP-AOD application beyond the results obtained by Torres et al (2017). The large number of parameters analyzed may have difficulted to follow the description of the comparisons presented. However, we think that no table has been "just listed" without its particular analysis (more or less detailed, depending on the relevance of its results) and conclusions. Nevertheless, we have tried to emphasize some of the results in the along the manuscript since we agree that some descriptions were a bit tedious.

**Specific and technical comments:**

Page 4, line 120: This is not the first time that GRASP has been mentioned in the text. Therefore, I recommend including the acronym once GRASP-AOD product is referred.

Thank you for your suggestion. Even though GRASP-AOD is forementioned previously in the manuscript, this is actually the first time that the whole GRASP project is mentioned. The logic of the introduction made us to mention the whole GRASP project at the end, even though GRASP-AOD is previously defined (with the reference of Torres et al. 2017).

Page 5, line 153: Please, correct the typo "teen".

Thank you very much for your comment. We have corrected the mistake.

Page 6, footer line: Please correct the Cimel version. It is not CE-310 but CE-318.

Yes, you are right. Thank you again for the correction.

Page 8, first paragraph: In this part of the text the authors stated that the priority was the selection of sites with high aerosol loads. However, some lines below, they stated that the GRASP-AOD products do not depend on aerosol load. This sentence seems confusing for the reader. It is also confusing the fact that, if your aim is including sites with predominantly clean conditions, why selecting only two among some hundred stations? Please clarify.

Validation and climatology studies are normally carried out at key AERONET sites which are determined by its aerosol load and the availability of a long-term time-series data (see for instance Dubovik et al 2002 or Gilles et al. 2012). In this sense, our work has tried to be consistent with previous studies and 28 sites out of 30 accomplish these requirements. The inclusion of Lanai and St. Denis was done to present a couple of examples of marine aerosol sites. These sites (as all marine aerosol sites - Smirnov et al. 2002) do not fit into the general rule of high aerosol load. Please also note that these two specific sites were selected since they presented the longest time series data of all marine aerosol sites.

Page 11, line 255: Are you using the AERONET Version 2 instead of Version 3? Is this specific process you are talking about in this paragraph not provided in Version 3?

As stated in section 2.1, all the data used in the study have been taken from AERONET Version 3. We just wanted to indicate here that this specific routine was implemented in Version 2 and has been kept like this in Version 3. In AERONET Version 1, the mode separation was different (fix cutoff at 0.6 $\mu$m). Anyway, to make it clearer for the

reader we have added that this routine has been kept in Version 3 (our only data source).

Page 19, second paragraph: The reason for having higher on-average AERONET retrievals in comparison to SDA and GRASP-AOD is attributed by the authors to the radius cut-off used in AERONET to define the two modes. I suggest the authors to describe briefly the differences between the three compared techniques. This description would be more enlightening than attributing beforehand the problem to the AERONET's cut-off.

Thank you for your comment. The effect of the cutoff was previously discussed in O'Neill et al. 2003 though we have agreed to recall it here. Following the logic of the manuscript structure, we think that the explanation about the separation of fine/coarse mode by the different methodology fits better in section 2 (and then we refer to it in section 3.1). In this sense, we have added the following paragraph in subsection 2.3:

*"The mechanical separation fine/coarse mode in the detailed size distribution is used as well to estimate the optical thickness for fine and coarse mode at 440, 675, 870 and 1020nm, from the AERONET aerosol retrieval algorithm outputs. The particular values at 500nm, $\tau_f(500)$, have been interpolated for our validation study. Note that the way that the two modes are separated by the AERONET aerosol retrieval algorithm represents itself an inherent source of error to estimate fine/coarse mode optical thickness. In fact, the distribution of fine and coarse particles are continuous entities which overlap between them and they spread beyond the border established by the separation point or cutoff. As explained by O'Neill et al. (2003), a simple analysis of Mie kernels would show that the optical depth due to coarse particles for radii smaller than the cutoff (wrongly included in $\tau_f(500)$ calculations) is larger than the optical depth due to fine particles for radii larger than the cutoff (wrongly excluded from $\tau_f(500)$calculations). Therefore, the fine mode optical depth is overestimated while the coarse mode optical depth is underestimated. This effect is typically small, and it is more significant if the coarse mode dominates. Neither SDA nor GRASP-AOD application present this issue since the two modes can overlap in both algorithms. In the case of GRASP-AOD, the primary outputs are two independent log-normal functions which represent separately the fine and coarse mode as aforementioned. The values of $\tau_f(500)$ and $\tau_c(500)$ are derived from the aerosol optical depth values calculated individually for each log-normal function."*

Page 23, last line: Is there a typo or a lost sentence within the text? Please, correct.

Thank you very much. It was part of the following sentence before modification. It's removed now.

Page 24, line 424: "the interest of presenting"

Yes, thank you again.

Page 36, second paragraph: In this part of the text is stated that the main interest of having aureole measurements is adding extra information for improving the coarse mode characterization in situations of partial cloudiness. However, I consider that this improvement cannot be linked only to conditions of partial cloudiness. Furthermore, there are other possible and important applications in the use of this type of measurements, such as quality control, cloud screening, among others, that should be acknowledge. Regarding the use of aureole measurements to improve the aerosol characterization, there are published papers that have also followed this philosophy, such as the work published by Román et al. (2017). These authors proposed the use of an all sky camera to add aureole information into the GRASP code. Please acknowledge in this Section.

Thank you for your comments here. However, we believe that there are some misunderstandings at this point.

First of all, when we talk about aureole measurement, we do it from the point of view of the retrieval improvements (information contained). In these regards, we can only conclude that the characterization of coarse mode is considerably better as shown in the section later on. Certainly, the use of new specifically designed aureole scenarios is helpful for other aspects as the referee stated. Nevertheless, these aspects are out of the scope/interest of this paper, moreover, considering that we have used aureole measurements from existing almucantar and not any kind of new specific scenarios. On the other hand, the mention of "cloudy conditions" refers to the fact that if there is clear sky conditions we can directly benefit from the use of the whole almucantar measurements (AERONET aerosol retrieval) instead of performing GRASP-AOD or GRASP-Aureole inversion.

Respect to the citation to the work by Roman et al. (2017), please note that both authors were co-authors of the study. Therefore, we are aware of the strengths and limitations of that study, especially in those aspects related to the use of the GRASP code. In this sense, the use of GRASP forward module to validate the normalized radiance measurements obtained from the sky camera was revealed as a solid tool. However, the results obtained by the use of GRASP retrieval in that work were quite criticized (internally by coauthors and by external referees) and still lack validation as highlighted in the conclusions of the paper. Nevertheless, the work was already cited in other parts of the study (overall we have a quite positive vision of that scientific work) but we do not acknowledge it intentionally in this section since we reckon it would be

misleading for the readers. Further details are given in the answer to the following comment since we think both comments are related.

Page 36, line 645: Refractive indices are necessary to run the GRASP-AOD, even when aureole measurements are performed. But, taking into account that aureole measurements are relatively insensitive to chemical composition, do the authors consider is still relevant the use of climatological data, or the effect of the uncertainty on the refractive index in this case is less important?

We do agree with the referee that aureole measurements are not sensitive to refractive index. This affirmation can not be concluded at all by the work of Roman et al. (2017). In that work, the first author proposed to retrieve the refractive index only with relative aureole measurements and aerosol optical depth measurements. It should be noted that the election of that inversion strategy was exclusively chosen by the first author and was done against the advice of GRASP retrieval experts of that study: O. Dubovik, D. Fuertes, T. Lapyonok and B. Torres. The use of pre-fixed refractive index (and sphericity parameter) was highly recommended, and this recommendation was done based on previous sensitivity studies.

Thus, the study "Accuracy assessments of aerosol optical properties retrieved from Aerosol Robotic Network (AERONET) Sun and sky radiance measurements" by Dubovik et al. (2000) concludes that to accurate retrieve the refractive index (and the derived single scattering albedo) scattering information between 3°-150° is needed. The study points out that if the scattering information is reduced to 3°-100° there is already some loss of information, but the results are still acceptable (especially if there is enough aerosol load). If the scattering information is smaller than 3°-100°, there is a dramatic decrease in retrieval accuracy for the refractive index (see for instance figures 4, 8, 10 and 12 of that study). As a result, traditionally AERONET only gives the label of quality assured if the solar zenith angle of the almucantar measurement is higher than 50° (which assures scattering information between 3°-100°). Other studies that came after as Torres et al. 2014 (figures 3 and 10) or the more recent Sinyuk et al. 2020 are in line with the aforementioned results. The latter study points out that the use of Hybrid scans reduces the requirement of solar zenith angle to 25°, but because in this new configuration a scattering range between 3°-100° is assured. It should be also remarked that all the aforementioned well-conceived sensitivity studies were based on well calibrated absolute radiances (with an estimated 5% error) while the study by Roman et al. (2017) made use of normalized radiance measurements with errors up to 10-14% in a much shorter spectral range (469 - 608 nm against 440 - 1020 nm), which implies much less information contained. For all these reasons, we think that to retrieve the aerosol refractive index with the methodology proposed by Roman et al. (2017) does not respond to a sufficient scientific evidence. That's why

we believe that to cite the study by Roman et al. 2017 here could mislead the readers and make them believe that aerosol refractive index could be actually retrieved by aureole measurements (added to aerosol optical depth measurements) while several other studies have shown the opposite.

To answer specifically the referee's question, there is a strong correlation between the real refractive index and the fine mode characteristics due to the anomalous diffraction theory of Van de Hulst (Van de Hulst, 1957) as primarily discussed in Yamamoto and Tanaka (1969) and later by King et al. 1978. In this sense, if the real refractive index is not correctly retrieved it represents a source of error in the retrieval of fine mode parameters. As previously commented, the aureole measurements used in this section do not present a particular advantage to retrieve the real refractive index, since we would need scattering information between 3°-100°. That's why we should provide exactly the same information (in terms of refractive index) as in the regular GRASP-AOD application.

Pages 38-40, lines 704, 711, 714 and 744: The statement about the excellent agreement for fine mode is repeated throughout the conclusion section. Please avoid using redundant conclusions in this section.

Thank you for your comment but we believe that we are talking about different things in each paragraph and we would like to keep as it is. First in line 704, we talk about the characterization of $\tau_f(500)$. In the paragraph from line 709 and 720, which includes lines 711 and 714, we summarize the results obtained for $R_{Vf}$ and $C_{Vf}$. Finally, the line 744 corresponds to the paragraph of the comparison of GRASP-AOD with other codes that performs only with AOD measurements. We think that it is important to highlight the fact that it is the only existing code that gives a characterization of the fine mode radius and volume concentration. Maybe to say again that it works only in certain circumstances (AOD(440) > 0.2 and AE >1.2) could be avoided, however, please note that the other referee's comments demand to recall this result (the aforementioned conditions) and the consequent reduction of data in its applicability.

Page 39, line 739: Spectral Deconvolution Algorithm is written here without the acronym, as the first time in the conclusion section, despite "SDA" has been mentioned in previous lines. Please homogenize the use of acronyms in the text.

Thank you very much for your comment. We have corrected it.

---

## Author Comment (AC2) · 26 Feb 2021

**General Comments:**

This paper presents a lengthy evaluation of the GRASP-AOD retrieval algorithm performance in comparison to both SDA and the Dubovik almucantar retrievals in AERONET.

Comparisons of fine mode AOD and also both fine and coarse size distribution parameters are made. Although these comparisons are comprehensive in some respects there is also a lack of analysis of why there are some biases in some of the results presented (see details below). Additionally, it should be noted that the author's suggested threshold of AOD(440) > 0.2 for retrieval of radii and other size distribution parameters results in the exclusion of most of the measurements in the global AERONET database. See Sinyuk et al. (2020) for the small errors in fine mode radius from the Dubovik retrievals for even very low values of AOD.

We would like to thank the anonymous referee 2 for the detailed review of the manuscript. We have added most of the suggestions proposed by the referee (as presented later) and we reckon that the manuscript has been improved.

At the same time, we would like to clearly state that the scope of the paper is not to compare the performance of both algorithms (GRASP-AOD and AERONET aerosol retrieval algorithm). As largely indicated in the introduction, the information contained in the sky radiance measurements provides the possibility to characterize even very minor features in the size distribution shape and to retrieve the spectral refractive indeces. In this context, AERONET aerosol retrieval algorithm has been used as a reference to perform the validation of GRASP-AOD (not a comparison).

Figures 26 and 27 in Sinyuk et al. (2020) show that the uncertainty in fine mode radius for fine mode dominated sites is less than 0.01 micron for AOD>0.10. This is much more accurate than the GRASP-AOD retrievals of fine mode radius (as expected when adding sky radiance information) and needs to be emphasized in this paper and included in discussions. The authors need to note that the percentage of cases excluded by the AOD(440)>0.2 is much larger for the entire AERONET database than for the 30 sites they have analyzed in this paper since they did not include many sites that have persistently low AOD (in Table 4).

Thank you for your comment, we have added the values obtained by Sinyuk et al. (2020) in the conclusion of the paper to put in perspective the results obtained here. Nevertheless, the term "good capacity" refers to the "more than acceptable results" given the information contained in only AOD measurements.

One issue that requires additional discussion in the GRASP-AOD Inversion section is the selection of the refractive indices. Please write a few sentences about how the complex refractive index is selected for each site (so that readers do not have to go to your 2017 paper). Also state what the radius limits are for the two modes in the bimodal assumption of GRASP-AOD. A discussion on the effect of errors/uncertainty in refractive index is also warranted in the paper. Additionally, please be clear here that you create a climatology of the complex refractive index for each site based on the full sky scan retrievals (that include spectral AOD) in the AERONET database. Therefore this retrieval is not independent and it also cannot be done for a new site since a 'climatology' of the retrievals for that site are required first. How many retrievals over how many seasons would be required to declare that a sufficient climatology exits to run the GRASP-AOD algorithm for a given site? Also for low AOD sites there will never be a robust refractive index climatology therefore it seems that GRASP-AOD retrievals would never be possible for such sites. It would be very useful to provide some information on the impact of the refractive indices on the retrieved parameters in this current paper or summarize the results from the 2017 paper. For example, what would the results be if the Real part was assumed to be 1.45 for all wavelengths and the imaginary part of 0.005 for all wavelengths?

We have followed the suggestion of the referee and we have added the remarks in a new subsection in the discussion. As suggested we have summarized the results of previous works and we have reprocessed the data from Mongu site assuming refractive indices of 1.45 – 0.005i in the whole period to see the effect of taking standard refractive index values instead of climatological values. Note that we have selected Mongu since the monthly climatological values are around 1.51-0.021i which differ the most with the proposed generic refractive index of 1.45-0.005i (sites like GSFC, Ispra or Shirahama would not result affected by this election).

The interval for the retrieved radii is always the same and goes from 0.07 to 0.7μm for the fine mode, and from 0.7 to 5 μm for the coarse mode. This information has been added to the paper.

There needs to be some expanded discussion about the differences in the definition of fine versus coarse modes for the different retrieval algorithms in this paper. For the Dubovik retrieval (Dubovik et al., 2006) which you call the AERONET aerosol algorithm (a confusing choice of terms in my opinion), there is a variable radius cutoff

from 0.44 to 0.99 micron depending on the minimum between modes in the retrieved size distribution, while for the SDA algorithm the fine mode includes the influence of the tails of the log-normal distributions. This results in some bias in the retrievals (see O'Neill et al. (2003) and Eck et al. (2010)) between these two independent retrieval methods. You should be clear about how the separation of fine and coarse modes are defined in the GRASP-AOD algorithm.

We have used the term AERONET aerosol retrieval algorithm since it has been the term used in the study Sinyuk et al. (2020). Maybe it is not a good choice, and Dubovik retrieval would be clearer, but we have prioritized to be consistent with the terminology of the latest paper (regarding retrieval/inversion) published by AERONET staff.

Regarding the separation of the fine/coarse mode, GRASP-AOD algorithm considers two independent modes (represented by two log-normal functions) that can overlap between them. In this sense, the separation is similar to the one made in SDA. Therefore, the overestimation of $\tau_f$ found for the retrievals of AERONET aerosol algorithm respect to GRASP-AOD can be partially justified with the same explanation as the one found in O'Neill et al. (2003) and Eck et al. (2010). We have added the following paragraph in section 2.3.2 (and we have referred to it in section 3.1).

*"The mechanical separation fine/coarse mode in the detailed size distribution is used as well to estimate the optical thickness for fine and coarse mode at 440, 675, 870 and 1020nm, from the AERONET aerosol retrieval algorithm outputs. The particular values at 500nm, $\tau_f(500)$, have been interpolated for our validation study. Note that the way that the two modes are separated by the AERONET aerosol retrieval algorithm represents itself an inherent source of error to estimate fine/coarse mode optical thickness. In fact, the distribution of fine and coarse particles are continuous entities which overlap between them and they spread beyond the border established by the separation point or cutoff. As explained by O'Neill et al. (2003), a simple analysis of Mie kernels would show that the optical depth due to coarse particles for radii smaller than the cutoff (wrongly included in $\tau_f(500)$ calculations) is larger than the optical depth due to fine particles for radii larger than the cutoff (wrongly excluded from $\tau_f(500)$ calculations). Therefore, the fine mode optical depth is generally overestimated while the coarse mode optical depth is generally underestimated. This effect is typically small, and it is more significant if the coarse mode dominates. Neither SDA nor GRASP-AOD application present this issue since the two modes can overlap in both algorithms. In the case of GRASP-AOD, the primary outputs are two independent log-normal functions which represent separately the fine and coarse mode as aforementioned. The values of $\tau_f(500)$ and $\tau_c(500)$ are derived from the aerosol optical depth values calculated individually for each log-normal function."*

Figure 2: This plot is quite highly correlated with the AOD magnitudes at each site, as expected. Therefore, it is of relatively limited usefulness and should probably be eliminated. A much more informative comparison would have been the fine mode fraction (FMF) of AOD at 500 nm for these retrievals, as this would be less dependent in magnitude on the AOD levels at each site.

Figure 2 shows a graphical evaluation of how the 3 methods compare with each other. We think that this figure allows the reader to get a general idea without needing to analyze in detailed Table 4. Otherwise, we agree with the referee about the correlation of the results respect to the fine mode fraction.

Please discuss the systematic underestimation by GRASP (Figure 7) of fine radius which gets significantly worse as fine radius increases, even for the best conditions of high AOD and high AE. It is surprising that the authors did not investigate this bias that occurred in multiple sites. Provide some analysis or at least speculation on the reasons for the GRASP-AOD underestimation of fine mode radius versus the Dubovik almucantar retrievals and why this error increases for the largest fine radius cases.

We would like to present first the differences (absolute first and then relative with sign) for diverse values of the fine mode median radius. Note that we use differences and not RMSE (as in the manuscript) to check the mentioned bias.

| Values $[R_{vf}(AERO)]$ | N° Data | $R_{vf}(GRASP) - R_{vf}(AERO)$ | $\frac{R_{vf}(GRASP) - R_{vf}(AERO)}{R_{vf}(AERO)}$ [%] |
|---|---|---|---|
| < 0.14 | 5313 | - 0.002 | - 1.4% |
| 0.14 - 0.17 | 12678 | - 0.007 | - 4.7% |
| 0.17 - 0.20 | 8849 | - 0.011 | - 6.3% |
| 0.20 - 0.23 | 4514 | - 0.018 | - 8.7% |
| > 0.23 | 2378 | - 0.029 | - 12.7% |
| **All** | **33732** | **- 0.011** | **- 5.7%** |

*Table A Averaged of the differences in the retrievals of $R_{vf}$ between AERONET and GRASP-AOD algorithms separated for divers $R_{vf}$ ranges*

As commented by the referee, there is certain bias between GRASP-AOD and AERONET retrievals which increases for higher values of the radius: from only -1.4% (when $R_{vf}$ < 0.14 µm) to -12.7% (when $R_{vf}$ > 0.23 µm). We have added some discussion in the text based on these results.

Regarding the reason, we think that there is a global loss of sensitivity as the fine radius increases. If we analyze the variation of the extinction coefficient in function of the size parameter (many examples from the literature, see for instance figure 2.10 of

Lenoble et al. 2013 or Figure 3 from Tonna et al. 1995), we observe that there is a strong variation from $\chi$=0.5-2.5 which becomes smoother from $\chi$>2.5 when the extinction coefficient is around its maximum. For radii around 0.14 $\mu$m the size parameters for all the considered wavelengths are between 0.6 ($\chi(\lambda$=1020nm)) and 2.2 ($\chi(\lambda$=340nm)). At 0.23 $\mu$m all the ultraviolet channels are already out of this "maximum sensitivity region" ($\chi(\lambda$=500nm) =2.9) and the situation gets worse around 0.3 $\mu$m, where even at 670nm ($\chi\approx$2.8), we are out of the maximum sensitivity region. Note here that even in the latter situation, the sensitivity is much higher than the one observed at coarse mode. Even if it is not ideal (as for $\chi$=0.5-2.5) there is still much larger variation compared to the one observed at $\chi$>7 (radii from 1 $\mu$m). In fact, the RMSE is only 0.04 $\mu$m (17%) even if we consider only the retrievals with $R_{vf}$>0.23 $\mu$m. In next reprocessings, we could add some higher initial guess values ($R_{vf}$=0.35 $\mu$m) that may reduce a bit the bias. Nevertheless, the data dispersion would be still higher for larger radii as a consequence of the loss of sensitivity explained here.

Also it is necessary to provide some analysis and discussion of the two distinct populations of the coarse mode radii in the top row plots in Figure 9. I suspect that the larger radii population is from fine mode dominated cases and the lower radii cluster from dust dominated cases, but this needs to be analyzed. If this is the case then the claim for higher accuracy that you imply is somewhat suspect since the accuracy of the coarse mode radii when fine mode dominates the signal is VERY low due to very low coarse mode AOD resulting in very little coarse mode information content in the spectra of total extinction AOD. Additionally, you have again neglected to include information from the study of Sinyuk et al. (2020) that shows that the accuracy of the retrieval of coarse mode radii is much less than that for fine mode aerosol.

As stated from the beginning of the subsection (and also a conclusion from Torres et al. 2017) the sensitivity of the coarse mode of only AOD measurements (in the spectral range that we use) is very little, and an accurate characterization is only possible if we have a priori information on $R_{vc}$ values. In this sense, we do not claim for higher accuracy if fine mode dominates, we just state that even if coarse mode dominates the situation does not improve.

Explain why the effective radius of both modes combined are analyzed at all in this paper. I have never seen a published peer-reviewed paper that shows the value or justification in combining the information from both modes into a total effective radius and total volume concentration value. If you have information that shows the value of these combined mode parameters, then please discuss it in the text plus provide references in order to convince the reader of their value. The separate fine and coarse mode parameters on the other hand have much value and have been utilized in numerous published papers in the scientific literature.

Certainly, analyses of aerosol properties are rarely given in terms of effective radius and/or total volume concentration if other (more detailed) parameters are available. More specifically, if the aerosol characterization comes from AERONET retrievals the authors typically choose the detailed information for each mode as pointed out by the referee. Nevertheless, the effective radius is commonly used in radiative transfer codes (Lenoble et al. 2013, Mishchenko et al. 2002). For instance, for those codes where aerosol size distributions are assimilated as gamma functions the effective radii are the main parameters. We understand that this use explains why AERONET aerosol retrieval algorithm still gives $r_{eff}$ (and its variance) for all the size distribution retrievals. Moreover, based in this "more basic" representation of aerosol size distributions, LET techniques used at lidar retrievals, or more recently used for only $\tau$ measurement retrievals (works cited in the paper such as Perez-Ramirez et al. 2015 or Kazadzis et 2014) give $r_{eff}$ and total concentration as main outputs.

Please quantify what you refer to as 'good capacity' of the GRASP-AOD retrieval of fine mode radius in the Conclusions section. For the Rvf the uncertainty of GRASP-AOD is ∼0.023 micron for fine mode dominated data while for the AERONET Dubovik algorithm almucantar retrievals the accuracy is ∼0.006 for AOD(440)>0.2 for the fine mode observations (large AE). You lack references to the values of Rvf and Rvc from Sinyuk et al. (2020) as a way to compare the accuracy of these retrievals (see Fig 27 for example for the fine mode sites Rvf uncertainty).

As discussed before, we have added a paragraph about that in the conclusions.

On a positive note: You should note that with the newer Cimel instruments the cross scan in the solar aureole is taken with every AOD spectra measurement sequence as a cloud screening data set for the detection of cirrus. This in effect provides aureole sky radiance values for every AOD measurement made with these newer Cimel instruments. This could provide a potentially powerful addition to your retrievals and should be explored for even the fine mode dominated cases to assess any impact of this added aerosol information.

It sounds very good and GRASP-AUR application would benefit from these systematic measurements. We do not expect big improvements in the characterization of fine mode since the maximum information contained concerning fine mode is between 60-100° of scattering (apart from AOD measurements). Nevertheless, the characterization of coarse mode would be definitely much better as shown in subsection 4.3 (before in section 4.2). We hope to coordinate ourselves with AERONET staff and try to use the new measurements that the referee mentions.

Specific Comments:

Line 9: Misspelling of 'diverse'

Corrected. Thank you.

Line 20: What about for low AOD cases? Sinyuk et al. (2020) show that the fine mode radius is retrieved very accurately down to very low AOD.

Discussed in the conclusions.

Line 21: Should be AE>1.2. Seems like this is a bit careless to get such a basic statement backwards in the Abstract.

Corrected. Thank you.

Line 23: This is an odd choice of words here: oscillations implies somewhat periodic variability between two states, not sure the authors really mean that here.

Yes, it is true. "Variations" is more appropriate here.

Line 27-28: Strange terminology for presenting statistics. What exactly is the RMSE values of a correlation? Please be clearer and more precise.

RMSE refers to root-mean-square-error which is first mentioned in one line above. We have added in parentheses -RMSE- the first time is cited in the abstract. Please note that RMSE and the rest of the considered metrics for comparison statistics are well-defined in section 2.5.

Line 50: Should be 'continuous' instead of 'continued'.

Thank you for the correction.

Line 54-55: This sentence has some very awkward English and should be re-written. Hard to know the exact meaning as it is now.

Reformulated: *"Aerosol prediction models typically use ground-based radiometer measurements to complete the information coming from satellite sensors (Randles et al., 2017; Rubin et al., 2017).*

Line 61: High accuracy is even more important than the high precision of the sun photometer measurements.

Yes, we agree. It's been corrected, thank you.

Line 73: "cloud processing" would be much more appropriate here than "cloud formation"

Changed. Thank you.

Line 73: 'plums' should be "plumes"

Changed. Thank you.

Line 77: Large solar zenith angles are no longer required with the Hybrid scan in AERONET, see a description of the hybrid scan in Sinyuk et al. (2020).

The use of hybrid scans and the effect on the solar zenith angle requirements is commented later on. Still Almucantar measurements (or similar in other network) are largely the most used in the AERONET network and the comment fits in a general introduction.

Line 110 & line 118: 'punctual studies': this is awkward English, better to choose a different word, perhaps 'specific studies'? However, not really sure what you are trying to say here.

Yes, we wanted to say specific studies. Punctual is a false friend in Spanish.

Line 145-146: This is a very strange and misleading statement. The only cloud screening check from Smirnov et al. (2000) that is also utilized in the V3 cloud screening is the triplet variability check and even then the magnitude of this triplet threshold has been changed plus spectrally limited to longer wavelengths in V3 (see Giles et al. 2019). Other checks are unique to V3 and also V3 is completely automatic, while the V2 cloud screening of Smirnov et al. required an analyst to remove numerous cloud contaminated observations. This sentence needs to be re-written to be more factual and informative.

The differences between V2 and V3 cloud screening is not critical in our study, since all the data used are from V3. We certainly thought that triplet variability check (with new modifications) still screens the largest amount of data in V3. Nevertheless, we have erased the reference to Smirnov 2000 at this point, and we have kept only Giles et al. 2019 in order to avoid misleading information.

Line 146: You need to state that the accuracy of the Level 2 spectral AOD is ∼0.01 and ∼0.02 in the UV (Eck et al. 1999) since highly accurate data is the key to the applicability of the GRASP-AOD retrievals you are discussing.

Done. Thank you.

Line 148-149: You should state here that the fine mode AOD from the Dubovik retrieval is given at 440 and 675 nm, not 500 nm. Since you are describing the data sources in this section you should be more accurate as there is no 500 nm fine mode AOD directly provided by the Dubovik retrieval. Please write how you computed the fine mode AOD at 500 nm from the Dubovik retrievals.

As explained before, we have added this information to the manuscript.

Line 153: 'teen' should be 'ten'

Thank you very much for your comment. We have corrected the mistake.

Line 155: It is common to most Cimels in the network, but the older PHOTONS group polarized Cimel model do not have the 340, 380 or 500 nm channels. Instead they have three polarized 870 nm channels. Five of your 30 selected sites Dakar, Capo Verde, Banizoumbo, Guadaloupe and Beijing do not have the 340, 380 and 500 nm channels for most or all years of this analysis. For Dakar 1997-2008 plus 2010 do not have the 340, 380 or 500 nm channels and for the Capo Verde site most of the record you analyzed 1997-mid 2016 lack these key channels. Additionally the Beijing site has spectral AOD only from 440, 675, 870 and 1020 nm for all the years 2002 through 2015. Guadaloupe lacks the 340, 380 and 500 nm AOD for 1999 through 2008. Banizoumbo lacks the AOD at 340, 380 and 500 nm for the entire measurement record. The spectral AOD information content of these instruments is much reduced compared to the full wavelength range, therefore it is very important that you mention this and address this issue in the analyses of these sites. You should compare your algorithm with and without the 340, 380 and 500 nm channels for a few sites that have the full wavelength suite of channels. Note that the AERONET group did a full analysis of comparisons of the SDA algorithm with various wavelength combinations in order to determine the wavelengths necessary for Level 2 quality retrievals. The SDA algorithm excludes the 340 and 1020 nm channels since the uncertainties in AOD are higher for these wavelengths. The 340 nm filters have been the least stable (temporal degradation) of all the other wavelength filters plus have out-of-band blockage issues in many 340 nm filter batches. At 1020 nm the silicon detector has a large temperature sensitivity and must be corrected using the sensor head temperature, plus there is significant water vapor absorption at 1020 nm that is accounted for from the retrievals made at 945 nm. These two factors increase the uncertainty at 1020 nm relative to the other wavelengths. The lack of discussion of these issues in this GRASP-AOD paper should be corrected.

This is a very interesting comment, and we agree that the number of wavelengths as well as the spectral range play an important role in the quality of the retrieval. In these regards, we did several processing with different configurations including and/or excluding one or several of the following wavelengths: 340 nm, 380 nm, 1020 nm and 1640 nm. The analysis of the results included all the retrieved parameters beyond the characterization of fine mode AOD. For instance, these tests showed us that the use of 1640 nm systematically enlarged the total retrieval error and the uncertainties of some parameters when compared to AERONET. That's why we decided to erase this channel for the final retrieval. Regarding the use of the other three channels, we did not find any conclusive results that recommend us to avoid its use. On the contrary, we noticed a general better agreement for $R_{Vf}$ when comparing to AERONET when using the 340 nm channel.

Additional tests were also conducted regarding the filtering by the retrieval errors (both absolute and relative, and analyzing the total and/or for each wavelength). Please note that all these tests and some previous carried out while the publication of Torres et al. 2017 are quite technical and we considered them out of the scope of a scientific paper. Nevertheless, the experience gained during all these processing/filtering tests allowed us to establish a first GRASP-AOD quality criteria which is summarized in subsection 2.2.

We believe that some of these requirements may evolve in the future, especially if GRASP-AOD products (maybe in combination with GRASP-AUR) will be publicly available (ideally within AERONET community). Certainly, new tests including all the database and considering particular recommendation of AERONET staff will be done. In that scenario, the publication of the tests will be done in the form of a technical report. Here we only pretended to show that the algorithm can be successfully applied to an extended data record and to carry out a first assessment regarding the retrieval quality.

To answer the specific comment on the spectral range of polarized photometer, we believe that the quality of the retrieval of fine mode AOD is assured by the criteria given in subsection 2.2. In these regards, we did not find a significant loss on the retrieval quality for the periods with only four channels (spectral range from 440-1020nm).

To illustrate this idea, figure A shows the comparison of fine mode AOD between GRASP-AOD/AERONET for two of the aforementioned sites, Guadeloup and Beijing, but separating the characterization for the periods with polarized photometers (4 wavelengths) and standard/extended photometers (7 wavelengths). For Guadeloup, we observe similar comparison results in both periods. It is even a bit better when we use 4 channels: RMSE=0.014 (32%) vs RMSE=0.018 (40%). Note that in both periods the average of $\tau(440)$ and Angstrom exponent were similar (4wl period: $<\tau(440)>$=0.16 and $<\alpha>$=0.30, 7 wl period $<\tau(440)>$=0.16 and $<\alpha>$=0.36). For Beijing, the slope and the correlation coefficient are similar for both periods. The RMSE is higher for the case of 4 wavelengths (0.055 vs 0.034) However, though the average of Angstrom exponent was similar in both periods ($<\alpha>$=1.1) the $<\tau(440)>$ was much higher in the polarized period (0.82 vs 0.58) as well as $<\tau_f(500)>$ (0.61 vs 0.41). That's why in relative terms the differences are quite similar 9.2% (4 wavelengths) vs 8.1% (7 wavelengths).

At this point, we would like to emphasize that the correlation obtained at Banizoumbou – which had a polarized photometer for the entire measurement record - is similar (same RMSE and in relative terms) to the one obtained at Solar Village - site with

similar characteristics but with a standard photometer – as pointed out in the manuscript.

Finally, we were also aware that some wavelengths may have larger errors in the AERONET AOD product. After a discussion with T. Eck while publishing the paper Torres et al. 2017, we decided to account for a different uncertainty at each wavelength which is possible in the multiterm LSM formulation of GRASP code. Thus, we assume double uncertainty for the wavelengths 340, 380 and 1020 nm.

[Figure]

*Figure A. Fine mode characterization for different spectral ranges*

Line 162: Are these multi-year averages computed from daily averages or from all individual instantaneous vales weighted equally? Averaging daily first and then monthly gives a more representative values of the monthly and annual aerosol loading. It is important to clearly write in the paper how you computed these averages.

No, they are not. We understand the importance of averaging first by days and then by months in a classical climatological analysis, however it was not the propose in this

study. Instead, please note that we have done a point-by-point validation which may partially justify the raw averages.

Line 197-198: This is not really true. The Lanai site does not have any L2 retrievals for refractive index since AOD(440)<0.4, but it does have very many L2 retrievals for the size distributions.

Thank you for the comment. We have corrected that in the manuscript.

Line 203-204: Please provide a sentence or two to describe how the options for the dominant mode radii initial guesses change as a function of Angstrom Exponent. I do not see this for the coarse mode as for coarse mode dominated cases AE<0.6 in Table 2 as there are only 2 static choices of coarse radius while for mixed modes 0.6<AE<1.2 there is one static and one dynamic coarse mode radius.

Thank you for comment. Checking the text, we have not done that for the fine mode either. The use of dynamic choices for AE>0.6 for coarse mode radius was already done in Torres et al. 2017. The idea is similar in both studies: the initial guess decreases with AE, and for high AE, it also increases with $\tau(440)$. For low AE, we have not seen the need to vary the initial guess values. In this study, we have given extra choices to improve the retrieval. It may have a little positive effect even though the information contained is quite limited as indicated along the text. In the fine mode, however, we have seen necessary to use the multiple-choice initial guess. In any case, we think that the information given in the Table 2 (and the reference to Torres et al. 2017) is enough for the reader and it does not worth going into these details.

Line 205-206: If the standard deviation (width) of each mode is fixed, then you need to give these values here instead of forcing the reader to look them up in another paper.

Thank you very much. We have added the values.

Line 220-222: Please explain the fitting here in more detail. I assume you compute spectral AOD based on the retrieved size distribution plus the assumed refractive indexes and then compare this to the measured spectral AOD. A written discussion in the text is needed.

Yes, it is exactly like this. We have added a short sentence explaining that.

Line 244-245: You need to be more precise here in your explanation for the lack of SDA retrievals at L2 for these sites that had old style polarization Cimels with only 4 wavelengths of measured AOD data. The reason for no L2 SDA retrievals is the lack of 380 and 500 nm AOD values for the instrument types deployed at these sites. You need to prove that the GRASP-AOD retrievals give the same values for 4 channel AOD input versus 7 channel AOD input. This should be especially important at the

Beijing site which is fine mode dominated and therefore has much greater non-linearity in the AOD spectra in logarithmic space. For coarse mode desert dust sites this will not matter nearly as much as the AOD spectra is relatively flat with little non-linearity in logarithmic coordinates.

Thank you very much for your comment. Even though it was not clearly explained, we had a link to SDA Level 2.0 criteria in AERONET webpage. We agree with the comment of the referee that an explanation regarding polarized photometer was needed and we have added that in the manuscript. Regarding the comment on Beijing, we have previously shown the comparison with 4 and 7 wavelengths.

Line 255: It should be noted that the fine/coarse mode radius separation value is the same for Version 3 as it was in Version 2.

Yes, it is done now, thank you.

Line 258-259: Please add "for each mode as well as for the entire size distribution".

Thank you very much for your comment. We have added it to the text.

Line 260: This is the wrong vocabulary word ('mechanical') here. I suggest that this word can be eliminated and the sentence will be clearer. I suggest: "The separation between fine/coarse mode..."

Ok, thank you very much.

Line 262: How do you make this interpolation? In log-log space by Angstrom Exponent relationship, or by 2nd order fit of AOD in log-log space which is the most accurate methodology.

We have used just log-log space by Angstrom Exponent relationship which we think it is a good approximation in first order giving that 440nm are 500nm are quite close.

Line 267: This is just way too simplistic an estimate for this paper. The number of AOD spectra measured per day in AERONET depends on site latitude and day of year, resulting in differing number of day-length hours. In addition, the newer instruments are set to take 5-minute sampling interval data versus 15-minute sampling intervals in the old Cimels for direct sun AOD observations. More details on the variable number of AOD measurements per day in AERONET are required in a paper that utilizes AOD spectra as the primary input parameter.

We are aware of the variation on the number of AOD measurements, depending on the site, instrument model or even the particular needs of site manager (since additional measurements can be added apart from the standardized sequences). We do not think that this information is so important in this paper. The number of thirty/forty measurements per day is a quite reasonable estimation for the period analyzed and

the sites accounted in this study. Nevertheless, we have added in parenthesis that this number can be different depending on the site latitude or the type of instrument.

Line 273: This is an inaccurate statement since some sites only have the 440, 675, 870 and 1020 nm AOD while most other sites add the 340, 380 and 500 nm channels to those.

Line 275-276: Except as you noted that the SDA does not make a retrieval when the 380 nm AOD are missing.

We answer here both questions.

First, we would like to say that we have treated this issue before, and we have already added that the polarized photometers do not have the 380 nm which is a requirement for SDA to explain the percentage of SDA in some sites. Here, we discuss about "Match-up methodology" and the different data measurements for the three algorithms. The truth is that SDA can be applied (as well as GRASP-AOD) to all AOD measurements, which includes polarized photometer. As a matter of fact, there is SDA data in Level 1 for the polarized photometers. The quality criteria (and the consequences) were already discussed before.

Line 322-327: No real surprise here as these 3 sites have the highest AOD levels in the entire AERONET network. I suggest adding the average AOD values in the table and plotting the RMSE versus this average AOD. For the La Reunion site you should add the phrase: "...because the AOD were lowest for this site."

Ok for La Reunion. Regarding the highest AOD, we understand that the fact that the site changes for the different comparison is interesting.

Line 350-351: Please include an investigation and explanation of some cases in the two branches of the Fig 5 plots for AE<0.6 of GRASP-AOD versus AERONET and SDA versus AERONET (Dubovik). An attempt should be made to explain these two data populations and why they diverge as fine AOD increases.

The reason is the third mode presented in some desert dust sites (section 4.1). For some of these retrievals the midsize mode is relatively high which makes that the minimum value of the volume size distribution in the cutoff radius range (from $0.439\ \mu$m to $0.992\ \mu$m) is found either at $0.992\ \mu$m or at $0.756\ \mu$m. Therefore, the AERONET retrieval assigns the midsize mode completely (or mostly) to the fine mode while the SDA and GRASP-AOD do not. We present 4 examples of this issue (retrievals at Ilorin site on 1 February 2000) at new figure 12. Moreover, in Table 9 we give the $\tau_f(500)$ values (estimated by GRASP forward code using as input the aerosol properties retrieved by AERONET) for different cutoffs. This and other results are discussed at the new subsection 4.1.2.

Line 371: Please mention that this is a quality control issue for SDA due to insufficient AOD wavelengths for highest accuracy of the retrievals.

We underlined that already in subsection 2.3.1, and here we referred to the subsection.

Line 381: It is not just 500 nm but also 340 and 380 nm that are not available in the old Polarized Cimels. Please add this to the text. To prove the level of robustness you have claimed, for Beijing you need to run the GRASP-AOD retrievals for the full 7 channels (340-1020 nm) for years when this type of Cimel was operating there and then subsequently run the GRASP retrievals with only the 440, 675, 870 and 1020 nm data as input for these same exact measurement scans. Only this direct comparison of the same AOD spectra and almucantars but with different spectral channels used as input can really determine just how robust the 4 channel GRASP-AOD retrievals are.

We have added 380 nm at this point of the text.

The test aforementioned was presented before for Beijing and Guadeloup by comparing the performance (against AERONET retrieval algorithm) with 4 wavelength periods and 7 wavelength periods and we have proved the robustness of GRASP-AOD retrievals. We are open to do more tests beyond the study here.

Line 413-415: It should be noted that the retrieval of the fine mode radius when the coarse mode dominates (AE<0.6) also has a large uncertainty in the Dubovik retrieval with sky radiance information, see Sinyuk et al. (2020). Therefore, the lack of correlation with GRASP is also due largely to very weak information and thus large uncertainty for fine radius in the AERONET almucantar retrievals for coarse mode cases.

Thank you very much, we have added that information here.

Line 423: This is an incomplete sentence here should probably be deleted.

Yes, we have erased it.

Line 431: Should change 'column' to 'row' here.

Yes, thank you very much.

Line 437: Please discuss the reasons for this systematic underestimation by GRASP which gets worse as fine mode radius increases in Figure 7 for all sites shown, even for the best conditions of high AOD and high AE.

It has been done as commented before.

Line 467: Please discuss the reason for the 2 populations that are obvious in most of the plots of Figure 8.

We think that they are due to the uncertainty when coarse mode dominates since it is less obvious when AE>1.2. However, we think that further tests should be done to include that statement in the manuscript.

Line 482: It is interesting that you mention 1640 nm here since the GRASP-AOD retrieval does not use this wavelength of AOD data. Theoretically inclusion of the 1640 nm AOD should indeed provide more information on the radius of the coarse mode, so you should discuss that here.

It is coming from the study Torres et al. 2017 where 1640 nm was included in some tests. We have changed to 1020 nm here.

Line 485: Please be clear here that these are AERONET climatological values.

Ok, thank you.

Line 695: This is the wrong word choice ('axes') here. Although the writing is in general relatively good from the English grammar and vocabulary aspects, please have a native English speaker review the manuscript to catch the various instances of awkward phrasings and/or poor vocabulary choices.

It is true. We have changed it for points. We think that we will ask AMT to revise the manuscript to improve the English.

Line 701: Nothing involving real data is ever a perfect correlation. Please give the exact value of correlation here even if it is very close to 1.

Yes, it is 0.997 in this case.

---

## Author Comment (AC3) · 26 Feb 2021

General comments:

The aim of this paper is to show that the GRASP-AOD code has the potential to be used for large scale datasets either for aerosol climate studies or for near real time modeler needs. The validation based on 2.8 million GRASP-AOD retrievals using AERONET AOD observations from 30 sites during 20 years makes the work robust enough to reach appropriate conclusions. The paper is to long taking into account the methodology used, the results and the prior knowledge published about this type of AOD inversion codes. I suggest making a synthesis relying on the bibliography already published, including the new considerations used that can improve this type of AOD inversion codes (comparative and differences with other papers already published). The paper is well written and into the scope of AMT. I recommend the publication of this paper, but there are some issues should be addressed prior to publication. The Editor will judge.

We would like to thank the anonymous referee 3 for reviewing the manuscripts and the positive comments.

The AOD inversion codes have used in different papers from many years. These type of inversion codes are based on the aerosol scattering equation that express the dependence of the spectral variation of AOD on the aerosol size distribution, and also depend of the $Q_{ext}$ parameters (particle extinction efficiency factors), which in turn depend on the wavelength, the refractive index and particle radius. As example, King et al. (1978) already pointed out that the definition of the particle radius interval on which the inversion method can be correctly used, and the assumption of realistic refractive index values are the most crucial points in any rigorous application of inversion methods applied to spectral series of the AOD. On the other hand, the independent information content on the optical characteristics of columnar aerosols is contained primarily in the particle radius interval from 0.1 to 2 microns, approximately, for AOD measurements covering spectral range 340-1020 nm. On the other hand, the iterative procedures modified the radius interval within the prescribed ranges, and the best results were obtained for reduced radius range. In this sense, with this type of codes the results are limited to the accumulation mode. On the other hand, some AOD

inversion algorithms use a single refractive index, while the true is dependent on wavelength. The assumption of an a priori defined refractive index in the AOD inversion procedures may lead to very different derive size distributions, but other authors (e.g., Yamamoto and Tanaka, 1969; King et al., 1978; González and Ogren, 1996) show that the shape of the retrieved aerosol size distribution is not substantially altered as a result of using such assumptions. In this sense, this paper should take into account previous work and show the improvements that can be made. Taking into account previous results, obviously these type of inversion algorithms would not work well for coarse particle modes just considering only the AOD spectral values. Spectral aureole data (sky radiances) are required to achieve good results in coarse mode.

We recognize the knowledge of the referee regarding aerosol property retrievals. We have added some of the comments from this paragraph along the document (especially in the introduction and new section 4.3). We reckon that this update has enriched the article. However, we would like to add some points in the discussion:

- The article of King et al. 1978 uses the interval 0.1-4.0 µm when inverting AOD measurements from 0.440-1.030 µm. Moreover, the authors add the following comment while setting their election: "Although this matter (referring to the election of the interval) has been considered by Yamamoto and Tanaka (1969) for both Junge -and Woodcock-type aerosol size distributions, it is very important to realize that there is no absolute rule which determines the radii limits having the most significant contribution to the attenuation measurements… Since the size distribution function is not known in advance, it is apparent that occasional trial and error is required in order to determine the radius range over which the inversion can be performed".

- In the work by Gonzalez and Ogren (1996), the interval is limited between 0.1-2.0 µm, maybe since the spectral range considered is slightly smaller: 0.35-0.88 µm. Note here that the claimed low sensitivity to radii variation (or we shall say the ratio between the radius and the wavelength known as size parameter) does not mean that the contribution of coarse particles to estimate the total extinction can be neglected to characterize the aerosol optical depth. Actually, the fact of reducing the radius interval to 0.1-2.0 µm at González and Ogren (1996) originated an irreal-excess of particles at smaller radii that tried to optically compensate that large particles were dismissed (see tests done with synthetic measurements, examples in fig.3 or fig.4). This "fake" effect adds more uncertainties in their size distribution characterization (moments, effective radius, etc.). We certainly admit the low sensitivity to retrieve coarse mode size parameters, but the effect of ignoring its contribution in the retrieval would create errors in an overall characterization of size properties.

- We agree that a basic analysis about the variation of $Q_{ext}$ functions would show that the coarse mode radii are very close to the geometrical-optic region (accounting the spectral range used in the study), and therefore, the $Q_{ext}$ values slightly vary from the asymptotic value of 2. The sensitivity of the AOD measurements to those radii is very small. This fact is not hidden along the work and it affects the characterization of the coarse mode as shown in the paper and as largely commented in Torres et al. 2017. But this does not mean that the optical extinction due to these particles is zero (see for instance the Modified Kernel Functions for Optical Thickness represented as function of the radius in figure 5.1b of King and Dubovik 2013). The fact of neglecting its contribution creates undesirable effects as the ones found in the work by González and Ogren (1996).

- The inversion strategy proposed here, which the solution is predefined by two log-normal functions, presents some advantages with respect to previous strategies (which resided in the multiplication of a rapid varying function - typically Junge - and another of slower variation - which is the one retrieved at each predefined interval -). These advantages are presented along the manuscript and they cannot be just summarized by a compilation of previous results. The most important are recapped here:

    1. It allows to separate the optical contribution of the modes. As the SDA, GRASP-AOD code separates fine mode optical depth (highly dependent on the wavelength) from the coarse mode contribution (almost spectrally independent in the range 340-1020 nm). We have largely proven the robustness of this retrieval through comparisons with AERONET retrievals.

    2. It allows to accurately characterize the fine mode radius under certain conditions ($\tau(440)>0.2$ and $\alpha>1.2$). The RMSE compared to AERONET retrieval (=$0.023\mu m$) is quite good considering the information contained. This detailed characterization represents an important novelty compared to the forementioned codes, or some others used for only $\tau$ measurements, such as the ones inspired by LET techniques.

    3. The coarse mode contribution is represented by only three parameters (two in fact since the standard deviation is quite constrained) and well characterized in terms of mode optical depth. We are aware that different pairs of $R_{Vc}$ and $C_{Vc}$ produce similar spectral coarse AOD values (larger concentrations compensates an increase of the mode radius), but coarse mode contribution is well accounted by GRASP-AOD. Note that most of the values of $R_{Vc}$ that are retrieved in the paper (AERONET

retrievals) are under the limits established by King et al. 1978 (<4.0 µm), the values of the volume distribution beyond this interval are forced/fixed by the log-normal function.

- The effect on the refractive index (due to anomalous diffraction theory of Van de Hulst (Van de Hulst, 1957) as primarily discussed in Yamamoto and Tanaka (1969)) would be commented later. At this point, we would like only to recall that it was already presented in Torres et al. 2017 with some ideas proposed by M. King who was one of the referees of that study.

Lines 85-95. To motive the importance of this work, the authors comments that many AERONET sites are plagued by several months of partial cloudiness (no sky radiance measurements) . . . but later they use climatological values for refractive index and information about radius modes. How it is possible for this type of AERONET stations, and how representative are these values? also for future applications to night measurements. The columnar aerosol properties change from day to night, depend on sources, the air masses transport, the planetary boundary layer high ... Also, a study of the GRASP-AOD sensitivity to the refractive index is needed.

The representativity of the chosen climatological values (based on retrievals with clear sky conditions) would depend on the site. The averaged value strategy presented here should be considered as a first reasonable approach. Though the dependence on refractive index (mostly real as indicated by King et al. 1978 and Torres et al. 2017) of GRASP-AOD application was deeply discussed at Torres et al. 2017, we have added a discussion point (new section 4.2) where we have treated Mongu site (with climatological values around 1.51-0.021i) with a generic refractive index of 1.45-0.005i.

- Line 185. The GRASP-AOD code assumes the refractive index as known. Which one has been chosen for each AERONET station and aerosol type? Can be Included in Table-1? On the other hand, the aerosol type selected for each station (Table 1) can be the more frequent (climatology), but not all ways are the same. As example, the Saharan dust outbreaks. How these facts affect the inversion products?

We have used moving monthly means (2 adjacent months) for all sites using Version3 AERONET aerosol retrieval. We cannot include a table containing all the values since one site would contain already 8x12 values. We present here as an example (Table A), the values found/used for GSFC.

The issue commented in the last lines was more or less discussed at pag. 24 lines 450-455 (at the end of Section 3.2.1 - discussion of $R_{Vf}$ characterization - which has been kept). We have suggested that future reprocessings may use more developed

climatologies (e.g. considering different values for different Ångström exponents) which may improve some of the results obtained in this study.

| | Real Refractive index | | | | Imaginary Refractive index | | | |
|---|---|---|---|---|---|---|---|---|
| | 440 | 670 | 870 | 1020 | 440 | 670 | 870 | 1020 |
| January | 1.443 | 1.437 | 1.435 | 1.429 | 0.004 | 0.004 | 0.005 | 0.005 |
| February | 1.462 | 1.447 | 1.444 | 1.434 | 0.005 | 0.005 | 0.005 | 0.005 |
| March | 1.473 | 1.463 | 1.461 | 1.454 | 0.005 | 0.005 | 0.006 | 0.006 |
| April | 1.467 | 1.455 | 1.453 | 1.447 | 0.005 | 0.005 | 0.006 | 0.006 |
| May | 1.468 | 1.456 | 1.453 | 1.449 | 0.005 | 0.005 | 0.006 | 0.006 |
| June | 1.458 | 1.441 | 1.438 | 1.433 | 0.005 | 0.005 | 0.006 | 0.006 |
| July | 1.456 | 1.440 | 1.436 | 1.431 | 0.005 | 0.006 | 0.006 | 0.006 |
| August | 1.449 | 1.435 | 1.431 | 1.425 | 0.005 | 0.005 | 0.006 | 0.006 |
| September | 1.441 | 1.427 | 1.424 | 1.420 | 0.004 | 0.005 | 0.005 | 0.005 |
| October | 1.432 | 1.424 | 1.422 | 1.419 | 0.004 | 0.004 | 0.004 | 0.004 |
| November | 1.426 | 1.420 | 1.419 | 1.418 | 0.004 | 0.004 | 0.004 | 0.004 |
| December | 1.425 | 1.428 | 1.427 | 1.425 | 0.003 | 0.004 | 0.004 | 0.004 |

*Table A Example of the refractive index values used to run GRASP-AOD. - GSFC site -*

Lines 190-195. If the refractive indices are assumed, what happens, as example, with stations where there are many clouds and cannot be computed with the sky radiance data? There are no data? Do you use the climatological value? How much data have you used to obtain this climatological value, and how is it distributed throughout the year? In order to these results will be realistic, an extensive database should be available and the appropriate refractive index value used for each atmospheric condition. The purpose of this work is to show that the GRASP-AOD application has the potential to be used for large scale datasets.

If not data at all is available, standard refractive indices should be considered. From the moment, that there will be some full AERONET inversions the existing archive of the hypothetical new site could be reprocessed. Further new reprocessing could be done as the climatological database is updated. To run a site with 20 years of data as the examples presented here takes around 6 hours with current processors (no much time needed).

As commented before, we have added a new section 4.2 discussing what happen if climatological refractive indices are not available by reprocessing one site with standard refractive indices.

Lines 480-525. Obviously, the algorithm does not work well for coarse particle mode just taking into account only the AOD spectral parameters and a climatological value of the refractive index. But we already knew these results from the papers published related with these type of inversion codes. The sky radiance data is needed to achieve good results in coarse mode. I think this section should be shortened or removed from

the paper. Also, the last sentence of the abstract is a well-known result and it is not new.

This point was partially discussed before. Nevertheless, we understand the comment of the referee and we have actually considered to erase the subsection. After a discussion with the editor, we have decided to keep it mainly for two reasons: a) Even though the results are similar to previous analysis, the strategy proposed by GRASP-AOD presents itself some novelties that are worth to comment. b) In future works, we will explore in detail the GRASP-AUR application which has the same strategy as GRASP-AOD to represent the size distribution. Certainly, the results obtained by GRASP-AOD in the coarse mode will be taken as a reference in the new characterizations as partly done in new section 4.3 (old 4.2)

Lines 200-225. The criteria are based mostly on analyst's experience. The authors show "Due to the low sensitivity of GRASP-AOD to the shape of the modes. . . we have used strong a priori constraints on the actual values for the standard deviation of both modes. . . in practice, their values are very similar to the given initial guess values". On the other hand, in Line 340 the authors show: "The larger uncertainties observed for Solar Village compared to GSFC can be extrapolated to all sites with coarse mode predominance with respect to the sites with fine mode predominance", and the following lines. Taking into account the papers published so far, it is clear that this methodology can only be applied to places where the fine mode predominates. In my opinion, this work should be drastically reduced, showing only those aspects that can improve the results of the works already published. On the other hand, the usefulness of using climatological values in the a priori assumptions should be better discussed.

The low sensitivity to standard deviation (even to fine mode) was discussed in Torres et al. 2017. We believe that the strategy of bimodal lognormal functions for only AOD measurements is a novelty of the work by Torres et al. 2017. In this sense, the low sensitivity to the standard deviation cannot be summarized from previous works.

Regarding the comments in Line 340 it refers explicitly to the separation of optical depth fine/coarse mode ($\tau_f(500)$). Different works about SDA algorithm (O'Neill et al. 2003 or Eck et al. 2010) obtained similar results as acknowledged along the article.

---

## Author Response (AR2)

*Characterisation of aerosol size properties from measurements of spectral optical depth:*

*a global validation of the GRASP-AOD code using long-term AERONET data.*

*By Torres, B and Fuertes, F.*

**Author's response**

Dear Editor,

We have uploaded the new version of the manuscript. In the new version we have included the three minor revision proposed by the referee#2.

Best regards,